# Protein Design with Agent Rosetta:
# A Case Study for Specialized Scientific Agents

**Jacopo Teneggi** [* 1 2]  **S.M. Bargeen A. Turzo** [* 3]  **Tanya Marwah** [4]  **Alberto Bietti** [1 5]  **P. Douglas Renfrew** [3]
**Vikram Khipple Mulligan** [3]  **Siavash Golkar** [1 6]

[code]  [data]

## Abstract

Large language models (LLMs) are capable of emulating reasoning and using tools, creating opportunities for autonomous agents that execute complex scientific tasks. Protein design provides a natural testbed: although machine learning (ML) methods achieve strong results, these are largely restricted to canonical amino acids and narrow objectives, leaving unfilled need for a generalist tool for broad design pipelines. We introduce *Agent Rosetta*, an LLM agent paired with a structured environment for operating Rosetta, the leading physics-based heteropolymer design software, capable of modeling non-canonical building blocks and geometries. Agent Rosetta iteratively refines designs to achieve user-defined objectives, combining LLM reasoning with Rosetta's generality. We evaluate Agent Rosetta on design with canonical amino acids, matching specialized models and expert baselines, and with non-canonical residues—where ML approaches fail—achieving comparable performance. Critically, prompt engineering alone often fails to generate Rosetta actions, demonstrating that environment design is essential for integrating LLM agents with specialized software. Our results show that properly designed environments enable LLM agents to make scientific software accessible while matching specialized tools and human experts.

## 1. Introduction

Autonomous agents built on large language models (LLMs) are becoming increasingly capable of executing complex, multi-turn tasks that demand the emulation of reasoning and the capability to use tools. A key strength of these agents lies in their ability to write, debug, and execute code (Chen et al., 2021; Rozière et al., 2024; Guo et al., 2024), making them well-suited for aiding the automation of scientific discovery workflows that rely on code-based interfaces, spanning chemistry (Bran et al., 2023; Arlt et al., 2024), mathematics (Shao et al., 2024), physics (Li et al., 2025; Microsoft Research AI4Science, 2023), and biology (Huang et al., 2025; Ghafarollahi & Buehler, 2024; Qu et al., 2025; Mitchener et al., 2025). By combining artificial reasoning with iterative feedback, agents offer new ways to address scientific problems beyond the reach of current machine learning (ML) methods.

Protein design—and, more generally, heteropolymer design—is a central scientific challenge, with broad implications for developing nanomaterials (King et al., 2012; 2014; Hsia et al., 2016; Gonen et al., 2015), medically- or industrially-relevant enzymes (Gordon et al., 2012; Pultz et al., 2021; Siegel et al., 2010; Jiang et al., 2008), and drugs (Mulligan et al., 2021; Hosseinzadeh et al., 2021; Mulligan, 2020). While ML-based protein modeling methods such as AlphaFold (Jumper et al., 2021; Abramson et al., 2024), RFdiffusion (Watson et al., 2023), and ProteinMPNN (Dauparas et al., 2022) have advanced the field, they focus on the 20 canonical amino acids, are specialized to particular design pipelines, and depend on large training datasets. In these models, small modifications of the task (*e.g.*, inversion of a chiral center) often result in poor performance, for lack of relevant training data such as non-canonical D-amino acid peptides (Childs et al., 2025). In contrast, the Rosetta Macromolecular Modeling Suite (Leaver-Fay et al., 2011), built on a largely physics-based energy function (Alford et al., 2017), requires minimal training data and can accurately model proteins containing non-canonical building blocks that have not been observed in experimentally-determined structures (Renfrew et al., 2012; Drew et al., 2013; Mills et al., 2013;

---

[1]Polymathic AI [2]Johns Hopkins University [3]Center for Computational Biology, Flatiron Institute [4]Google DeepMind (work was done while at the Simons Foundation) [5]Center for Computation Mathematics, Flatiron Institute [6]New York University. Correspondence to: Jacopo Teneggi <jtenegg1@jhu.edu>.

*Proceedings of the 43rd International Conference on Machine Learning*, Seoul, South Korea. PMLR 306, 2026. Copyright 2026 by the author(s).

Bhardwaj et al., 2016; Mulligan, 2022b; Watkins et al., 2017; Renfrew et al., 2014; Dang et al., 2017). This dramatically expands the accessible design space and has enabled advances in, for example, drug development (Hosseinzadeh et al., 2017; Bhardwaj et al., 2022; Mulligan et al., 2021; Hosseinzadeh et al., 2021; Mulligan, 2020). Yet effective use of Rosetta demands not only deep biophysical expertise and coding proficiency, but also familiarity with its unconventional input formats, such as RosettaScripts (Fleishman et al., 2011). Because these barriers limit accessibility—a major aspect of computational macromolecule design (Mulligan, 2021)—hybrid approaches that combine ML models with general physics-based methods could be transformative. Here, we identify an opportunity for LLM-based agents: by combining multi-turn reasoning with code generation and tool use, such agents could accelerate Rosetta protocol development even for advanced users, while making Rosetta's powerful capabilities available to non-expert users in the broader scientific community.

We develop *Agent Rosetta*, a single-agent, multi-turn agentic framework that follows a user-defined brief and progressively refines protein designs through structured dialogue and feedback with an OpenAI Gym-like RosettaScripts environment (Brockman et al., 2016; Fleishman et al., 2011). Our approach leverages the biophysical knowledge and code-writing capabilities of the base LLM through artificial reasoning, and combines it with the constraints of XML scripting in Rosetta. We find this framework an interesting test case for building autonomous agents aimed at real-world scientific workloads. For example, we find that even though many examples of RosettaScripts are included in LLM training corpora, the rich and complex RosettaScripts syntax—unlike that of mainstream software packages—creates many challenges. In specific cases, we found that these difficulties proved significant enough that prompt engineering alone was insufficient to reliably extract scientifically sound outputs from frontier LLMs.

We evaluate Agent Rosetta on two real-world protein design pipelines: designing sequences that stabilize polypeptide backbone conformations with canonical amino acids only, and inclusion of a non-canonical amino acid (NCAA) in the core of a given protein. For the former task, we compare the performance of the agent with ProteinMPNN (Dauparas et al., 2022) and BoltzGen (Stark et al., 2025)—two specialized ML models for protein design—and with two human baselines of varying RosettaScripts complexity. On the latter task, given ML models restriction to canonical residues, we can only compare the agent with a human baseline. For both tasks, we use AlphaFold 3 (AF3) and molecular dynamics (MD) simulations for final validation.

We briefly summarize the main contributions of this work:

- We introduce Agent Rosetta, an LLM agent capable of executing broad user-defined tasks via multi-turn interactions with a tailored RosettaScripts environment that ensures a robust integration of the agent with Rosetta.

- In building our agent, we found that prompting alone can fail to bridge general-purpose LLMs with specialized scientific software. Instead, tool and environment design are crucial.

- We demonstrate that our framework enables a broad range of real-world design pipelines. In particular, we evaluate Agent Rosetta on fixed-backbone sequence design with canonical amino acids only and on inclusion of a non-canonical residue in the core of a protein.

- We compare Agent Rosetta with specialized ML models and human written protocols. Validation with AF3 and molecular dynamics simulations confirmed that Agent Rosetta is competitive with specialized ML models and biomolecular scientists.

This work represents a step towards understanding the merits and limitations of LLM-based agents for scientific tasks. Our results demonstrate that coupling frontier LLMs with domain-specific scientific tools can yield flexible frameworks that achieve performance competitive with deep learning models trained for narrowly defined design tasks and human scientists. This suggests that generalist agentic approaches can combine accuracy with versatility, opening new opportunities for scientific discovery.

Our work focuses on low-level integration of an LLM agent with specialized scientific software, reflecting deployment in real-world pipelines. We differ from existing works addressing high-level planning or orchestration of several LLM-friendly tools (Schmidgall et al., 2025; Ghafarollahi & Buehler, 2024; Bran et al., 2023). For example, Agent Rosetta could be invoked by ProtAgents (Ghafarollahi & Buehler, 2024) to perform specific tasks. We refer readers to Appendix A for a detailed discussion of prior works on scientific agents, ML for protein design, and heteropolymer design with Rosetta.

## 2. Agent Rosetta

The Rosetta modeling suite (Leaver-Fay et al., 2011; Leman et al., 2020a) is a general collection of C++ libraries and programs supporting a wide range of applications such as heteropolymer structure prediction, docking, and design. A central component of Rosetta's methodology is the use of Monte Carlo algorithms to optimize an *energy function* that approximates the true energy of a molecular conformation (Alford et al., 2017). Optimization is performed by exploring conformation space (sampling favorable candidate folds or docked configurations given a fixed sequence of

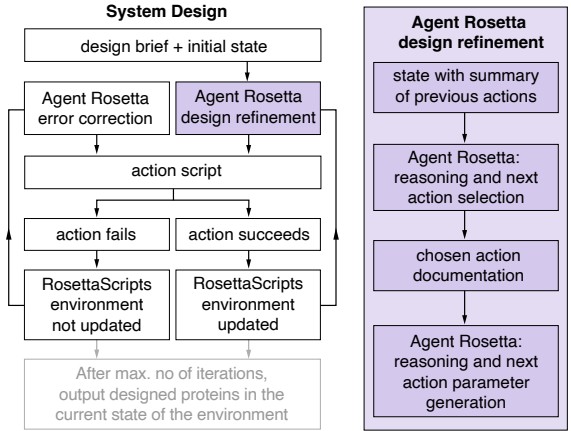

**System Design**

design brief + initial state

Agent Rosetta error correction | Agent Rosetta design refinement

action script

action fails | action succeeds

RosettaScripts environment not updated | RosettaScripts environment updated

After max. no of iterations, output designed proteins in the current state of the environment

**Agent Rosetta design refinement**

state with summary of previous actions

Agent Rosetta: reasoning and next action selection

chosen action documentation

Agent Rosetta: reasoning and next action parameter generation

*Figure 1.* Illustration of our multi-turn agentic system.

chemical building blocks) or sequence space (sampling favorable sequences of chemical building blocks given a fixed heteropolymer backbone conformation) (Bonneau et al., 2001; Kuhlman & Baker, 2000). To perform this exploration, Rosetta defines *Movers*, actions that alter a structure to guide the sampling process and navigate the energy landscape, as well as *Filters*, modules that measure a structure's property and decide whether to continue or to start over.

Scientists compose design protocols with multiple Movers and Filters that often require hours to days to complete, depending on polymer length, structural complexity, and exhaustiveness of exploration of the optimization space. As a consequence, researchers must commit to execute entire protocols before assessing their success, potentially foregoing valuable information in intermediate states.

In contrast, LLM agents can interact with Rosetta more frequently, selecting the next Mover immediately after the previous one completes. This enables adaptive guidance of the design process informed by the properties of intermediate structures. When randomized across parallel runs, this approach also facilitates a broader and more diverse exploration of protocols than is typically feasible for scientists. To realize these advantages, we introduce Agent Rosetta, an agentic framework that successively determines the best action, as illustrated in Figure 1.

We now describe Agent Rosetta's system design and the main components of the multi-turn interaction protocol. We include all prompts in Appendix B.

**System prompt.** Given the online presence of many Rosetta scientific papers, much Rosetta lecture material, and considerable Rosetta software documentation, it is clear that frontier LLMs have had exposure to Rosetta and RosettaScripts during training. Hence, in the system prompt, we directly instruct the agent to act as a RosettaScripts coder supporting a team of scientists. We include a summary of

the available actions in Python-style docstrings format, and the rules of the interaction protocol and of the structured response format (see Appendix B.1 for the system prompt).

**Design brief.** Rosetta can be used for a variety of tasks. For example, it can modify the amino acid sequence of natural proteins to alter their function (Thyme & Baker, 2014; Moretti et al., 2016; Thieker et al., 2022), construct antibodies that can bind to particular antigens (Adolf-Bryfogle et al., 2018; Schoeder et al., 2021), or design exotic, synthetic peptides that can bind to target proteins of therapeutic interest (Mulligan et al., 2021; Hosseinzadeh et al., 2021). Agent Rosetta is instructed to follow a design brief that states these goals in terms of both quantitative and qualitative objectives. For example, an entry-level design brief could be to *"Replace all amino acids in ubiquitin's sequence so that its core is well-packed, the protein structure is energetically stable, and it adopts the same fold as the initial protein"*. Together with the design brief, the user defines an initial structure: in our example, this would be the Protein Data Bank (PDB) file 1UBQ (Vijay-Kumar et al., 1987) describing the 3D conformation of ubiquitin. In other tasks, such as *de novo* design (Koga et al., 2012), the agent could first generate the initial backbone conformation with internal Rosetta modeling tools or generative ML models like RFdiffusion (Watson et al., 2023), and then proceed to designing a stabilizing amino acid sequence for that conformation. See Appendix B.2 for the design brief prompt, and Appendices E.1 and F.1 for the briefs used in our experiments.

**Environment state and task metrics.** The Monte Carlo search algorithms in Rosetta make design protocols stochastic. To account for this, at each step of a trajectory, we execute Agent Rosetta's action across several parallel processes, generating an ensemble of designs rather than a single one. Each design is stored as a data structure called a *Pose* (Leaver-Fay et al., 2011), which can be written to a PDB file. Pose objects are complex, and PDB files can easily contain hundreds of lines each, making them impractical as raw context for the agent's LLM—especially for ensembles.

Instead, we make this state legible to the agent with task-dependent *surrogate metrics* that approximate the energetic and structural stability of the candidate designs. The choice of metrics is not unique, as stability remains difficult to quantify without experimental validation. In this work, we pick metrics that are representative of the way scientists use Rosetta. In particular, to estimate packing of the hydrophobic core (a key feature of stable proteins), we compute radius of gyration (Leaver-Fay et al., 2011), cavity volume (Sheffler & Baker, 2009), and buried unsatisfied hydrogen bonds (Mulligan et al., 2021). These metrics help identify anomalies in protein structure modeling or design protocols.

Furthermore, we identify common problematic residue identities and positions by analyzing the per-residue van der Waals energy (*i.e.*, the `fa_rep` term in Rosetta's energy function) and the favorability of each residue's combination of $\phi$ and $\psi$ mainchain dihedral angles given its Ramachandran map (*i.e.*, the `rama_prepro` term). These terms are used by researchers to detect side-chain clashes and strained backbone conformations, respectively. This way, the agent knows the most common problematic residues.

Finally, we enrich the state with task-specific *progress metrics*. For example, for fixed-backbone sequence design with canonical amino acids only, we use ESMFold to predict the native folds of the candidate designs, and we compute the root mean squared deviation (RMSD) between the predicted fold and the initial backbone conformation. Since the goal is to design a sequence that gives rise to the desired fold, a successful design should yield a small RMSD between the desired structure and the structure predicted from the sequence. We complement the RMSD with the average pLDDT of the $C_\alpha$ atoms: values close to 1 indicate high confidence in ESMFold's prediction, providing further evidence for stability. We include the RMSD and pLDDT in the state as easy-to-compute but imperfect proxies for stability. We note, however, that ESMFold (like many ML models) may not reliably capture the stability of Rosetta designs that deviate from its training set of naturalistic sequences.

For example states see Appendices E.3 and F.4.

**Available actions.** RosettaScripts accepts XML scripts as structured actions, but having the agent write them from scratch leads to lengthy, error-prone responses that are costly to correct. Instead, we use RosettaScripts' XML schema to define *types* of actions, whose templates are filled with the parameters generated by the agent. Not only does this allocate Agent Rosetta's reasoning budget towards the scientifically-relevant parameters, but also it guarantees semantically correct actions where prompting alone fails.

We define three types of actions that fit several design pipelines and allow the agent to explore a vast sequence space (see Appendix G for an example of each action):

- `rotamer_change` uses Rosetta's *FastDesign* Mover (Bhardwaj et al., 2016) to sample energy-minimizing side chains of a molecule, relaxing the backbone conformation in the process. The agent guides FastDesign's sequence optimization by generating amino acid composition constraints that are incorporated into Rosetta's `aa_composition` scoring term during design (Hosseinzadeh et al., 2017; Mulligan, 2022a). With `aa_composition`, scientists can realize actions such as *"Design the core of a protein with at most 15% polar residues and at most 5% glycine. Do not allow any hydrophilic residues in the core, and keep the boundary of*

*the protein unchanged"*. Beyond composition constraints, Agent Rosetta can restrict the base residue types allowed during design with *TaskOperations*, or disable design completely, leaving *repacking* with identity fixed. Finally, the agent can compose penalty blocks and TaskOperations in different regions of a molecule using *ResidueSelectors*, Movers that select residues by identity, burial, secondary structure, or other properties.

- `backbone_change` implements three RosettaScripts Movers that randomly perturb the torsion angles of the backbone to modify its conformation: `Small`, `Shear`, and `Backrub` (Smith & Kortemme, 2008). Agent Rosetta can use backbone changes to discover nearby conformations that are compatible with much more stabilizing sequences. The agent generates the parameters specific to each Mover, and ResidueSelectors.

- `go_back_to` reverts to a previous step in the trajectory, which Agent Rosetta may do in case the design has diverged too far from the task objectives.

**Multi-turn interaction workflow.** Agent Rosetta refines designs following these steps:

1. **Structured reasoning and action selection:** First, the agent receives the current state of the environment and a summary of the previous actions. Summarization is important to prevent the context of the LLM from saturating over long trajectories (Zhong et al., 2024; Nathani et al., 2025; Shinn et al., 2023; Sumers et al., 2023). We use a tabular summary of all previous actions and their effects on key design metrics so that the LLM's context window contains the last 2 interaction steps only.

   Then, we prompt Agent Rosetta to reflect on the progress and choose the next action. We explicitly instruct the agent to take into consideration both energetic and structural information, the trajectory history, and to write an action plan that *"expert scientists can understand, review, and criticize"*. Structured reasoning is important to bring forth the biophysical knowledge encoded in the LLM, and to allow scientists to verify the coherence of the actions with the objectives of the design brief.

2. **Structured reasoning and parameter generation:** After the agent selects the action, the environment responds with the necessary RosettaScripts documentation. This ensures access to syntax and parameter information, which differs from the majority of code LLMs are trained on, and it varies significantly across types of actions. Instead of the online RosettaScript's documentation, we wrote *ad-hoc* summaries to clarify the aspects the agent struggles the most with.

   We append the documentation to the context of the LLM, and we prompt the agent to reason again and gener-

**Agent Rosetta's reasoning**

```
To improve the design:
Penalize more than 5 prolines.
```

**Generated compositional penalty block**

```
PENALTY_DEFINITION
TYPE PRO
ABSOLUTE 5
DELTA_START -5
- DELTA_END 0 <= WRONG
- PENALTIES 0 0 0 0 -10 -20 <= WRONG
+ DELTA_END 1 <= INTENDED
+ PENALTIES 0 0 0 0 0 10 <= INTENDED
BEFORE_FUNCTION CONSTANT
AFTER_FUNCTION LINEAR
END_PENALTY_DEFINITION
```

*Figure 2.* A failure example of prompting for generation of composition penalties. Even though Agent Rosetta wants to reduce proline content, the penalty block achieves the opposite effect.

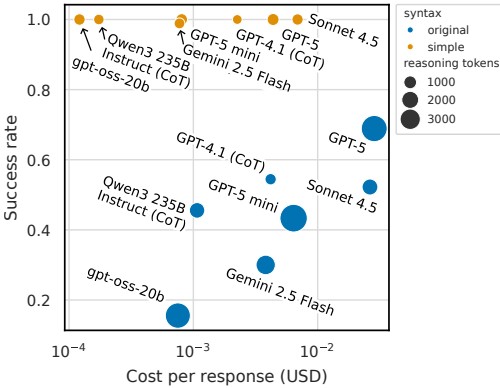

*Figure 3.* Comparison of the performance of different LLMs at generating amino acid compositional penalty blocks with the original RosettaScripts syntax and our simplified syntax.

ate the full action call with its parameters. We use the same structured reasoning steps as for action selection. Repeated reasoning is important at this stage because the original plan of the agent—generated without documentation—may not be implementable.

3. **Action parsing, execution, and feedback:** We fill the XML template with the generated parameters and execute the action. At each step, we only refine the Pareto-optimal candidates from the previous round according to task-dependent quality metrics. If the action did not match the expected response format, it did not satisfy RosettaScripts syntax, or it encountered runtime errors, we prompt the agent with the error message to fix its mistakes. Otherwise, we update the state and the agent proceeds to the next refinement turn.

We include prompts for all steps of the refinement loop in Appendix B.3, and error correction examples in Appendix H.

## 3. Results

Scientists can use Agent Rosetta to perform a broad range of tasks with both canonical and non-canonical residues. We evaluate the agent on stabilizing fixed backbone conformations using canonical amino acids only, and on insertion of 1 non-canonical amino acid (NCAA) in the core of a protein. The former task enables rigorous comparison with existing baselines, and the latter realizes Agent Rosetta's capability for use cases beyond the reach of current ML approaches.

In our experiments, we evaluated Qwen3 {235B, 30B} Instruct (Yang et al., 2025), GPT-{OSS-20B, 4.1, 5} (Agarwal et al., 2025; Achiam et al., 2023; Singh et al., 2025), Gemini 2.5 Flash (Comanici et al., 2025), and Claude Sonnet 4.5 (Anthropic, 2025).[1] We note that we prompt non-reasoning

models (i.e., those LLMs not explicitly trained to produce reasoning traces) to reason step-by-step (Wei et al., 2022).

### 3.1. The Importance of Environment Design

Rosetta and RosettaScripts have extensive online presence, and information about Rosetta is certainly included in the training set of frontier LLMs. This suggests that prompting should suffice to bring forth the necessary knowledge. Instead, we found that LLMs know general facts about Rosetta and RosettaScripts, but they regularly fail to generate semantically correct actions with prompting alone.

For example, consider the composition penalties in the `rotamer_change` action: they follow a particular syntax to select residue types, a target fractional or absolute range, and a sequence of penalties for deviations from the desired range. Furthermore, the number of penalties changes with the width of the target range, and whether the target composition is fractional or absolute. All these factors are both counter-intuitive to first-time RosettaScripts users, and also sources of error for the agent, even with plausible reasoning.

Figure 2 includes an example from one of our experiments. Agent Rosetta correctly recognizes that excessive proline content may be problematic, and wants to penalize more than 5. However, the lines `DELTA_END 0` and `PENALTIES 0 0 0 0 -10 -20` implement the opposite of the agent's intent (they favor more than 4 prolines). We found that extending prompting instructions only yielded minor improvements. Furthermore, this type of error is difficult to recover from in a multi-turn environment as the agent often fails to recognize that it made a mistake.

To solve these issues, we abstracted the syntax of composition penalties and TaskOperations to simplify their use (see Appendix I). Then, we wrote simple Python scripts to con-

---

[1]All models were accessed using OpenRouter. Providers of open-source models were chosen by OpenRouter in real-time depending on latency, throughput, and cost.

vert this simplified syntax into its RosettaScripts equivalent. In Figure 3, we compare the performance of different LLMs at generating composition penalties with the original RosettaScripts syntax versus our simplified one. We compiled a list of 9 prompts with the most commonly-used penalty shapes: above, below, or outside a specified target and range (*e.g.*, *"Write a composition penalty that penalizes more than 5 alanines. Use a linear boundary with slope of 10."*). We averaged results over 10 generations per prompt. For each syntax type, we include the rules in the system prompt, and we verify both the syntactic correctness and the semantics of the responses (see Appendix J for prompts and responses).

We found that both closed- and open-source models fail at reliably generating penalty blocks with the original RosettaScripts syntax, with GPT-5 (medium reasoning effort) achieving the best performance of $\approx 70\%$. We found this value to be insufficient for developing an autonomous agent capable of sustaining multi-turn interactions, especially since generating penalty blocks is an essential component of design in Rosetta. Not only does our simplified syntax reduce costs by orders of magnitude, but also it guarantees robust performance (in this subtask) across all models, irrespective of size and cost ($\geq 98.88\%$ success rate). Building autonomous agents using open-source models, while minimizing monetary and energetic costs, is paramount for democratizing scientific software.

Finally, we performed an ablation study to isolate the effect of our environment compared to prompt-only tool usage. We designed four verifiable tasks of increasing difficulty, from in-context example recall to general Rosetta knowledge. Each task instructs the LLM to generate parts of the RosettaScripts XML scaffold usually automated by our environment. Generalization to unseen RosettaScripts syntax and components is necessary since including all possible instructions or examples in the context is infeasible. All tested LLMs failed sharply when knowledge outside the prompting instructions was required (results are included in Appendix C). We found that without our environment, frontier LLMs cannot generate valid RosettaScripts XML protocols from scratch. We believe this failure generalizes to other scientific software, and not be specific to Rosetta.

These findings stress the crucial role of environment design to bridge general-purpose LLMs with scientific tools.

### 3.2. Stabilizing Backbone Conformations

In order to compare with ML baselines, we evaluated Agent Rosetta on the task of designing sequences to stabilize protein conformations, using canonical amino acids only. The initial state of the trajectory is a poly-glycine backbone in a target conformation, and the task brief instructs the agent to change the side chains so that the molecule is energetically stable and it adopts the same fold as the target (the design

brief is included in Appendix E.1). As target conformations, we chose 8 PDB structures—both natural and synthetic—of length between 74 and 125 residues. Appendix E.2 includes the selection criteria, preprocessing steps, and images of the backbones. We ran the agent for a total of 30 LLM queries, and we configured the environment such that each refinement step generated an ensemble of 128 candidate designs. We selected the Pareto front of the ensembles according to the RMSD between the predicted fold (calculated via ESMFold (Lin et al., 2023)) and the target conformation, the ESMFold pLDDT, and total Rosetta `ref2015` energy (Alford et al., 2017). We chose these proxies for the quality of the designs because their low computational cost allows to compute them after each step of the design process. A low RMSD means that the predicted fold aligns with the target conformation, a high pLDDT conveys good confidence of ESMFold in its prediction, and low Rosetta energy suggests stability in the target conformation. The Pareto candidates were used as initial states for the successive design step.

**Comparison methods.** We compared the designs generated with Agent Rosetta to two ML models, two human written Rosetta protocols, and two scripted policies.

For comparison with ML models, we used ProteinMPNN (Dauparas et al., 2022) and BoltzGen (Stark et al., 2025)—generative models trained on known protein sequences and structures to approximate the conditional distribution of amino acid sequences given an input target conformation.

For comparison with humans, we considered two handwritten protocols composed of the same blocks as the agent's: FastDesign and backbone Movers with ResidueSelectors and TaskOperations. The first is a *one-shot* protocol that applies FastDesign to sample the entire sequence at once. The second is a *staged* protocol that first designs the core, then the boundary, and finally the surface of the molecule—a common practice to reduce the size of the sequence space that FastDesign must explore (see Appendix E.4 for the protocols). We stress that the human written protocols are fixed and independent of the given backbone. In practice, scientists fine-tune them after analyzing rounds of results. However, this process often requires multiple days, making it prohibitive for systematic evaluation across several structures. Therefore, we chose baseline protocols that would reflect a scientist's first few days of work on the same task, using the Rosetta tools available to the agent. We include an experiment with iterative improvement of the one-shot protocol on two backbone conformations in Appendix E.5.

Finally, we compared with two scripted policies: one deterministic, and one randomized. The deterministic policy naïvely runs FastDesign on the Pareto optimal designs without the use of any ResidueSelectors, TaskOperations, or

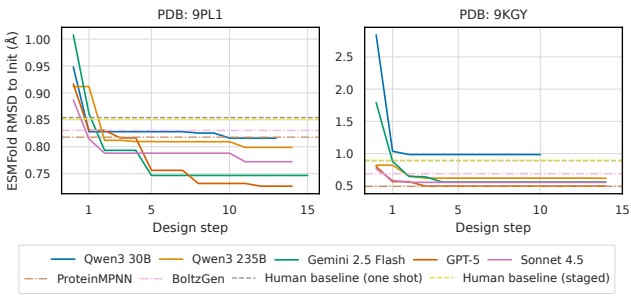

*Figure 4.* Comparison of best running RMSD as a function of design step for 2 backbone conformations. On 9PL1, Agent Rosetta outperforms competing methods, and on 9KGY it helps close the gap between the human written protocols and ProteinMPNN.

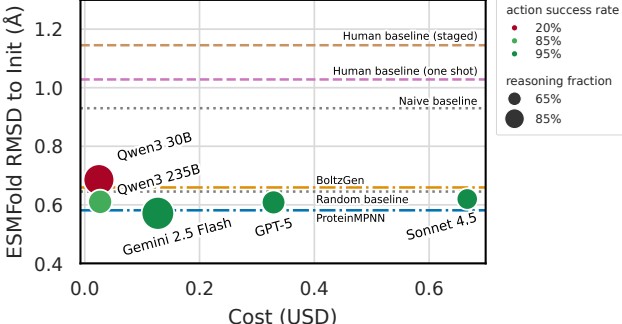

*Figure 5.* Summary of results for stabilizing backbone conformations with canonical amino acids only. We report the average cost of one run of 30 model queries, the average action success rate, and the fraction of output tokens that were reasoning.

composition penalties—simply minimizing Rosetta energy. The randomized policy samples at random between FastDesign and one of the supported backbone Movers with default parameters. These baselines are useful to isolate the role of LLMs making decisions about how to alter protocols as they perform multi-turn scientific tasks.

For a fair comparison, we matched the number of candidate designs across methods (30 LLM queries and 128 designs per step amount to approximately 2048 designs).

**Evaluation.** We ran 16 independent trials per method, and, for each design step, we computed the $5^{th}$ percentile (lower is better) of the ESMFold RMSD and the $95^{th}$ percentile (higher is better) of the pLDDT to summarize the quality of the ensembles.[2] We simulated *best-of-n* scaling over 1,000 bootstrap samples of 8 trials from the 16, and within each sample, we selected the trial with the smallest RMSD.

Figure 4 shows the best running RMSD (*i.e.*, over all previous steps) across bootstrap samples as a function of design step for 2 representative backbones out of the 8 used in this experiment. We note that baselines are shown as horizontal lines because they are single-step methods. On 9PL1 (Masoumzadeh et al., 2025), Agent Rosetta, through iterative refinement, outperforms all comparison methods, whereas on 9KGY (Zhang et al., 2025), the agent helps close the gap between the human written protocols and ProteinMPNN. Figure E.2 includes figures for all backbones.

We summarize results by computing the median RMSD across the 1,000 selected best steps, and average across all backbones, as illustrated in Figure 5. For Agent Rosetta, we show results as a function of average cost of one trajectory with different LLMs. We note that, compared to agents, ProteinMPNN and BoltzGen have negligible computational cost, an advantage of specialized ML models over general-purpose LLMs. Furthermore, we include the RosettaScripts

action success rate and the portion of output tokens that were reasoning. With our tailored environment, all frontier LLMs achieve action success rates $\geq 85\%$. There is a gap in action success rate between frontier LLMs and the smaller Qwen3 30B. We remark that, even though smaller models may be worse out-of-the-box, our environment is a first step towards training specialized scientific agents with reinforcement learning (*e.g.*, GRPO (Shao et al., 2024)).

We found that the agent's designs are on par with ProteinMPNN and BoltzGen (within a 0.20Å tolerance), with Gemini 2.5 Flash and Qwen3 235B Instruct (CoT) achieving the best cost-performance tradeoff. From the detailed results in Tables E.2 and E.3, we found that our agent generates designs with lower RMSD for 4 out of the 8 conformations, but ProteinMPNN and BoltzGen usually achieve higher pLDDTs (see Figure E.3).

We investigated the robustness of these findings in several ways. First, we repeated the experiment with Qwen3 235B Instruct (CoT) omitting the ESMFold quality metrics from the context of the LLM. This ablation evaluates whether the LLM can leverage biophysical knowledge or it overfits to this supervision signal. We found no significant change in performance ($\Delta$RMSD $< 0.1$Å, $\Delta$pLDDT $< 1\%$). Then, we validated designs using AF3. We predicted the fold of the top-10 designs in terms of ESMFold RMSD and pLDDT. We found Agent Rosetta with GPT-5 to be comparable with ProteinMPNN (AF3 RMSD [Å] 0.8 vs. 0.5, pLDDT 91 vs. 94). Finally, we simulated 3 independent MD trajectories with duration of $1\mu s$ starting from the best design generated with GPT-5 and ProteinMPNN. We evaluated stability of the designs with the relative absolute change of the fraction of residues composing helices, strands, and loops. Large variations indicate structural instability. We found no significant difference between Agent Rosetta and ProteinMPNN with a two-sided paired Wilcoxon signed-rank test with family-wise error control rate at level 0.05 (median change in helices: 2% vs. 4%, strands: 4% vs. 3%, and loops: 4%

---

[2] For Agent Rosetta, a trial refers to a full run of the agent. For Rosetta-based baselines, a trial is a single run of the protocol. For ProteinMPNN and BoltzGen, a trial is a single run of the model.

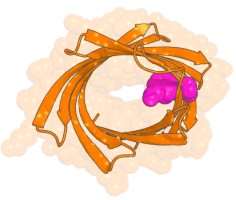 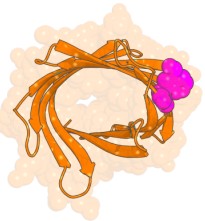

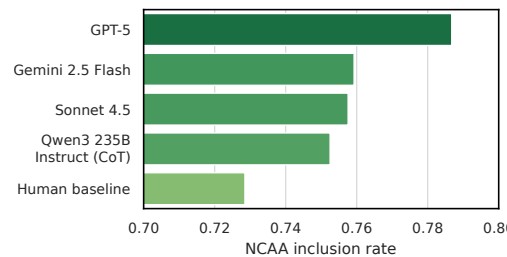

*Figure 6.* Example inclusion of TRF (colored in magenta) in the designed $\beta$-barrel 8UZL. **(left)** Favorable orientation towards the core. **(right)** Unfavorable orientation exposed to the solvent.

*Figure 7.* Median best rate of inclusion of TRF across 1,000 bootstrap samples of 8 out of 16 trajectories, averaged over 4 input PDBs.

vs. 6%). We include MD simulation details in Appendix D and per-PDB trajectories in Figure E.4.

Lastly, Figure E.5 shows the best design for each method on every backbone. Our findings highlight the absence of a single dominating design methodology. Qualitatively, the main failure mode we observed in this task was self-correction loops. In these, the agent would get stuck alternating between wrong action syntax or option names. This could be addressed by increasing the context of the agent beyond the last 2 interactions, or allowing the agent to call the `-info MoverName` flag of Rosetta, which displays a Mover's documentation. We consider this as future work.

### 3.3. Designing an NCAA into the Core of a Protein

Non-canonical amino acids (NCAAs) are important for real-world applications of protein design, but their sparsity limits data-driven approaches. Rosetta's physics-based algorithms, on the other hand, support flexible palettes that can incorporate nonstandard residues. To investigate Agent Rosetta's capabilities beyond the reach of current ML models, we explore the task of inserting an NCAA into the core of an existing protein. The initial state of the trajectory is a PDB structure, and the task brief instructs the agent to include exactly 1 NCAA residue in the core of the protein without compromising its fold and energetic stability (see Appendix F.1 for the task brief). In this experiment, we consider N1-formyl-tryptophan (TRF), a post-translational modification of tryptophan with a formyl group attached to the indole ring. We chose TRF because it is rare in the PDB (it appears in 2 calmodulin complexes only) yet compatible with AF3 as a proxy for structural stability.[3]

We chose 4 *de novo* protein folds of 40 to 153 residues to evaluate the agent across structures of varying rigidity. We report selection criteria, preprocessing steps, and images in Appendix F.3. As an illustrative example, Figure 6 shows a good and bad placement of TRF in 8UZL (Berhanu et al.,

2024), a designed transmembrane $\beta$-barrel among the 4 selected PDBs. In the left panel (good placement), TRF is facing the core and preserves the fold. In the right (bad placement), TRF is exposed to the solvent. This placement is unfavorable because TRF is mostly hydrophobic.

Similar to Section 3.2, we ran Agent Rosetta for a total of 30 LLM queries, and generate step-wise ensembles of 128 candidates. However, we cannot use ESMFold to predict the native fold of the designs because it does not support TRF, and AF3's slow inference time is prohibitive for use at each refinement step. Therefore, we select the Pareto optimal candidates by inclusion of TRF in the core, total Rosetta energy, cavity volume, radius of gyration, and RMSD between the Rosetta Pose and the native PDB.

**Comparison methods.** We compare the performance of our agent with a fixed (*i.e.*, protein-independent) human written protocol composed of the same types of actions available to the agent. We include the human written protocol in Appendix F.5. As in fixed backbone sequence design with canonical amino acids only, this protocol represents a scientist's few days of work.

**Evaluation.** We ran 16 independent design trajectories per method, and emulated *best-of-n* sampling with 1,000 bootstrap replicates of 8 out of the 16. Within each replicate, we selected the step with the highest rate of inclusion of exactly 1 TRF in the core. We computed the median inclusion rate of the 1,000 selected steps, and we summarize results in Figure 7 by averaging across all 4 protein structures. We found that Agent Rosetta with GPT-5 achieves the highest success rate. The designed $\beta$-barrel structure 8UZL proved to be the most challenging: the human baseline failed to include any TRF, and GPT-5 achieved the highest rate of $\approx 14.5\%$. 8UZL does not present an obvious core, and the most common failure mode we observed is that although the agent's action successfully includes one TRF residue in the protein, the inclusion changes the location of the core, which excludes the TRF position. This points precisely to the difficulty of this task: to find positions that allow inclusion of TRF without perturbing conformation.

---

[3]In actual use, Agent Rosetta may be applied to designs with NCAAs incompatible with any ML model, though this would require experimental validation of the design to confirm success. As a proof of principle in this work, we confined ourselves to an NCAA for which AF3 could provide predictions for validation.

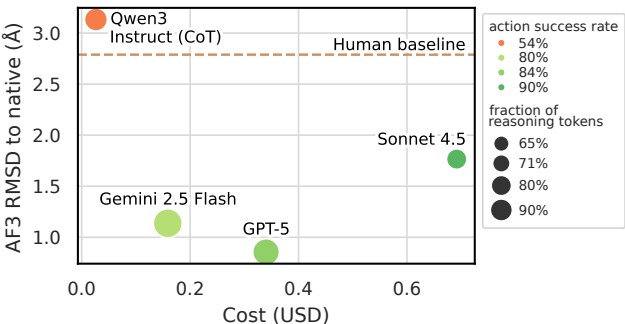

*Figure 8.* Summary of results for including 1 TRF residue in the core of an input protein. We report the average cost of one run with 30 model queries, the average action success rate, and the fraction of output tokens that were reasoning.

*Table 1.* MD simulation results for the NCAA task. We report median (95% CI) across PDB structures, designs, and replicates.

| Method | RMSD (Å) | Relative absolute change (%) | | |
|---|---|---|---|---|
| | | Helix | Strand | Loop |
| Agent Rosetta (GPT-5) | 2.6 (2.3/4.5) | 2 (0/10) | 3 (0/20) | 13 (1/60) |
| Human baseline | 3.1 (2.5/7.1) | 3 (0/37) | 8 (0/69) | 22 (3/137) |

For validation with AF3, we filtered successful designs, selected the top 10 in order of increasing RMSD to the native structure, and predicted their fold with AF3. Figure 8 summarizes the RMSD between the AF3 prediction and the native structure, and we include detailed tabular results in Tables F.4 and F.5. We found that Agent Rosetta outperforms the human baseline in terms of both AF3 RMSD and pLDDT (see Figure F.3), with GPT-5 performing best overall.

Across base LLMs, Qwen3 235B Instruct (CoT) showed the largest performance drop compared to designing with canonical amino acids. Inspection of the design trajectories revealed that errors concentrated on `backbone_change` actions. This is because incorporating an NCAA often requires targeted backbone perturbations to accommodate the residue without compromising the fold. Agent Rosetta can combine backbone Movers with ResidueSelectors to achieve this, but the selected residues need to define valid *segments* for perturbation. In many cases, the ResidueSelectors specified by Agent Rosetta with this model did not meet this criterion, causing the action to fail at runtime.

As an independent proxy to experimental validation, we performed MD simulations to compare the stability of Agent Rosetta's designs with the human written protocol's. Given the high computational cost of MD simulations, we validated GPT-5's designs only. We selected the top-5 candidates in terms of RMSD to the native structure, and we simulated 3 replicates of $1\mu s$ each (we include details in Appendix D). We evaluated designs at the end of the simulation in terms of RMSD to the native structure and relative abso-

lute change of the fraction of helices, strands, and loops in the secondary structure. A large RMSD suggests unfolding due to bad TRF placement in the code, and large variations in secondary structure indicate instability. Table 1 reports median and 95% CI across PDB structures, designs, and replicates (we exclude 8UZL from the comparison since the human baseline fails on this structure). We found that Agent Rosetta with GPT-5 is statistically better than the human written protocol using paired Wilcoxon signed-rank tests with family-wise error rate control at level 0.05. We show changes in RMSD and secondary structure as a function of simulation time in Figures F.5 and F.6.

Finally, in Figure F.4, we include some example good and bad designs generated by Agent Rosetta and the human written protocol.

## 4. Conclusion

We have introduced Agent Rosetta, a framework that leverages the reasoning capabilities of Large Language Models to automate protein design within the Rosetta Macromolecular Modeling Suite. A central finding of our study is that prompt engineering alone is insufficient for interfacing with complex, domain-specific scripting languages; instead, success relies on a structured environment that abstracts syntactic intricacies into semantic actions. By implementing this design, we enabled the agent to iteratively refine protocols and recover from errors that typically stall autonomous workflows while minimizing computational and economic cost.

Our evaluations confirm that this approach effectively bridges the gap between generalist reasoning models and specialized scientific software. In fixed-backbone sequence design with canonical amino acids, Agent Rosetta achieves performance parity with state-of-the-art models like ProteinMPNN and BoltzGen. Moreover, in the data-sparse regime of non-canonical amino acid design, the agent outperforms expert human baselines, tackling tasks where purely data-driven methods are currently inapplicable.

Our work is limited to RosettaScripts and a few representative PDB structures. Generalization of our findings to other scientific fields, different software (*e.g.*, PyRosetta (Chaudhury et al., 2010)), and diverse protein backbones will be central for future work. Based on the empirical evidence presented in this work, we expect our results to hold when LLMs have seen related scientific literature and the scientific software has a structured interface.

Our results suggest that agentic frameworks, when coupled with robust environment design, offer a powerful paradigm for unlocking the full potential of physics-based simulations in biology discovery.

## Impact Statement

This paper presents work whose goal is to advance the fields of machine learning and biology. There are many potential societal consequences of our work. We highlight potentially harmful consequences for dual use. First, our agent could be prompted to design harmful molecules (e.g., viruses, toxins). Second, our agent could lead to confidently wrong designs that may have unwanted consequences for non-expert users. Safety guardrails such as flagging unusual requests or designs for expert review are important for the broader use of our work.

## Acknowledgments

We would like to acknowledge the support of the Simons Foundation and of Schmidt Sciences. This work was supported in part by the AI2050 program at Schmidt Sciences (Grant G-25-70028). PDR and VKM are wholly funded by the Simons Foundation. We thank Lucy Reading-Ikkanda and Aditya Chhatrala for their contribution to the figures presented in this paper. We thank the Flatiron Institute's Scientific Computing Core for ongoing support. The computations reported in this paper were performed in-part using resources made available by the Flatiron Institute. The Flatiron Institute is a division of the Simons Foundation. We thank the anonymous NeurIPS reviewers for their suggestions, which strengthened our empirical results.

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

# A. Related Works

In this section, we provide a detailed overview of relevant related works.

**LLM agents.** Through text generation, modern LLMs are capable of using external tools such as web browsers (Zhou et al., 2023; Abuelsaad et al., 2024; Wei et al., 2025; He et al., 2024; Ning et al., 2025), email (Ruan et al., 2023; Debenedetti et al., 2024), and, more broadly, code-based interfaces (Jimenez et al., 2023; Hou et al., 2024; Yang et al., 2024). Coupled with reasoning and *chain-of-thought* prompting (Wei et al., 2022; Shao et al., 2024), LLMs can act as efficient *agents*, automating repetitive workflows and solving complex tasks that require multi-turn interactions (Wang et al., 2025b).

**Scientific agents.** Scientific discovery workflows are generally well-structured, which makes them attractive for the development of *AI scientists* (Gottweis et al., 2025; Lu et al., 2024; Team et al., 2025). Recent works have explored LLMs agents for idea and hypothesis generation (Baek et al., 2024; Li et al., 2024; Pu et al., 2025; Radensky et al., 2024; Guo et al., 2025), experiment design (Boiko et al., 2023; Ghafarollahi & Buehler, 2024), and paper writing (Yamada et al., 2025; Chen et al., 2025; Ghafarollahi & Buehler, 2025). These approaches have found several applications in math (Trinh et al., 2024; Luong & Lockhart, 2025), chemistry (Bran et al., 2023; Boiko et al., 2023; Narayanan et al., 2025), and therapeutics (Wang et al., 2025a). Most closely related to our work, ProtAgents (Ghafarollahi & Buehler, 2024) is a multi-agent system that orchestrates the use of existing machine learning tools for *de novo* protein discovery; CRISPR-GPT (Qu et al., 2025) studies LLM agents for gene editing tasks; and Biomni (Huang et al., 2025) develops an agentic framework for general biomedical tasks. We refer interested readers to (Gridach et al., 2025; Zhang et al., 2024) for comprehensive reviews.

**Manually-crafted Rosetta protocols for heteropolymer design.** Rosetta has been extensively used for rational design of new protein folds (Kuhlman et al., 2003; Koga et al., 2012; Rocklin et al., 2017), rigidly-structured peptides (Bhardwaj et al., 2016; Hosseinzadeh et al., 2017; Dang et al., 2017; Mulligan, 2022b), and other heteropolymers (Renfrew et al., 2012; Drew et al., 2013; Renfrew et al., 2014). Proteins have been rationally designed with Rosetta that self-assemble into sheets (Gonen et al., 2015) or cages (King et al., 2014; Hsia et al., 2016), that bind metals (Mills et al., 2013) or small molecules (Tinberg et al., 2013), that catalyze new enzymatic reactions (Jiang et al., 2008; Siegel et al., 2010), and that specifically recognize target proteins (Strauch et al., 2017). Non-canonical peptides have also been designed which bind target proteins (Mulligan et al., 2021; Hosseinzadeh et al., 2021) or passively diffuse across cell membranes (Bhardwaj et al., 2022). Rosetta's infrastructure, energy function, and development are described in (Leaver-Fay et al., 2011; Alford et al., 2017; Leman et al., 2020a;b). Rosetta's scripting interfaces are described in (Chaudhury et al., 2010; Fleishman et al., 2011). Non-canonical Rosetta design methods are reviewed in (Mulligan, 2020; 2022a; Mulligan & Hosseinzadeh, 2022; Dodd-O et al., 2023). Recently, the Masala libraries were introduced to permit easy, plugin-based development of new optimizers and other algorithms to extend Rosetta or other heteropolymer modeling software (Zaborniak et al., 2025).

**ML for protein structure prediction and design.** Deep learning methodologies have proved successful at training models for particular design pipelines. For example, models exist for both monomer (AlphaFold 2 (Jumper et al., 2021), RoseTTAFold (Baek et al., 2021; Humphreys et al., 2021), ESMFold (Lin et al., 2023)) and complex structure prediction (AlphaFold 3 (Abramson et al., 2024) and Multimer (Evans et al., 2021), OpenFold (Ahdritz et al., 2024) RosettaFold All-Atom (Krishna et al., 2024), the Chai (Chai Discovery, 2024; 2025) and Boltz (Wohlwend et al., 2024; Passaro et al., 2025) series of models); sequence generation (ProteinMPNN (Dauparas et al., 2022), the ProGen (Nijkamp et al., 2023; Bhatnagar et al., 2025) series, ProtGPT2 (Ferruz et al., 2022)); and structure generation and docking (RFdiffusion (Watson et al., 2023), DiffDock (Corso et al., 2023), and the Genie (Lin & AlQuraishi, 2023; Lin et al., 2024) series). We point to (Lee et al., 2025) for a recent review on language models for protein design, and (Mulligan, 2021) for a review of earlier ML approaches used for protein modeling.

# B. Prompts

In this section, we include all prompt templates. In the prompts, fields enclosed within curly brackets (*e.g.*, {field_name}) are populated at runtime.

## B.1. System Prompt

Here, we include Agent Rosetta's system prompt. {docs} is populated with Python-style docstrings that summarize the available actions and their parameters, and {reasoning_formatting} varies depending on the base LLM: for

reasoning models this field is omitted, whereas for models that require chain of thought (CoT) prompting (Wei et al., 2022) we include instructions to first reason step-by-step with special delimiters like `<think></think>` for OpenAI models, or `<reasoning></reasoning>` for Qwen models.

---

**Box 1: System prompt**

```
You are an expert RosettaScripts coding agent that supports scientists in biomolecular
    design tasks:

- Rosetta is a computational toolkit for modeling, predicting, and designing
    biomolecular structures and interactions, using physics-based energy functions and
    stochastic search.
- RosettaScripts is Rosetta's XML-based interface for assembling custom modeling
    protocols by combining movers, filters, and scoring terms.

Your goal is to follow a user-defined brief and to generate a valid RosettaScripts
    protocol that captures the brief precisely.

You have interactive access to a RosettaScripts environment with the following actions:

{docs}

**Instructions:**

You will follow a two-step interaction schema with the RosettaScripts environment.

At the beginning of each interaction round, you will be presented a summary of the
    current state of the environment. You must follow these instructions carefully in
    order to choose and execute the next best action that will achieve the design
    goals:
1. STEP 1: Reason about the biophysical implications of the design brief for the
    current state of the environment and CHOOSE the next action to take.
2. STEP 2: Only the environment has displayed the full documentation of the action you
    chose, WRITE the full action call with all its arguments.

You must follow the formatting instructions below.

**Formatting:**

{reasoning_formatting}

Write your action call inside <action></action> tags, following this exact format:

<action tag="choose" or "run">
<name>action_name</name>
<arg_name1>arg_value1</arg_name1>
<arg_name2>arg_value2</arg_name2>
...
</action>

Rules:
- Use <action tag="choose"></action> to CHOOSE the action.
- Use <action tag="run"></action> to RUN the action.
- Use <action_name>action_name</action_name> to specify the name of the action.
- For each argument of the action, use XML tags with the name and value of the
    argument.
- Write one (1) action call per response only. The environment will only parse the
    first <action></action> block.
- Do not include any reasoning or comments in your action call. Write valid action
    content only.
- Do not code fence your action call.

**Example:**
```

```
- FIRST, write:

<action tag="choose">
<name>action_name</name>
</action>

- THEN, only after the environment has displayed the full action documentation, write:

<action tag="run">
<name>action_name</name>
<arg_name1>arg_value1</arg_name1>
<arg_name2>arg_value2</arg_name2>
...
</action>

Now, read the design brief and start designing your RosettaScripts protocol.
```

The docstrings for our three types of actions are:

**Box 2: Action docstrings**

```
- rotamer_change
"""Perform a Rosetta FastDesign action with design enabled to change the rotamers of
    the sequence without perturbing the backbone. This action combines residue
    selectors, compositional penalties, residue restrictions, and packing restrictions
    to guide the search process.

Arguments:
  - residue_selectors (string, optional): the XML definitions of the residue selectors.
  - penalties (array, optional): a list of compositional penalty definitions. Each
    item includes the following parameters:
    - comp (string, optional): the penalty definition blocks specifying the
    compositional constraints.
    - comp_selector_name (string, optional): the name of the residue selector to apply
    the compositional constraints to.
  - residue_restrictions (array, optional): a list of residue restriction definitions.
    Each item includes the following parameters:
    - type (string, optional): whether to `restrict` or `prohibit` residue types.
    - residues (string, optional): the list of one- or three-letter residue codes
    separated by a comma.
    - selector_name (string, optional): the name of the residue selector to apply the
    restriction to.
  - packing_restrictions (string, optional): A list of residue selector names
    separated by commas.
"""

- backbone_change
"""Perturb the backbone conformation. This action implements three Rosetta backbone
    movers: `small`, `shear`, and `backrub`. Specify the mover to use with
    `mover_name`, the parameters of the mover in XML format with `mover_params`, and
    the residues to perturb with `residue_selector`.

Arguments:
  - mover_name (string, required): the name of the Rosetta backbone mover to use (can
    be `small`, `shear`, or `backrub`).
  - mover_params (string, optional): the parameters of the mover in XML format.
  - residue_selectors (string, optional): the XML definitions of the residue selectors.
  - mover_selector_name (string, optional): the name of the residue selector to apply
    the mover to.
"""

- go_back_to_step
"""Revert the environment state to a previous step in the trajectory. This action
    resets the environment to the state at the end of the specified step, allowing you
```

```
    to retry actions or explore different paths.

Arguments:
  - step (integer, optional): the number of the step to revert to, starting from 0.
"""
```

## B.2. Design Brief

Here, we include Agent Rosetta's design brief prompt. {prompt} is populated with the user-specified task description and objectives, {state} contains the textual representation of the initial environment state, and {action_formatting_instructions} varies by model. Similarly to the {reasoning_formatting} parameter in the system prompt, {action_formatting_instructions} instructs reasoning models to directly generate the final response, without including any reasoning, and non-reasoning models that require CoT prompting to first summarize step-by-step reasoning within special delimiters. This ensures the same response content and format across different LLMs.

---

**Box 3: Design brief**

```
**Design Brief:**

{prompt}

**Starting State:**

{state}

Now choose the next best action following the formatting instructions in the system
    prompt:
{action_formatting_instructions}
```

---

## B.3. Multi-turn Interaction Workflow

Here, we include the prompts for each stage of the multi-turn interaction workflow, starting from action selection. {header}, {step}, and {act_name} are populated at runtime depending on the exit status and environment state, {state} contains the textual representation of the environment state, {summary} reports the tabular summary of the action history, and {action_formatting_instructions} varies depending on model as described in Appendix B.2 for the design brief.

---

**Box 4: Structured artificial reasoning and action selection**

```
{header}

**Step Number:** {step}
**Action Name:** {act_name}
**Results:**

{state}

---

**History Summary:**

{summary}

---

**Revision Instructions:**

First review the last action call, then judge the results of the last action call, and
    finally choose the next best action to achieve the design goals.

Make sure to take into consideration all information provided, including the history
    summary. Integrate this information with your expert biophysics knowledge. Some
    example questions to guide your judgment are:
```

---

```
- Did the last action worsen any important metrics compared to the previous step?
- Were there any penalty definition blocks that did not affect the results at all? For
    example, because their residue selectors were syntactically valid but semantically
    empty?
- Were there any penalty definition blocks that had unintended consequences? For
    example, because their residue selectors were misspecified or because their
    penalties introduced unfavorable residues?
- Are there any important pieces of information missing from the results that would
    help you craft the next action better?
- Can you infer those missing pieces of information based on your expert knowledge of
    RosettaScripts and biomolecular design?

Now choose the next best action following the formatting instructions in the system
    prompt:
{action_formatting_instructions}
```

After choosing the next action, the agent is prompted again to generate the full action call with all its parameters with the necessary RosettaScripts documentation. {act_name} contains the name of the chosen action and {docs} its documentation.

---

**Box 5: Structured reasoning and parameter generation**

```
You chose to perform action '{act_name}'. This is the full documentation of the action:

{docs}

---

**Action Instructions:**

First carefully read the documentation of the action you chose, then reason about a
    practical implementation that achieves your intended effects, and finally write
    the full action call with all its arguments.

Remember that the documentation is not exhaustive, and you need to integrate it with
    your expert knowledge of RosettaScripts.
Be specific in your reasoning: the expert biomolecular scientists on your team should
    be able to understand, review, and critique your plan.

Now write the full action call with all its arguments following the formatting
    instructions in the system prompt:
{action_formatting_instructions}
```

In case the generated action incurs in RosettaScripts syntax validation or runtime errors, the agent is prompted to correct its mistakes. {error} is populated with the error message from the RosettaScripts environment.

---

**Box 6: Error correction**

```
The RosettaScripts environment failed to run your action with the following error:

{error}

**Instructions:**

First carefully read the error message, then reason about the possible causes of the
    error, and finally write the corrected full action call with all its arguments.

Your corrected action call must preserve the intent of your previous action. Some
    common sources of errors are:
    - Syntactic mistakes in the action call: wrong parameter names, wrong XML
    structure.
    - Semantic mistakes in the action call: wrong parameter values, logically invalid
    combinations of parameters.
Be specific in your reasoning: the expert biomolecular scientists on your team should
```

```
    be able to understand, review, and critique your solution.

Now write the corrected full action call with all its arguments following the
    formatting instructions in the system prompt:
{action_formatting_instructions}
```

## C. Environment Ablation

*Table C.1.* Results of our environment ablation study where LLMs are instructed to generate parts of RosettaScripts XML scaffold usually automated by our environment. We average success rates over 10 independent query for each task.

| Model | Success rate (%) | | | |
| --- | --- | --- | --- | --- |
| | Task 1 (easy) | Task 2 (medium) | Task 3 (medium) | Task 4 (hard) |
| GPT-5 | 100 | 100 | 40 | 0 |
| Gemini 2.5 Flash | 90 | 10 | 0 | 0 |
| Qwen3 235B Instruct (CoT) | 90 | 20 | 0 | 0 |
| Sonnet 4.5 | 90 | 100 | 90 | 0 |

In this section, we include an ablation study to isolate the effect of our environment compared to prompt-only tool usage. We designed four verifiable tasks of increasing difficulty, from in-context example recall to general Rosetta knowledge. Including instructions or examples for all possible RosettaScripts pipelines is infeasible, so generalization to unseen syntax is necessary. Each task instructs the LLM to generate parts of RosettaScripts XML scaffold that is usually automated by our environment. The system prompt describes the XML schema, section contents, and examples. The tasks are:

1. **Level: easy.** Define a custom score function with modified weights. We provide explicit examples in the context.

2. **Level: medium.** Store the amino acid sequence of the RosettaScripts Pose. We explicitly provide the name of the required *SimpleMetric* in the context, but we do not include examples.

3. **Level: medium.** Store the per-residue `fa_rep` and `rama_prepro` energy terms. We do not provide the names of the required SimpleMetrics in the context, but we do mention similar components. The LLM should understand how to generalize the pattern in the names of the SimpleMetrics to obtain the required one.

4. **Level: hard.** Estimate and store the cavity volume and radius of gyration of the Pose. This task is difficult because it requires ad-hoc RosettaScripts knowledge. In particular, cavity volume requires a Filter, whereas radius of gyration is an energy term. This require memorization of RosettaScripts documentation, which is infeasible to do for all possible use cases.

For each task, we execute the LLM-generated RosettaScripts protocols and verify they match the expected results. We evaluate each LLM by averaging success rate over 10 independent queries for each task. Table C.1 includes results for 4 frontier LLMs. We found that all LLMs fail sharply when the task requires ad-hoc Rosetta knowledge that cannot be easily included in the context. These results highlight the crucial role of our structured environment in enabling sustained interaction with Rosetta. Without it, frontier LLMs cannot generate valid RosettaScripts XML protocols from scratch.

## D. Molecular Dynamics Simulations

In this section, we provide details on the molecular dynamics (MD) simulation validation pipeline. MD simulations were performed using OpenMM 8.2 (Eastman et al., 2023) with CHARMM-format topology and coordinate files (Brooks et al., 2009). Initial system preparation and parameterization were carried out using VMD (Visual Molecular Dynamics, version 1.9.3 (Humphrey et al., 1996)) with the `psfgen` plugin. Protein topology and coordinates were generated from the final designed structures using the CHARMM36 Force Field (Best et al., 2012), with additional modifications to parameterize the non-canonical residue TRF. Systems were then solvated using the VMD `solvate` plugin with a padding distance of 12Å. Counter-ions were added using the `autoionize` plugin to neutralize the systems and achieve a physiological salt concentration of 0.15M.

Within the MD simulations, long-range electrostatic interactions were computed using the particle mesh Ewald (PME) method with a real-space cutoff of 12Å. Bonds involving hydrogen atoms were constrained, and simulations were propagated using a Langevin integrator with a friction coefficient of $1\text{ps}^{-1}$. Production simulations used an integration time step of 2fs.

Each system was first subjected to energy minimization to remove steric clashes and relax the starting structure. The minimized systems were then gradually heated using a staged temperature-ramping protocol to improve equilibration stability. Heating was performed using a reduced integration time step of 0.5fs. Systems first underwent a short relaxation at 50K for 2000 integration steps, after which the temperature was increased from 50K to 300K in increments of 50K, with 2000 MD steps performed at each stage.

Following temperature ramping, the systems were equilibrated in the canonical (NVT) ensemble at 300K using a Langevin thermostat. The equilibrated systems were then transitioned to the isothermal-isobaric (NPT) ensemble at 300K and 1bar using a Monte Carlo barostat, allowing the periodic box dimensions and system density to relax. After equilibration, production MD simulations were carried out in the NPT ensemble for $1\mu s$ per trajectory. To improve sampling, three independent replicate simulations were performed for each system using different initial velocity assignments.

## E. Fixed Backbone Sequence Design

In this section, we include further experimental details on stabilizing an initial backbone conformation with canonical amino acids only.

### E.1. Design Brief

Here, we include the specific design brief used in this experiment. This design brief is composed at runtime with the general design brief prompt template presented in Appendix B.2.

---

**Box 7: Design brief**

```
You are given an initial backbone conformation composed of glycine residues only.

Your task is to design (i.e., change) the residues of the sequence such that:
1. It is energetically stable according to biophysical principles.
2. It has low Rosetta energy.
3. The sequence folds into the same shape as the initial backbone structure.

You will be provided the following proxy metrics to help you assess task progress:
1. Total Rosetta energy.
2. The RMSD between the ESMFold predicted fold and the initial backbone conformation.
3. The uncertainty (i.e., average pLDDT) of the ESMFold prediction.

At each step:
- The Rosetta environment will execute your action and produce an ensemble of
    candidate designs.
- You will be shown summary statistics of the ensemble.
- You will choose the next action to take, which will be applied to Pareto optimal
    candidates only in terms of the provided proxy metrics.
```

---

### E.2. Initial Backbone Conformations

In this section, we describe the inclusion criteria and preprocessing steps used to select 8 backbone conformations to use in this experiment (see Figure E.1).

**Selection of PDBs for design.** Protein backbones for canonical amino acid design were selected from the PDB to ensure structural diversity and designability. Structures were selected based on release date cutoff of after May 15, 2025, single polymer chain, and chain length of less than 150 residues. The release-date cutoff was chosen to reduce the likelihood that the exact structures were present in the ProteinMPNN training or test sets, thereby reflecting a realistic design campaign setting. However, we could not exclude that ProteinMPNN was trained on proteins with similar structure. The selected PDBs were: 9PL1 (Masoumzadeh et al., 2025), 9VYW (Gaikwad et al., 2025), 9VYQ (Gaikwad et al., 2025), 9C14 (de Oliveira Silva et al., 2025), 8YS1 (Baba et al., 2025), 8VWO (Hassan et al., 2025), 9IFR (Costa et al., 2025), and 9KGY

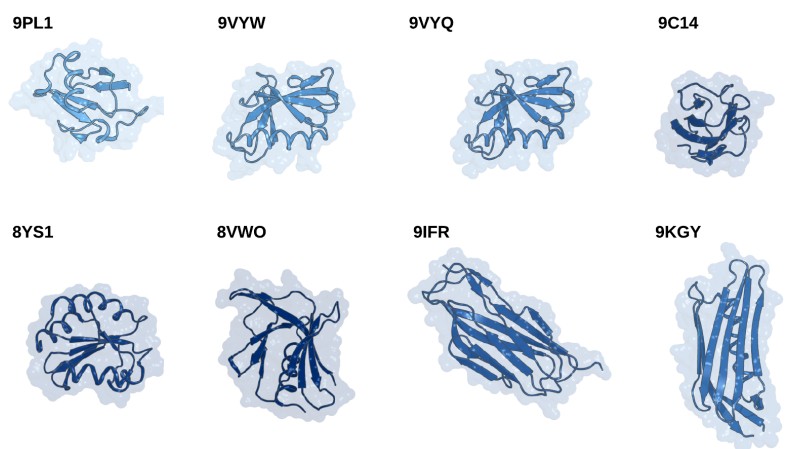

*Figure E.1.* The 8 backbone conformations to stabilize with canonical amino acids only.

(Zhang et al., 2025).

**Preprocessing of PDBs.** All structures chosen as initial scaffolds for design with canonical amino acids were processed in PyMOL (Schrödinger, LLC, 2025b;a) to remove alternative conformations (`altlocs B-G`) and all heteroatoms and X-ray crystallographic copies. All native residues were replaced with glycine without changes to the positions of the C$\alpha$ atoms of the backbone conformations.

### E.3. Environment State

Here, we include an illustrative example of the textual representation of the RosettaScripts environment state for the fixed-backbone sequence design task. We omit the tabular history summary for the sake of readability.

---

**Box 8: Environment State**

```
The RosettaScripts environment successfully executed your action:

**Step Number:** 4
**Action Name:** rotamer_change
**Results:**

- Number of designs: 114 (6 Pareto optimal)

- Average total Rosetta energy: -385.47 ± 12.39

- Average compositional before design 607.89, and after design 333.33 (-274.56)

- Top 5 most common outlier residue types. A residue is an outlier if its energy term
    is above the 90-th quantile of the per-residue energy term for that design. For
    each outlier residue type, we include the most common outlier positions along the
    sequence (positions are 1-based):

-- interresidue_repulsion:

Average per-sequence 90-th quantile: 2.31

PHE: 99.12%  (positions: 87,65,25)
VAL: 98.25%  (positions: 17,122,4,91,89,25,99)
LEU: 96.49%  (positions: 103,50,85)
CYS: 96.49%  (positions: 95,99,55)
ILE: 91.23%  (positions: 91,42,72,25,101,31,43,23)

-- ramachandran_preference:
```

---

```
Average per-sequence 90-th quantile: 0.56

THR: 100.00% (positions: 33,97)
ASP: 98.25%  (positions: 67)
PRO: 97.37%  (positions: 76,48)
TYR: 95.61%  (positions: 98)
CYS: 94.74%  (positions: 116,95)

- Structural metrics:

Cavity volume (Å^3)                                :   12.24 ± 11.58
Radius of gyration (Å)                             :   15.22 ± 0.01
Penalty for buried unsatisfied Hydrogen bonds      : 5883.11 ± 1362.39
RMSD of ESMFold prediction to initial structure (Å):   1.84 ± 1.50
CA pLDDT of ESMFold prediction                     :    0.79 ± 0.07

---

**History Summary:**

{history_summary}
```

## E.4. Expert Human Baseline Protocols

In this section, we include the expert human design protocols used as baselines in the fixed-backbone sequence design task.

### E.4.1. ONE-SHOT DESIGN PROTOCOL

This protocol samples the entire sequence at once with the use of ResidueSelectors, TaskOperations, and compositional penalty blocks

---

**Box 9: One-shot design protocol**

```
<ROSETTASCRIPTS>
<SCOREFXNS>
    <ScoreFunction name="r15_regular" weights="ref2015_cst.wts" />
    <ScoreFunction name="r15_design" weights="ref2015.wts">
        <Reweight scoretype="aa_composition" weight="1.0" />
        <Reweight scoretype="atom_pair_constraint" weight="1.0" />
    </ScoreFunction>
</SCOREFXNS>

<RESIDUE_SELECTORS>
    <Layer name="core" select_core="true" select_boundary="false"
    select_surface="false" use_sidechain_neighbors="true" />
    <Layer name="boundary" select_core="false" select_boundary="true"
    select_surface="false" use_sidechain_neighbors="true" />
    <Layer name="surface" select_core="false" select_boundary="false"
    select_surface="true" use_sidechain_neighbors="true" />
</RESIDUE_SELECTORS>

<RESIDUE_LEVEL_TASK_OPERATIONS>
    <RestrictToRepackingRLT name="RestrictToRepacking" />
</RESIDUE_LEVEL_TASK_OPERATIONS>

<TASKOPERATIONS>
    <IncludeCurrent name="include_current_rotamer"/>
    <ExtraRotamersGeneric name="extra_sample_rotamers_design" ex1="1" ex1aro="1" />
    <ExtraRotamersGeneric name="extra_sample_rotamers_relax" ex1="1" ex2="1"
    ex1aro="1" ex2aro="1" />

    <ProhibitSpecifiedBaseResidueTypes name="prohibit_aa_in_core"
    base_types="ARG,LYS,ASP,GLU,CYS,PRO" selector="core" />
```

```
        <ProhibitSpecifiedBaseResidueTypes name="prohibit_aa_in_boundary"
    base_types="ARG,LYS,ASP,GLU,CYS,PRO,MET,HIS" selector="boundary" />
        <ProhibitSpecifiedBaseResidueTypes name="prohibit_aa_in_surface"
    base_types="ILE,LEU,VAL,PHE,TRP,MET" selector="surface"/>
</TASKOPERATIONS>

<SIMPLE_METRICS>
        <SequenceMetric name="record_sequence" output_mode="basename" />
</SIMPLE_METRICS>

<MOVERS>
        <Small name="small_move" scorefxn="r15_regular" temperature="0.5" nmoves="1000"
    angle_max="2.0" preserve_detailed_balance="0" />

        <FastDesign name="design_all" scorefxn="r15_design" disable_design="false"
    task_operations="include_current_rotamer,extra_sample_rotamers_design,
    prohibit_aa_in_core,prohibit_aa_in_boundary,prohibit_aa_in_surface" repeats="4"
    relaxscript="default" min_type="lbfgs_armijo_nonmonotone" />

        <AddCompositionConstraintMover name="comp_core" filename="{comp_core}"
    selector="core"/>
        <AddCompositionConstraintMover name="comp_boundary" filename="{comp_boundary}"
    selector="boundary"/>
        <AddCompositionConstraintMover name="comp_surface" filename="{comp_surface}"
    selector="surface"/>

        <AddConstraints name="geom_constraint">
            <AtomPairConstraintGenerator name="gen_geom_csts" ca_only="1" use_harmonic="1"
    native="1" />
        </AddConstraints>
        <RemoveConstraints name="rm_geom_csts" constraint_generators="gen_geom_csts" />

        <FastRelax name="fast_relax" scorefxn="r15_regular" disable_design="true"
    task_operations="include_current_rotamer,extra_sample_rotamers_relax" repeats="3"
    relaxscript="default" min_type="lbfgs_armijo_nonmonotone" />
</MOVERS>

<PROTOCOLS>
        <Add mover="geom_constraint" />
        <Add mover="small_move" />
        <Add mover="comp_core"/>
        <Add mover="comp_boundary"/>
        <Add mover="comp_surface"/>
        <Add mover="design_all" />
        <Add mover="rm_geom_csts" />
        <Add mover="fast_relax" />
        <Add metrics="record_sequence" />
</PROTOCOLS>

<OUTPUT scorefxn="r15_design" />
</ROSETTASCRIPTS>
```

### E.4.2. STAGED DESIGN PROTOCOL

This protocol first samples the core, then the boundary, and finally the surface of the molecule. It also uses ResidueSelectors, TaskOperations, and compositional penalty blocks

---

**Box 10: Staged design protocol**

```
<ROSETTASCRIPTS>
<SCOREFXNS>
        <ScoreFunction name="r15_regular" weights="ref2015_cst.wts" />
        <ScoreFunction name="r15_design" weights="ref2015.wts">
            <Reweight scoretype="aa_composition" weight="1.0" />
```

```
            <Reweight scoretype="atom_pair_constraint" weight="1.0" />
        </ScoreFunction>
</SCOREFXNS>

<RESIDUE_SELECTORS>
    <Layer name="core" select_core="true" select_boundary="false"
    select_surface="false" use_sidechain_neighbors="true" />
    <Layer name="boundary" select_core="false" select_boundary="true"
    select_surface="false" use_sidechain_neighbors="true" />
    <Layer name="surface" select_core="false" select_boundary="false"
    select_surface="true" use_sidechain_neighbors="true" />
</RESIDUE_SELECTORS>

<RESIDUE_LEVEL_TASK_OPERATIONS>
    <RestrictToRepackingRLT name="RestrictToRepacking" />
</RESIDUE_LEVEL_TASK_OPERATIONS>

<TASKOPERATIONS>
    <IncludeCurrent name="include_current_rotamer" />
    <ExtraRotamersGeneric name="extra_sample_rotamers_design" ex1="1" ex1aro="1" />
    <ExtraRotamersGeneric name="extra_sample_rotamers_relax" ex1="1" ex2="1"
    ex1aro="1" ex2aro="1" />

    <DesignRestrictions name="not_core">
        <Action selector_logic="NOT core"
    residue_level_operations="RestrictToRepacking" />
    </DesignRestrictions>
    <DesignRestrictions name="not_surface">
        <Action selector_logic="NOT surface"
    residue_level_operations="RestrictToRepacking" />
    </DesignRestrictions>
    <DesignRestrictions name="not_boundary">
        <Action selector_logic="NOT boundary"
    residue_level_operations="RestrictToRepacking" />
    </DesignRestrictions>

    <ProhibitSpecifiedBaseResidueTypes name="prohibit_aa_in_core"
    base_types="ARG,LYS,ASP,GLU,CYS,PRO" selector="core" />
    <ProhibitSpecifiedBaseResidueTypes name="prohibit_aa_in_boundary"
    base_types="ARG,LYS,ASP,GLU,CYS,PRO,MET,HIS" selector="boundary" />
    <ProhibitSpecifiedBaseResidueTypes name="prohibit_aa_in_surface"
    base_types="ILE,LEU,VAL,PHE,TRP,MET" selector="surface" />
</TASKOPERATIONS>

<SIMPLE_METRICS>
    <SequenceMetric name="record_sequence" output_mode="basename" />
</SIMPLE_METRICS>

<MOVERS>
    <Small name="small_move" scorefxn="r15_regular" temperature="0.5" nmoves="1000"
    angle_max="2.0" preserve_detailed_balance="0" />

    <FastDesign name="design_core" scorefxn="r15_design" disable_design="false"
    task_operations="not_core,prohibit_aa_in_core,extra_sample_rotamers_design"
    repeats="4" relaxscript="default" min_type="lbfgs_armijo_nonmonotone" />
    <FastDesign name="design_boundary" scorefxn="r15_design" disable_design="false"
    task_operations="not_boundary,prohibit_aa_in_boundary,extra_sample_rotamers_design"
    repeats="2" relaxscript="default" min_type="lbfgs_armijo_nonmonotone" />
    <FastDesign name="design_surface" scorefxn="r15_design" disable_design="false"
    task_operations="not_surface,prohibit_aa_in_surface,extra_sample_rotamers_design"
    repeats="2" relaxscript="default" min_type="lbfgs_armijo_nonmonotone" />

    <AddCompositionConstraintMover name="comp_core" filename="{comp_core}"
    selector="core" />
```

```
    <AddCompositionConstraintMover name="comp_boundary" filename="{comp_boundary}"
    selector="boundary" />
    <AddCompositionConstraintMover name="comp_surface" filename="{comp_surface}"
    selector="surface" />

    <AddConstraints name="geom_constraint">
        <AtomPairConstraintGenerator name="gen_geom_csts" ca_only="1" use_harmonic="1"
    native="1" />
    </AddConstraints>
    <RemoveConstraints name="rm_geom_csts" constraint_generators="gen_geom_csts" />

    <FastRelax name="fast_relax" scorefxn="r15_regular" disable_design="true"
    task_operations="include_current_rotamer,extra_sample_rotamers_relax" repeats="3"
    relaxscript="default" min_type="lbfgs_armijo_nonmonotone" />
</MOVERS>

<PROTOCOLS>
    <Add mover="geom_constraint" />
    <Add mover="small_move" />

    <Add mover="comp_core" />
    <Add mover="design_core" />

    <Add mover="comp_boundary" />
    <Add mover="design_boundary" />

    <Add mover="comp_surface" />
    <Add mover="design_surface" />

    <Add mover="rm_geom_csts" />
    <Add mover="fast_relax" />

    <Add metrics="record_sequence" />
</PROTOCOLS>

<OUTPUT scorefxn="r15_design" />
</ROSETTASCRIPTS>
```

In both protocols, {comp_core}, {comp_boundary}, and {comp_surface} are:

---

**Box 11: {comp_core}**

```
PENALTY_DEFINITION
TYPE ALA
FRACTION 0.10
FRACT_DELTA_START -0.05
FRACT_DELTA_END 0.05
BEFORE_FUNCTION CONSTANT
AFTER_FUNCTION QUADRATIC
PENALTIES 0 0 25
END_PENALTY_DEFINITION

PENALTY_DEFINITION
TYPE GLY
FRACTION 0.05
FRACT_DELTA_START -0.05
FRACT_DELTA_END 0.05
BEFORE_FUNCTION CONSTANT
AFTER_FUNCTION QUADRATIC
PENALTIES 0 0 25
END_PENALTY_DEFINITION

PENALTY_DEFINITION
PROPERTIES AROMATIC
```

---

**Box 11: {comp_boundary}**

```
PENALTY_DEFINITION
TYPE ALA
FRACTION 0.10
FRACT_DELTA_START -0.05
FRACT_DELTA_END 0.05
BEFORE_FUNCTION CONSTANT
AFTER_FUNCTION QUADRATIC
PENALTIES 0 0 25
END_PENALTY_DEFINITION

PENALTY_DEFINITION
TYPE GLY
FRACTION 0.05
FRACT_DELTA_START -0.05
FRACT_DELTA_END 0.05
BEFORE_FUNCTION CONSTANT
AFTER_FUNCTION QUADRATIC
PENALTIES 0 0 25
END_PENALTY_DEFINITION

PENALTY_DEFINITION
PROPERTIES HYDROPHOBIC
```

---

**Box 11: {comp_surface}**

```
PENALTY_DEFINITION
TYPE ALA
FRACTION 0.10
FRACT_DELTA_START -0.05
FRACT_DELTA_END 0.05
BEFORE_FUNCTION CONSTANT
AFTER_FUNCTION QUADRATIC
PENALTIES 0 0 25
END_PENALTY_DEFINITION

PENALTY_DEFINITION
TYPE GLY
FRACTION 0.05
FRACT_DELTA_START -0.05
FRACT_DELTA_END 0.05
BEFORE_FUNCTION CONSTANT
AFTER_FUNCTION QUADRATIC
PENALTIES 0 0 25
END_PENALTY_DEFINITION
```

---

```
FRACTION 0.15                          FRACTION 0.50
FRACT_DELTA_START -0.05                FRACT_DELTA_START -0.05
FRACT_DELTA_END 0.05                   FRACT_DELTA_END 0.05
BEFORE_FUNCTION CONSTANT               BEFORE_FUNCTION CONSTANT
AFTER_FUNCTION QUADRATIC               AFTER_FUNCTION QUADRATIC
PENALTIES 0 0 25                       PENALTIES 0 0 25
END_PENALTY_DEFINITION                 END_PENALTY_DEFINITION

PENALTY_DEFINITION                     PENALTY_DEFINITION
TYPE MET                               PROPERTIES AROMATIC
ABSOLUTE 1                             FRACTION 0.10
DELTA_START -1                         FRACT_DELTA_START -0.05
DELTA_END 1                            FRACT_DELTA_END 0.05
BEFORE_FUNCTION CONSTANT               BEFORE_FUNCTION CONSTANT
AFTER_FUNCTION QUADRATIC               AFTER_FUNCTION QUADRATIC
PENALTIES 0 0 25                       PENALTIES 0 0 25
END_PENALTY_DEFINITION                 END_PENALTY_DEFINITION

PENALTY_DEFINITION                     PENALTY_DEFINITION
PROPERTIES POLAR                       TYPE TRP
FRACTION 0.15                          ABSOLUTE 1
FRACT_DELTA_START -0.05                DELTA_START -1
FRACT_DELTA_END 0.05                   DELTA_END 1
BEFORE_FUNCTION CONSTANT               BEFORE_FUNCTION CONSTANT
AFTER_FUNCTION QUADRATIC               AFTER_FUNCTION QUADRATIC
PENALTIES 0 0 25                       PENALTIES 0 0 25
END_PENALTY_DEFINITION                 END_PENALTY_DEFINITION
```

### E.5. Iterative Human Baseline

In this section, we include an experiment on manual improvement of the one-shot human baseline on two backbone conformations: 9KGY and 9C14.

1. First, we added a hydrogen bond score term (hydrogen bond networks are important for stability),

2. Then, we increased the weights of `hbon_lr_bb` and `hbond_sr_bb` to capture $\beta$-sheet and $\alpha$-helical hydrogen bonding based on the geometry of the two conformations.

Neither modification improved performance relative to the agent's. On 9KGY, the ESMFold RMSD increased to $\approx 20$Åfor both iterations, and pLDDT decreased to $\approx 69$. On 9C14, RMSD increased to $\approx 24$Åand pLDDT decreased to $\approx 83$.

## E.6. Extended Results

In this section, we include further results that were omitted from the main text for the sake of presentation.

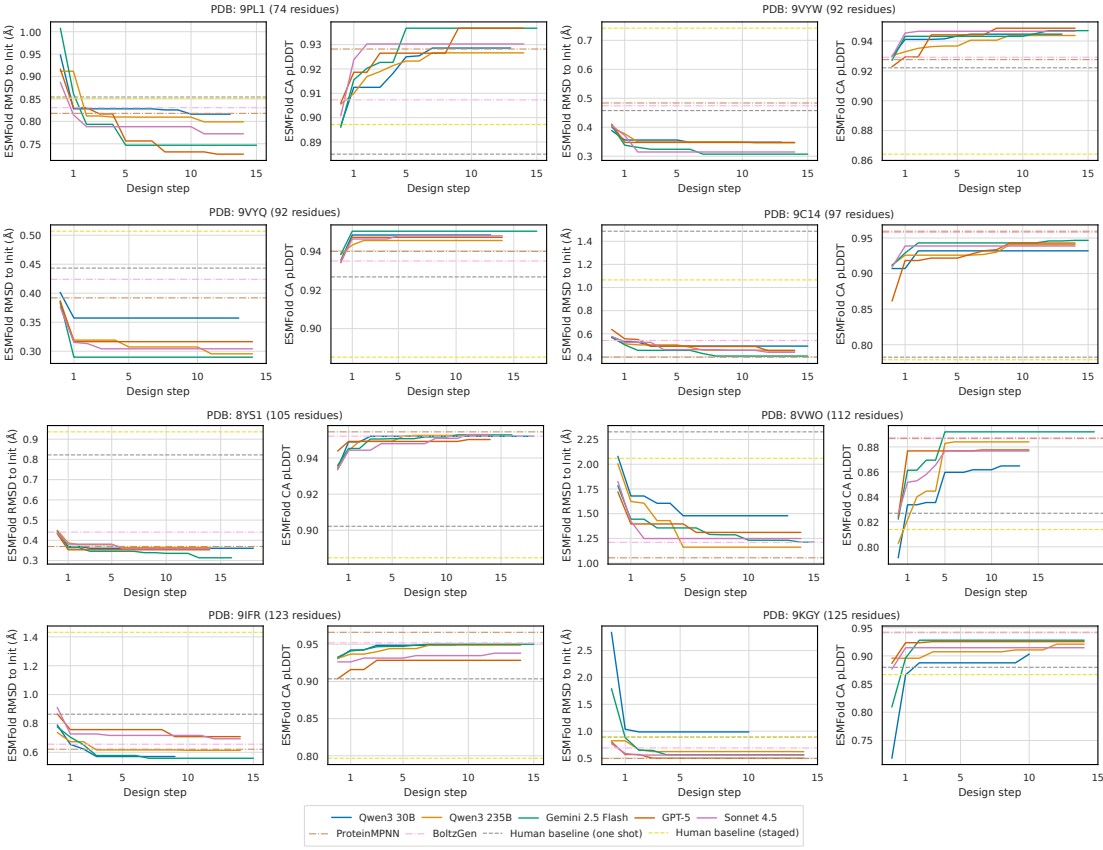

*Figure E.2.* Running best (*i.e.*, over all previous steps) ESMFold RMSD and pLDDT across 1,000 bootstrap samples of 8 trials out of the 16 total.

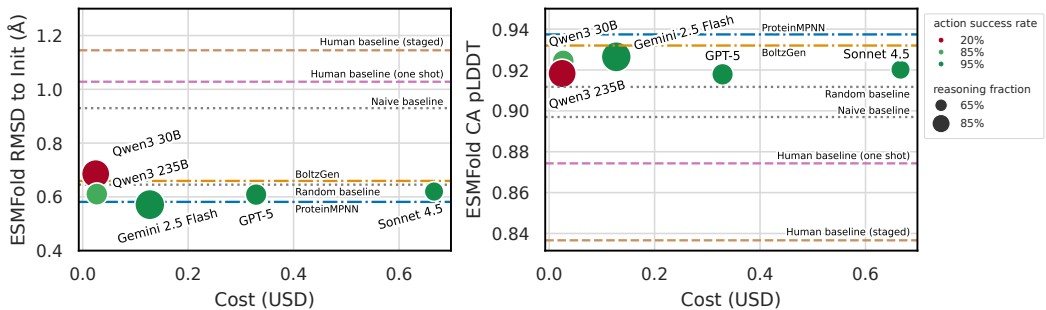

*Figure E.3.* Summary of median ESMFold RMSD and pLDDT across all 8 target backbone conformations used in this experiment.

| | 9PL1 | | 9VYW | | 9VYQ | | 9C14 | |
|---|---|---|---|---|---|---|---|---|
| Method | RMSD to init (mÅ) | pLDDT (0-100) | RMSD to init (mÅ) | pLDDT (0-100) | RMSD to init (mÅ) | pLDDT (0-100) | RMSD to init (mÅ) | pLDDT (0-100) |
| Qwen3 30B Instruct (CoT) | 816 (+0/+46) | 91.1 (-0.6/+0.2) | 359 (-11/+29) | 94.3 (-1.5/+0.2) | 363 (-6/+21) | 93.4 (-0.5/+1.5) | 493 (+0/+90) | 92.8 (-2.9/+0.0) |
| Qwen3 235B Instruct (CoT) | 809 (-11/+15) | 92.1 (-1.7/+0.1) | 349 (-3/+24) | 93.5 (-0.1/+0.2) | 321 (-26/+14) | 94.3 (-0.9/+0.2) | 481 (-23/+30) | 92.7 (-0.8/+1.4) |
| Gemini 2.5 Flash | 758 (-11/+27) | 92.5 (-2.4/+1.2) | **324 (-17/+12)** | 94.1 (-0.2/+0.2) | 306 (-16/+10) | 94.4 (-0.5/+0.2) | 421 (-14/+41) | 94.0 (-1.3/+0.6) |
| GPT-5 | **732 (-5/+37)** | 91.4 (-1.5/+0.1) | 351 (-4/+22) | 92.9 (-0.2/+0.7) | 326 (-9/+25) | **94.5 (-0.8/+0.2)** | 489 (-36/+45) | 93.8 (-1.8/+0.3) |
| Sonnet 4.5 | 788 (-16/+47) | 91.8 (-0.5/+1.2) | 341 (-26/+14) | **94.4 (-1.4/+0.2)** | **305 (-1/+16)** | 94.3 (-0.3/+0.1) | 462 (-23/+25) | 92.4 (-0.7/+1.1) |
| ProteinMPNN | 822 (-4/+7) | **92.7 (-0.0/+0.1)** | 486 (-2/+1) | 92.6 (-0.1/+0.1) | 394 (-2/+2) | 93.9 (-0.0/+0.1) | **398 (-0/+1)** | **95.9 (-0.1/+0.0)** |
| BoltzGen | 831 (-1/+3) | 90.7 (-0.1/+0.0) | 474 (-0/+2) | 92.8 (-0.0/+0.1) | 425 (-1/+1) | 93.5 (-0.0/+0.0) | 543 (-2/+2) | 95.8 (-0.1/+0.0) |
| Naive baseline | 866 (-3/+28) | 90.4 (-0.8/+1.0) | 424 (-7/+27) | 93.9 (-0.8/+0.5) | 499 (-19/+22) | 93.4 (-1.5/+0.5) | 556 (-62/+41) | 91.2 (-0.3/+1.1) |
| Random baseline | 749 (-3/+72) | 91.1 (-0.8/+1.6) | 334 (-1/+20) | 93.0 (-0.2/+1.5) | 382 (-40/+23) | 94.2 (-1.0/+0.1) | 465 (-32/+17) | 93.3 (-1.4/+0.0) |
| Human baseline (one shot) | 877 (-22/+26) | 88.5 (-1.2/+0.0) | 473 (-15/+13) | 91.9 (-0.5/+0.3) | 450 (-6/+7) | 92.6 (-0.3/+0.1) | 1503 (-15/+188) | 78.1 (-2.4/+0.2) |
| Human baseline (staged) | 865 (-13/+11) | 88.9 (-0.6/+0.8) | 757 (-15/+117) | 86.3 (-1.0/+0.1) | 565 (-58/+19) | 88.5 (-1.1/+0.1) | 1277 (-211/+284) | 76.7 (-1.7/+1.2) |
| | 8YS1 | | 8VWO | | 9IFR | | 9KGY | |
| Method | RMSD to init (mÅ) | pLDDT (0-100) | RMSD to init (mÅ) | pLDDT (0-100) | RMSD to init (mÅ) | pLDDT (0-100) | RMSD to init (mÅ) | pLDDT (0-100) |
| Qwen3 30B Instruct (CoT) | 370 (-9/+65) | 94.9 (-1.1/+0.3) | 1514 (-35/+388) | 86.0 (-6.5/+0.2) | **583 (-13/+238)** | 93.4 (-3.7/+1.4) | 986 (+0/+705) | 88.8 (-11.0/+0.0) |
| Qwen3 235B Instruct (CoT) | 377 (-13/+12) | 95.1 (-1.5/+0.1) | 1281 (-119/+139) | 88.3 (-2.3/+0.1) | 643 (-29/+51) | 94.7 (-3.7/+0.2) | 620 (-2/+201) | 88.9 (-0.8/+3.2) |
| Gemini 2.5 Flash | 336 (-22/+28) | 94.4 (-0.4/+0.8) | 1231 (-20/+128) | 87.2 (-3.6/+0.8) | 597 (-38/+14) | 94.4 (-0.4/+0.6) | 596 (-34/+57) | 90.1 (-1.9/+1.0) |
| GPT-5 | 368 (-15/+19) | 94.3 (-1.1/+0.1) | 1318 (-7/+111) | 85.7 (-2.1/+2.0) | 757 (-49/+42) | 91.3 (-0.3/+0.6) | 526 (-30/+56) | 90.4 (-1.1/+2.1) |
| Sonnet 4.5 | 377 (-18/+20) | 94.8 (-0.3/+0.5) | 1382 (-133/+39) | 85.3 (-0.4/+1.8) | 718 (-25/+27) | 92.7 (-0.5/+1.1) | 585 (-29/+103) | 90.5 (-2.0/+1.0) |
| ProteinMPNN | 370 (-0/+1) | **95.4 (-0.0/+0.0)** | **1060 (-6/+5)** | **88.6 (-0.2/+0.1)** | 624 (-3/+12) | **96.6 (-0.0/+0.0)** | **496 (-4/+6)** | **94.2 (-0.0/+0.0)** |
| BoltzGen | 442 (-1/+2) | 95.2 (-0.0/+0.0) | 1213 (-3/+3) | 88.4 (-0.1/+0.3) | 658 (-2/+2) | 95.2 (-0.0/+0.0) | 688 (-0/+3) | 94.0 (-0.1/+0.1) |
| Naive baseline | 563 (-3/+30) | 94.9 (-2.6/+0.5) | 1516 (-2/+192) | 88.5 (-1.1/+0.6) | 877 (-18/+52) | 92.4 (-2.6/+0.0) | 2139 (-110/+643) | 72.9 (-5.4/+2.0) |
| Random baseline | **349 (-1/+24)** | 94.5 (-0.5/+0.8) | 1335 (-6/+156) | 85.3 (-2.1/+2.3) | 590 (-2/+53) | 94.4 (-1.4/+0.1) | 957 (-203/+321) | 83.6 (-5.3/+5.4) |
| Human baseline (one shot) | 825 (-3/+17) | 90.2 (-0.5/+0.1) | 2334 (-7/+68) | 81.4 (-0.2/+1.2) | 868 (-4/+7) | 89.7 (-0.0/+0.6) | 897 (-5/+111) | 87.0 (-1.2/+0.9) |
| Human baseline (staged) | 952 (-16/+37) | 87.6 (-1.4/+0.9) | 2247 (-187/+149) | 78.2 (-1.4/+1.3) | 1474 (-43/+39) | 77.4 (-1.1/+2.3) | 1024 (-125/+89) | 85.7 (-0.9/+1.0) |

*Table E.2.* Per-PDB results for all methods on fixed-backbone sequence design with canonical amino acids only. We report the best step-wise median and 95% confidence interval across 1000 bootstrap samples of 8 trajectories out of 16 independent trials per method. We select the best step in terms of the smallest $5^{th}$ percentile of the RMSD between the ESMFold predicted fold and the target conformation.

| Method | RMSD to init (mÅ) | pLDDT (0-100) | Cost (USD) | Rosetta action success rate (%) | Fraction of reasoning tokens (%) |
|---|---|---|---|---|---|
| Gemini 2.5 Flash | 571 ± 312 | 92.6 ± 2.6 | 0.126 ± 0.007 | 91 ± 3 | 93 ± 1 |
| ProteinMPNN | 581 ± 245 | 93.7 ± 2.5 | na | na | na |
| GPT-5 | 608 ± 331 | 91.8 ± 2.9 | 0.329 ± 0.009 | 92 ± 2 | 69 ± 1 |
| Qwen3 235B Instruct (CoT) | 610 ± 319 | 92.4 ± 2.6 | 0.027 ± 0.003 | 86 ± 5 | 70 ± 3 |
| Sonnet 4.5 | 620 ± 355 | 92.0 ± 3.1 | 0.669 ± 0.017 | 93 ± 1 | 64 ± 2 |
| Random baseline | 645 ± 353 | 91.2 ± 4.3 | na | na | na |
| BoltzGen | 659 ± 264 | 93.2 ± 2.5 | na | na | na |
| Qwen3 30B Instruct (CoT) | 686 ± 406 | 91.8 ± 3.1 | 0.024 ± 0.002 | 25 ± 11 | 87 ± 3 |
| Naive baseline | 930 ± 601 | 89.7 ± 7.1 | na | na | na |
| Human baseline (one shot) | 1028 ± 619 | 87.4 ± 5.1 | na | na | na |
| Human baseline (staged) | 1145 ± 529 | 83.7 ± 5.3 | na | na | na |

*Table E.3.* Summary of results across all 8 PDBs for all methods on fixed-backbone sequence design. We report mean and standard deviation of the *best-of-8* median.

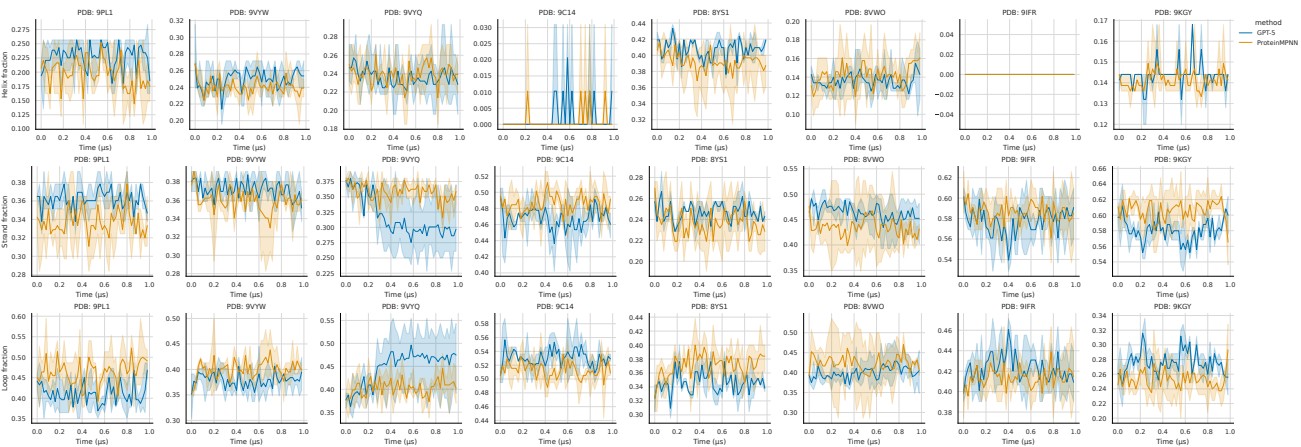

*Figure E.4.* Fraction of helices, strands, and loops as a function of MD simulation time across replicates for each target conformation.

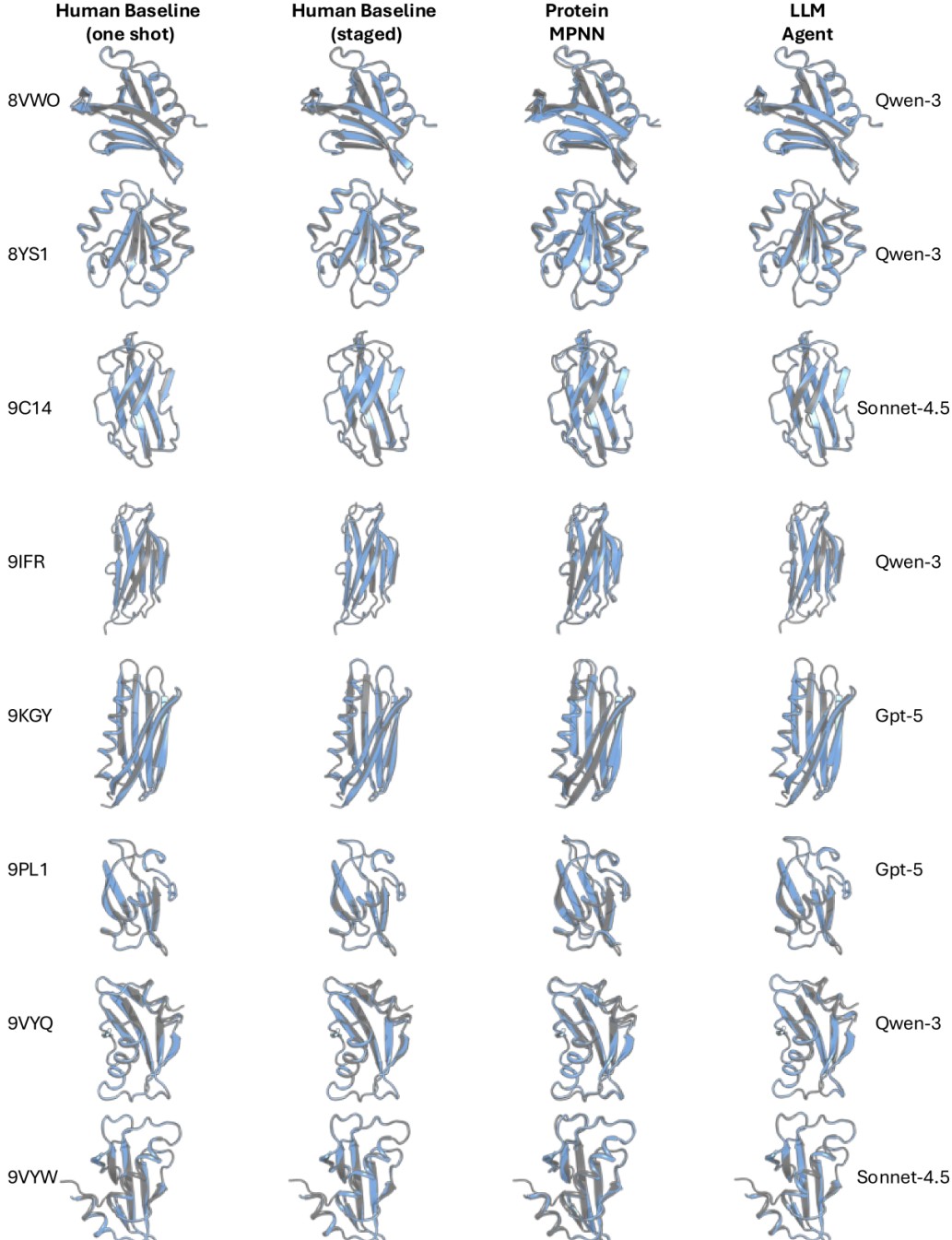

*Figure E.5.* Visualization of the best designs for each method on all backbones used in the fixed-backbone sequence design experiment.

# F. Inclusion of a Non-Canonical Amino Acid in the Core of a Protein

In this section, we include further experimental details on including N1-formyl-tryptophan (TRF)—a post-translational modification of tryptophan—in the core of an initial protein.

## F.1. Design Brief

Here, we include the specific design brief used in this experiment. This design brief is composed at runtime with the general design brief prompt template presented in Appendix B.2.

---

**Box 12: Design brief**

```
You are given an initial protein composed of standard (i.e., canonical) amino acids.

Your task is to redesign the sequence such that:
1. It includes exactly one (1) non-canonical amino acid in its core.
2. The inclusion of the non-canonical amino acid does not compromise the energetic
   stability of the protein.
3. The sequence folds into the same shape as the initial protein.

The non-canonical amino acid (NCAA) to include is N1-formyl-tryptophan, a tryptophan
   with a formyl group attached to the nitrogen in the indole ring. The 3-letter code
   for this NCAA is TRF.

You will be provided the following proxy metrics to help you assess task progress:
1. Total Rosetta energy.
2. Cavity volume and radius of gyration.
3. The RMSD between the Rosetta design and the initial protein.

At each step:
- The Rosetta environment will execute your action and produce an ensemble of
   candidate designs.
- You will be shown summary statistics of the ensemble.
- You will choose the next action to take, which will be applied to Pareto optimal
   candidates only in terms of the provided proxy metrics.
```

---

## F.2. Description of TRF

In this section, we compare the structure of TRF with the canonical amino acid TRP, and include the `params` file used to define TRF in Rosetta.

*(a)* TRP.    *(b)* TRF.

*Figure F.1.* Comparison of the structure of tryptophan (TRP) and N1-formyl-tryptophan (TRF).

---

**Box 13: TRF `params` file**

```
# Rosetta residue topology file for N1-formyl-tryptophan (aka TRF)
NAME TRF
IO_STRING TRF X
TYPE POLYMER
AA UNK
ROTAMER_AA TRP
```

---

```
BACKBONE_AA TRP
ATOM  N    Nbb  NH1  -0.5872428  -0.317
ATOM  CA   CAbb CT1   0.1074333   0.133
ATOM  C    CObb C     0.7058698   0.583
ATOM  O    OCbb O    -0.6711044  -0.517
ATOM  CB   CH2  CT2  -0.1388399   0.158
ATOM  CG   CH0  CY   -0.0086544  -0.092
ATOM  CD1  aroC CA    0.0477593  -0.092
ATOM  CD2  CH0  CPT   0.0000246   0.033
ATOM  NE1  Ntrp NY   -0.5120383  -0.367
ATOM  CE2  CH0  CPT   0.1302101   0.033
ATOM  CE3  aroC CA   -0.0824262  -0.092
ATOM  CZ1  COO  CC    0.4512171   0.133
ATOM  CZ2  aroC CA   -0.0824262  -0.092
ATOM  CZ3  aroC CA   -0.0824262  -0.092
ATOM  OH1  OOC  OC   -0.5773882  -0.517
ATOM  CH2  aroC CA   -0.0824262  -0.092
ATOM  H    HNbb H     0.4161782   0.283
ATOM  HD1  Haro HP    0.1171916   0.158
ATOM  HZ1  Hapo HA    0.0561730   0.033
ATOM  HZ2  Haro HP    0.1171916   0.158
ATOM  HH2  Haro HP    0.1171916   0.158
ATOM  HZ3  Haro HP    0.1171916   0.158
ATOM  HE3  Haro HP    0.1171916   0.158
ATOM  HA   Hapo HB    0.1331620   0.033
ATOM 1HB   Hapo HA    0.0954940   0.033
ATOM 2HB   Hapo HA    0.0954940   0.033

ATOM_ALIAS 1HB    HB2
ATOM_ALIAS 2HB    HB3

LOWER_CONNECT N
UPPER_CONNECT C
BOND  N    CA
BOND  N    H
BOND  CA   C
BOND  CA   CB
BOND  CA   HA
BOND_TYPE  C    O    2
BOND  CB   CG
BOND  CB  1HB
BOND  CB  2HB
BOND_TYPE  CG   CD1  ARO
BOND_TYPE  CG   CD2  ARO
CUT_BOND   CG   CD2
BOND_TYPE  CD1  NE1  ARO
BOND  CD1  HD1
BOND_TYPE  CD2  CE2  ARO
CUT_BOND   CD2  CE2
BOND_TYPE  CD2  CE3  ARO
BOND_TYPE  NE1  CE2  ARO
BOND_TYPE  CE2  CZ2  ARO
BOND_TYPE  CE3  CZ3  ARO
BOND  CE3  HE3
BOND_TYPE  CZ2  CH2  ARO
BOND  CZ2  HZ2
BOND_TYPE  CZ3  CH2  ARO
BOND  CZ3  HZ3
BOND  CH2  HH2
BOND  NE1  CZ1
BOND_TYPE  CZ1 OH1 2
BOND CZ1  HZ1
CHI 1  N    CA   CB   CG
CHI 2  CA   CB   CG   CD1
```

```
CHI 3  CD1  NE1  CZ1  OH1
PROTON_CHI 3 SAMPLES 2 0 180 EXTRA 1 15
ADD_RING 1 AROMATIC  CG   CD1  NE1  CE2  CD2
NU  1  CD2  CG   CD1  NE1
NU  2  CG   CD1  NE1  CE2
NU  3  CD1  NE1  CE2  CD2
NU  4  NE1  CE2  CD2  CG
NU  5  CE2  CD2  CG   CD1
ADD_RING 2 AROMATIC  CE2  CD2  CE3  CZ3  CH2  CZ2
NU  6  CZ2  CE2  CD2  CE3
NU  7  CE2  CD2  CE3  CZ3
NU  8  CD2  CE3  CZ3  CH2
NU  9  CE3  CZ3  CH2  CZ2
NU 10  CZ3  CH2  CZ2  CE2
NU 11  CH2  CZ2  CE2  CD2
PROPERTIES PROTEIN ALPHA_AA L_AA HYDROPHOBIC AROMATIC SC_ORBITALS METALBINDING CYCLIC
METAL_BINDING_ATOMS O
NBR_ATOM CB
# APL CB to sidechain heavyatom distance; swept all chi combos at 5 degree intervals
NBR_RADIUS 5.37514
FIRST_SIDECHAIN_ATOM CB
RAMA_PREPRO_FILENAME all.ramaProb prepro.ramaProb
ACT_COORD_ATOMS CD2 CE3 END
ICOOR_INTERNAL    N      0.000000    0.000000    0.000000  N    CA   C
ICOOR_INTERNAL    CA     0.000000  180.000000    1.458001  N    CA   C
ICOOR_INTERNAL    C      0.000000   68.800003    1.523258  CA   N    C
ICOOR_INTERNAL  UPPER  149.999985   63.800007    1.328685  C    CA   N
ICOOR_INTERNAL    O   -180.000000   59.200005    1.231015  C    CA   UPPER
ICOOR_INTERNAL    CB  -122.800000   69.625412    1.521736  CA   N    C
ICOOR_INTERNAL    CG     0.000068   66.465866    1.498746  CB   CA   N
ICOOR_INTERNAL    CD1    0.000148   53.263374    1.362720  CG   CB   CA
ICOOR_INTERNAL    NE1 -179.969818   69.843536    1.372938  CD1  CG   CB
ICOOR_INTERNAL    CE2   -0.111805   71.100000    1.372132  NE1  CD1  CG
ICOOR_INTERNAL    CZ2  179.958145   49.861912    1.385949  CE2  NE1  CD1
ICOOR_INTERNAL    CH2 -179.974182   62.500000    1.395024  CZ2  CE2  NE1
ICOOR_INTERNAL    CZ3    0.127723   58.500000    1.372108  CH2  CZ2  CE2
ICOOR_INTERNAL    CE3   -0.067888   59.000000    1.389848  CZ3  CH2  CZ2
ICOOR_INTERNAL    CD2    0.021383   61.268948    1.400378  CE3  CZ3  CH2
ICOOR_INTERNAL    HE3 -179.851593   59.300000    1.089539  CE3  CZ3  CD2
ICOOR_INTERNAL    HZ3  179.965530   59.271282    1.090294  CZ3  CH2  CE3
ICOOR_INTERNAL    HH2  179.952423   60.590862    1.090289  CH2  CZ2  CZ3
ICOOR_INTERNAL    HZ2 -179.891754   57.900000    1.090237  CZ2  CE2  CH2
ICOOR_INTERNAL    CZ1  179.658524   54.960213    1.373036  NE1  CD1  CE2
ICOOR_INTERNAL    OH1  179.658524   54.960213    1.373036  CZ1  NE1  CD1
ICOOR_INTERNAL    HZ1  179.658524   54.960213    1.373036  CZ1  NE1  OH1
ICOOR_INTERNAL    HD1  179.990616   55.100000    1.088516  CD1  CG   NE1
ICOOR_INTERNAL    1HB  121.200000   70.500000    1.090167  CB   CA   CG
ICOOR_INTERNAL    2HB  117.600000   70.500000    1.089792  CB   CA   1HB
ICOOR_INTERNAL    HA  -119.000000   71.500000    1.089883  CA   N    CB
ICOOR_INTERNAL  LOWER -150.000000   58.300003    1.328685  N    CA   C
ICOOR_INTERNAL    H   -180.000000   60.849998    1.010000  N    CA   LOWER
```

### F.3. Initial Proteins

In this section, we describe the inclusion criteria and preprocessing steps used to select 4 proteins to use in this experiment (see Figure F.2).

**Selection of PDB structures for NCAA design.**    We selected a small set of structurally distinct *de novo* proteins to enable evaluation of NCAA incorporation across diverse structural contexts and to assess whether incorporation of non-canonical amino acids can further stabilize existing de novo designs. The selected structures were: 6V67 (Bryan et al., 2021), 1QYS (Accuracy, 2003), 8UZL (Berhanu et al., 2024), and 7SQ3 (Doyle et al., 2023). PDB 6V67 is a compact mini-protein (40 residues) that provides a minimal system for probing NCAA compatibility in highly constrained backbones. 1QYS is a

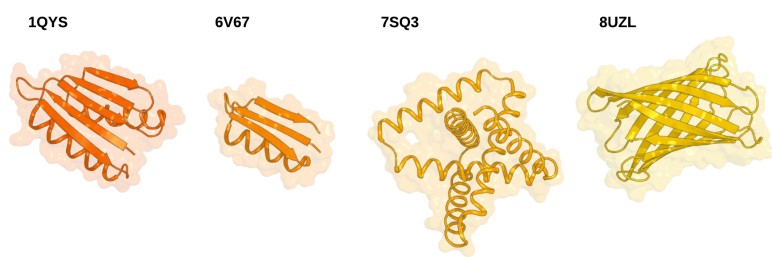

*Figure F.2.* The 4 proteins for non-canonical design.

106-residue Rosetta-designed protein that serves as a foundational reference for computational protein design. 8UZL is a designed transmembrane $\beta$-barrel (149 residues), serving as a scaffold for NCAA design in membrane-associated contexts. Finally, 7SQ3 is a trefoil-knot protein (153 residues) representing a topologically complex fold and a stringent test for design methods.

**Preprocessing of PDB structures.** All structures chosen for non-canonical amino acid design were processed in Py-MOL (Schrödinger, LLC, 2025b;a) to remove alternative conformations (`altlocs B-G`) and all heteroatoms and X-ray crystallographic copies.

## F.4. Environment State

Here, we include an illustrative example of the textual representation of the RosettaScripts environment state for the task with a non-canonical residue. We omit the tabular history summary for the sake of readability

---

**Box 14: Environment state**

```
The RosettaScripts environment successfully executed your action:

**Step Number:** 3
**Action Name:** rotamer_change
**Results:**

- Number of designs: 120 (21 Pareto optimal)

- Average total Rosetta energy: -154.82 ± 6.62

- Average compositional before design 733.33, and after design 16.67 (-716.67)

- TRF inclusion summary:
-- List of core residue indices after design: 5,23,26,30,37
-- Percentage of designs with at least one TRF residue: 100.00% (min: 12, max: 18)
-- Percentage of designs with exactly one TRF residue in the core: 13.33% (min: 1,
    max: 4)
-- Most common TRF residue positions:
    8,25,37,35,30,11,13,39,5,27,3,15,32,23,19,20,24,28,21
-- Average interresidue_repulsion at TRF residues: 1.01 ± 0.59
-- Average ramachandran_preference at TRF residues: 0.05 ± 0.29

- Top 5 most common outlier residue types. A residue is an outlier if its energy term
    is above the 90-th quantile of the per-residue energy term for that design. For
    each outlier residue type, we include the most common outlier positions along the
    sequence (positions are 1-based):

-- interresidue_repulsion:

Average per-sequence 90-th quantile: 1.41

TRF: 100.00% (positions: 25,5,15,39,13,33)
MET: 62.50%  (positions: 40,26)
LEU: 33.33%  (positions: 28,15,14)
```

```
CYS: 21.67%  (positions: 4,29)
TYR: 20.00%  (positions: 33)

-- ramachandran_preference:

Average per-sequence 90-th quantile: 0.58

TRF: 100.00% (positions: 8,16,10)
ASN: 96.67%  (positions: 9,17)
MET: 73.33%  (positions: 40)
CYS: 63.33%  (positions: 6)
ASP: 20.00%  (positions: 18)

- Structural metrics:

Cavity volume (Å^3)                        :    13.07 ± 11.98
Radius of gyration (Å)                     :     9.90 ± 0.01
Penalty for buried unsatisfied Hydrogen bonds:  142.08 ± 108.35
RMSD to initial structure (Å)              :     0.19 ± 0.02

---

**History Summary:**

{history_summary}
```

## F.5. Expert Human Baseline Protocol

In this section, we include the expert human written protocol used as a baseline in the design task with non-canonical amino acids.

### Box 15: Human design protocol

```
<ROSETTASCRIPTS>
<SCOREFXNS>
    <ScoreFunction name="r15_regular" weights="ref2015_cst.wts"/>
    <ScoreFunction name="r15_design" weights="ref2015.wts">
        <Reweight scoretype="aa_composition" weight="1.0"/>
        <Reweight scoretype="atom_pair_constraint" weight="1.0"/>
    </ScoreFunction>
    <ScoreFunction name="r15_post" weights="ref2015_cst.wts">
        <Reweight scoretype="aa_composition" weight="1.0"/>
        <Reweight scoretype="atom_pair_constraint" weight="1.0"/>
        <Reweight scoretype="rg" weight="1.0" />
        <Reweight scoretype="buried_unsatisfied_penalty" weight="1.0" />
    </ScoreFunction>
</SCOREFXNS>

<PACKER_PALETTES>
    <CustomBaseTypePackerPalette name="ncaa_palette" additional_residue_types="TRF"/>
</PACKER_PALETTES>

<RESIDUE_SELECTORS>
    <Layer name="core" select_core="true" select_boundary="false"
    select_surface="false" use_sidechain_neighbors="true"/>
</RESIDUE_SELECTORS>

<RESIDUE_LEVEL_TASK_OPERATIONS>
    <RestrictToRepackingRLT name="RestrictToRepacking"/>
</RESIDUE_LEVEL_TASK_OPERATIONS>

<TASKOPERATIONS>
    <IncludeCurrent name="include_current_rotamer"/>
    <ExtraRotamersGeneric name="extra_sample_rotamers_design" ex1="1" ex1aro="1"/>
```

```
    <ExtraRotamersGeneric name="extra_sample_rotamers_relax" ex1="1" ex2="1"
    ex1aro="1" ex2aro="1"/>

    <DesignRestrictions name="not_core">
        <Action selector_logic="NOT core"
    residue_level_operations="RestrictToRepacking"/>
    </DesignRestrictions>

    <ProhibitSpecifiedBaseResidueTypes name="limited_aa_design"
    base_types="ARG,LYS,ASP,GLU,CYS,PRO"/>
</TASKOPERATIONS>

<SIMPLE_METRICS>
    <SequenceMetric name="record_sequence" output_mode="basename"/>
</SIMPLE_METRICS>

<FILTERS>
    <CavityVolume name="cav_vol" confidence="0.0" />
</FILTERS>

<MOVERS>
    <Small name="small_move" scorefxn="r15_design" temperature="0.5" nmoves="1000"
    angle_max="2.0" preserve_detailed_balance="0"/>

    <FastDesign name="design_with_ncaa_only" scorefxn="r15_design"
    disable_design="false"
    task_operations="not_core,limited_aa_design,include_current_rotamer,
    extra_sample_rotamers_design" packer_palette="ncaa_palette" repeats="5"
    relaxscript="default" min_type="lbfgs_armijo_nonmonotone"/>

    <AddCompositionConstraintMover name="comp_ncaa" filename="{comp_ncaa}"
    selector="core"/>
    <AddCompositionConstraintMover name="comp_core" filename="{comp_core}"
    selector="core"/>

    <AddConstraints name="geom_constraint">
        <AtomPairConstraintGenerator name="gen_geom_csts" ca_only="1" use_harmonic="1"
    native="1"/>
    </AddConstraints>
    <RemoveConstraints name="rm_geom_csts" constraint_generators="gen_geom_csts"/>

    <FastRelax name="fast_relax" scorefxn="r15_regular" disable_design="true"
    task_operations="include_current_rotamer,extra_sample_rotamers_relax" repeats="3"
    relaxscript="default" min_type="lbfgs_armijo_nonmonotone"/>
</MOVERS>

<PROTOCOLS>
    <Add mover="geom_constraint"/>
    <Add mover="small_move"/>
    <Add mover="comp_ncaa"/>
    <Add mover="comp_core"/>
    <Add mover="design_with_ncaa_only"/>
    <Add mover="rm_geom_csts"/>
    <Add mover="fast_relax"/>
    <Add filter="cav_vol"/>
    <Add metrics="record_sequence"/>
</PROTOCOLS>

<OUTPUT scorefxn="r15_post"/>
</ROSETTASCRIPTS>
```

where {comp_ncaa} and {comp_core} are:

**Box 16: {comp_ncaa}**

```
PENALTY_DEFINITION
TYPE TRF
ABSOLUTE 1
PENALTIES 1000 0 100
DELTA_START -1
DELTA_END 1
BEFORE_FUNCTION CONSTANT
AFTER_FUNCTION QUADRATIC
END_PENALTY_DEFINITION
```

**Box 16: {comp_core}**

```
PENALTY_DEFINITION
TYPE ALA
FRACTION 0.10
FRACT_DELTA_START -0.05
FRACT_DELTA_END 0.05
BEFORE_FUNCTION CONSTANT
AFTER_FUNCTION QUADRATIC
PENALTIES 0 0 25
END_PENALTY_DEFINITION

PENALTY_DEFINITION
TYPE GLY
FRACTION 0.05
FRACT_DELTA_START -0.05
FRACT_DELTA_END 0.05
BEFORE_FUNCTION CONSTANT
AFTER_FUNCTION QUADRATIC
PENALTIES 0 0 25
END_PENALTY_DEFINITION

PENALTY_DEFINITION
PROPERTIES AROMATIC
FRACTION 0.15
FRACT_DELTA_START -0.05
FRACT_DELTA_END 0.05
BEFORE_FUNCTION CONSTANT
AFTER_FUNCTION QUADRATIC
PENALTIES 0 0 25
END_PENALTY_DEFINITION

PENALTY_DEFINITION
TYPE MET
ABSOLUTE 1
DELTA_START -1
DELTA_END 1
BEFORE_FUNCTION CONSTANT
AFTER_FUNCTION QUADRATIC
PENALTIES 0 0 25
END_PENALTY_DEFINITION

PENALTY_DEFINITION
PROPERTIES POLAR
FRACTION 0.15
FRACT_DELTA_START -0.05
FRACT_DELTA_END 0.05
BEFORE_FUNCTION CONSTANT
AFTER_FUNCTION QUADRATIC
PENALTIES 0 0 25
END_PENALTY_DEFINITION
```

## F.6. Extended Results

In this section, we include further results that were omitted by the main text for the sake of presentation.

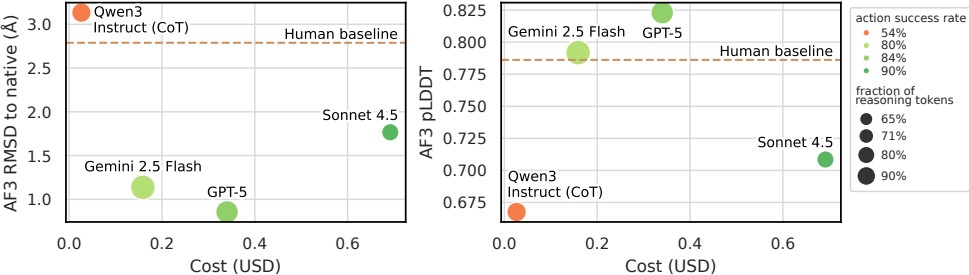

*Figure F.3.* Summary of results for including 1 TRF residue in the core of an input protein. We report the average cost of one run with 30 model queries, the average action success rate, and the fraction of output tokens that were reasoning.

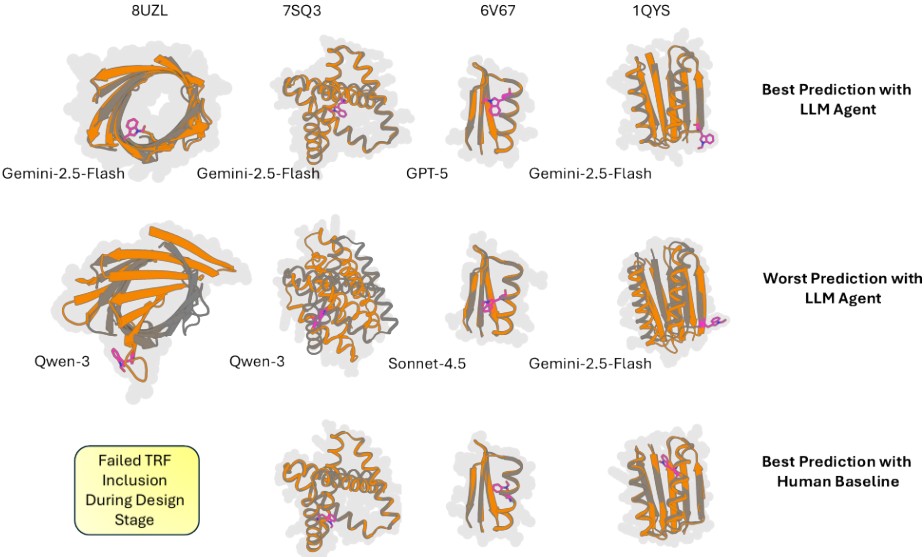

*Figure F.4.* Illustrations of good and bad designs for inclusion of TRF in the core of a given protein across all 4 PDBs considered in the experiment.

*Table F.4.* Summary of results across all 4 PDBs for all methods on inclusion of 1 TRF residue in the core of a given protein. We report mean and standard deviation of results in Table F.5.

| Method | Cost (USD) | Rosetta action success rate (%) | Fraction of reasoning tokens (%) | Rosetta RMSD to native (mÅ) | AF3 RMSD to native (mÅ) | AF3 pLDDT (0-100) |
|---|---|---|---|---|---|---|
| GPT-5 | 0.34 ± 0.03 | 84.0 ± 5.2 | 81.0 ± 1.2 | 162 ± 45 | 855 ± 526 | 82.2 ± 3.6 |
| Gemini 2.5 Flash | 0.15 ± 0.01 | 80.1 ± 7.5 | 88.1 ± 1.5 | 164 ± 41 | 1,137 ± 882 | 79.2 ± 4.7 |
| Sonnet 4.5 | 0.70 ± 0.01 | 89.1 ± 4.0 | 65.2 ± 2.6 | 171 ± 45 | 1,766 ± 1,447 | 70.8 ± 11.2 |
| Qwen3 235B Instruct (CoT) | 0.03 ± 0.01 | 53.4 ± 22.9 | 71.0 ± 2.0 | 195 ± 70 | 3,133 ± 3,023 | 66.8 ± 19.3 |
| Human baseline | na | na | na | 201 ± 34 | 2,789 ± 3,574 | 78.6 ± 9.8 |

*Table F.5.* Per-PDB results for all methods on inclusion of a TRF residue in the core of an input protein structure. For each method, we filter successful designs, and select the top 10 in order of increasing RMSD between the Rosetta Pose and the native PDB. We validate the fold of the selected designs with AF3, and report RMSD between the predicted fold and the native structure as well as pLDDT. We report average and standard deviation over the top 10 designs. The human baseline is marked as "na" for 8UZL because it fails to include any TRF residues.

| | 1QYS | | | 6V67 | | |
| --- | --- | --- | --- | --- | --- | --- |
| Method | Rosetta RMSD (mÅ) | AF3 RMSD (mÅ) | AF3 pLDDT (0-100) | Rosetta RMSD (mÅ) | AF3 RMSD (mÅ) | AF3 pLDDT (0-100) |
| Qwen3 235B Instruct (CoT) | 174 ± 4 | **627 ± 119** | **86.6 ± 4.0** | 111 ± 2 | 444 ± 189 | 79.0 ± 3.1 |
| Gemini 2.5 Flash | 170 ± 10 | 798 ± 677 | 85.9 ± 2.3 | 109 ± 2 | **347 ± 38** | 78.2 ± 0.8 |
| GPT-5 | 166 ± 1 | 796 ± 81 | 84.6 ± 2.2 | **104 ± 2** | 374 ± 60 | 81.0 ± 1.7 |
| Sonnet 4.5 | **157 ± 5** | 773 ± 252 | 82.3 ± 1.2 | 114 ± 3 | 604 ± 143 | 78.5 ± 1.6 |
| Human baseline | 182 ± 3 | 985 ± 158 | 84.8 ± 2.9 | 181 ± 6 | 476 ± 0 | **83.7 ± 0.0** |
| | 7SQ3 | | | 8UZL | | |
| Method | Rosetta RMSD (mÅ) | AF3 RMSD (mÅ) | AF3 pLDDT (0-100) | Rosetta RMSD (mÅ) | AF3 RMSD (mÅ) | AF3 pLDDT (0-100) |
| Qwen3 235B Instruct (CoT) | 221 ± 6 | 5,280 ± 5,102 | 56.2 ± 22.5 | 276 ± 18 | 6,182 ± 4,212 | 45.2 ± 13.6 |
| Gemini 2.5 Flash | 209 ± 7 | 2,394 ± 3,814 | 75.2 ± 20.8 | 168 ± 11 | **1,008 ± 998** | 77.4 ± 10.9 |
| GPT-5 | 215 ± 2 | **652 ± 38** | **85.7 ± 4.1** | **164 ± 6** | 1,599 ± 2,046 | **77.9 ± 13.8** |
| Sonnet 4.5 | **205 ± 6** | 3,744 ± 4,998 | 59.6 ± 16.6 | 208 ± 17 | 1,941 ± 2,862 | 63.0 ± 7.3 |
| Human baseline | 241 ± 4 | 6,906 ± 8,206 | 67.3 ± 27.7 | na | na | na |

*Figure F.5.* RMSD to native structure as a function of MD simulation time. We show mean and 95% CI across PDBs, designs, and replicates.

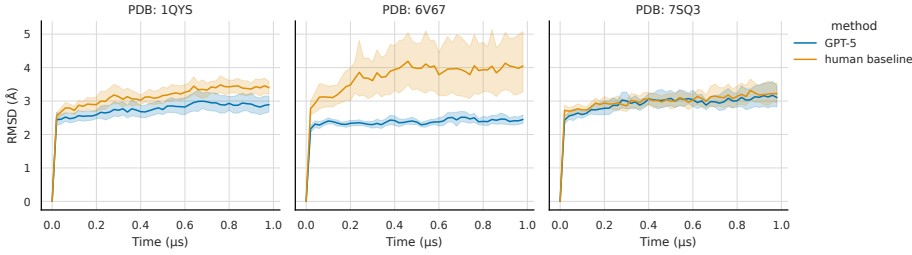

*Figure F.6.* Fraction of helices, loops, and strands as a function of MD simulation time. We show mean and 95% CI across PDBs, designs, and replicates.

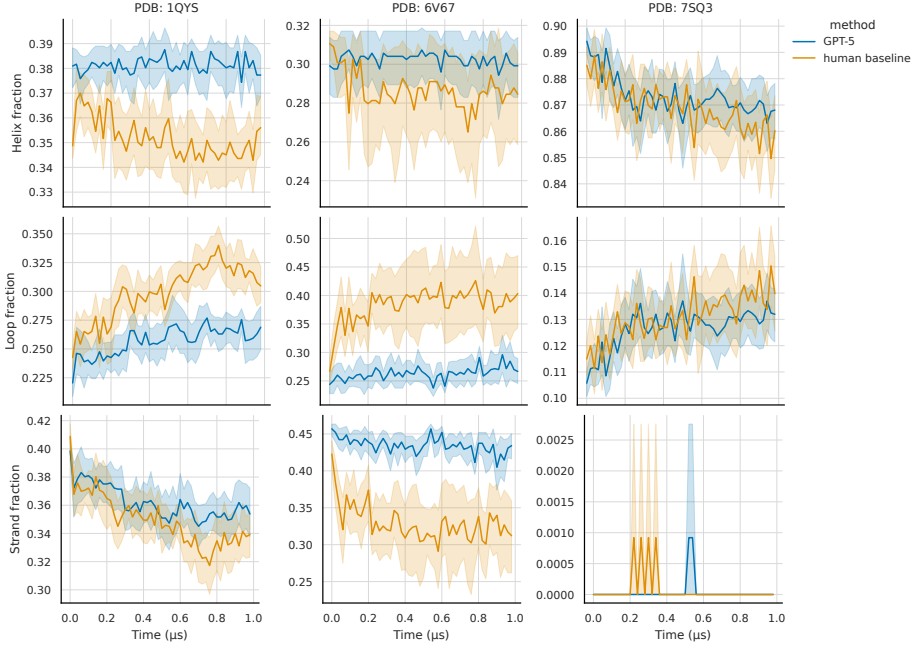

# G. RosettaScripts Environment Action Types

In this section, we include further details on the action types available to Agent Rosetta. For each action, we include the RosettaScripts XML template which is populated at runtime. In the templates:

- Fields enclosed within curly brackets (*e.g.*, {field_name}) are populated with the parameters generated by the agent.

- Fields enclosed within squared brackets (*e.g.*, [field_name]) are populated programmatically by the environment, depending on the action and the user-defined task parameters.

We remark that the go_back_to action does not call Rosetta and it does not need an XML template.

### G.1. `rotamer_change`

The RosettaScripts XML template for the rotamer_change action is:

---
**Box 17: `rotamer_change` XML action template**

```
<ROSETTASCRIPTS>
<SCOREFXNS>
    <ScoreFunction name="scoring_pre" weights="ref2015">
        <Reweight scoretype="aa_composition" weight="1.0" />
    </ScoreFunction>
    <ScoreFunction name="guidance" weights="ref2015">
        [guidance_weights]
    </ScoreFunction>
    <ScoreFunction name="scoring_post" weights="ref2015">
        [scoring_weights]
    </ScoreFunction>
</SCOREFXNS>

<PACKER_PALETTES>
    <CustomBaseTypePackerPalette name="design_palette" [additional_residue_types] />
</PACKER_PALETTES>

<RESIDUE_SELECTORS>
    <Layer name="_core_residues" select_core="true" select_boundary="false"
    select_surface="false"/>
    {residue_selectors}
</RESIDUE_SELECTORS>

<TASKOPERATIONS>
    {task_operations}
    <IncludeCurrent name="include_current" />
</TASKOPERATIONS>

<SIMPLE_METRICS>
    [simple_metrics]
    <TotalEnergyMetric name="aa_composition_pre" scorefxn="scoring_pre"
    scoretype="aa_composition"/>
    <SequenceMetric name="record_sequence" output_mode="basename" />
</SIMPLE_METRICS>

<FILTERS>
    [filters]
</FILTERS>

<MOVERS>
    {aa_comp_mover}
    <AddConstraints name="geom_constraint">
        <AtomPairConstraintGenerator name="gen_geom_csts" ca_only="1" use_harmonic="1"
    native="1" />
    </AddConstraints>
```
---

```
    <FastDesign name="design" scorefxn="guidance" repeats="1"
    packer_palette="design_palette"
    task_operations="include_current[operations_names]" />
</MOVERS>

<PROTOCOLS>
    [protocol]
    <Add metrics="record_sequence" />
</PROTOCOLS>

<OUTPUT scorefxn="scoring_post"/>
</ROSETTASCRIPTS>
```

The agent-generated parameters of the action are:

- {aa_comp_mover}: the amino acid compositional penalty blocks guiding the Monte Carlo search during Rosetta's FastDesign Mover.

- {residue_selectors}: the definitions of regions of the protein sequence compositional penalty blocks and task operations should apply to.

- {task_operations}: the design restrictions which define the base residue types allowed or prohibited in different regions of the protein.

The user-specified parameters of the action are:

- [guidance_weights], which edit the weights of the energy function used during design. For example, in our experiments, we set aa_composition and atom_pair_constraint to 1.

- [scoring_weights], which edit the weights of the energy function used for scoring post-design. These weights are a superset of the guidance terms, and they may include terms like rg to compute the radius of gyration of the output protein, or buried_unsatisfied_penalty to estimate the number of buried unsatisfied hydrogen bonds.

- [additional_residue_types], which include the additional non-canonical amino acids to use (*e.g.*, TRF in our experiment).

- [simple_metrics], which include further evaluation terms such as the post-design amino acid compositional penalty energy or the per-residue stratification of the van der Waals force and the pre-proline Ramachandran preference.

- [filters], which specify additional quality metrics such as cavity volume.

An example action call is:

---

**Box 18: Agent Rosetta's reasoning (GPT-5)**

```
*Clarifying penalty definitions**

I see that the documentation for penalties mentions 'selector_name,' but the example
    uses <comp_selector_name>. I'm thinking we should use comp_selector_name to be on
    the safe side. Although the detailed docs say selector_name, the examples suggest
    otherwise. I'll need to align with the expected parser in the environment.

Next, I'll craft the XML strings for the residue selectors and penalties for core and
    surface constraints. Hm, I need to clarify if we're working with counts or ratios,
    as the system can auto-detect ambiguities.**Defining residue restrictions**

I'm assuming that using a ratio will work well here. For the "prohibit Pro" globally
    via penalty, I plan to set the parameters for that penalty definition.

Now, regarding residue restrictions, I think we can create a selector for all
    residues. However, the requirement states that the selector_name is necessary.
    Since there's no "All" selector available, I can define a Layer that captures
```

---

core, boundary, or surface. Alternatively, using an "Or" approach for these three layers seems simpler, so I might go with that and define an all_res layer accordingly.**Compiling residue restrictions**

I'm thinking about how to use residue_restrictions items effectively. I'll set the type to prohibit for residues P with the selector_name as all_res.

Next, I plan to restrict the core to hydrophobic residues: A, V, L, I, M, F, W, Y. For the boundary, I'll include A, V, L, I, M, F, W, Y, S, T, N, Q, and H. Finally, I'll restrict the surface to D, E, K, R, H, N, Q, S, T, Y, and A. Including A on the surface is fine.

Packing restrictions-none for now. Let's compile this all together!

---

**Box 19: Agent Rosetta's action call (GPT-5)**

```
<action tag="run">
<name>rotamer_change</name>
<residue_selectors>
<Layer name="core" select_core="true" select_boundary="false" select_surface="false"/>
<Layer name="boundary" select_core="false" select_boundary="true"
    select_surface="false"/>
<Layer name="surface" select_core="false" select_boundary="false"
    select_surface="true"/>
<Or name="all_res" selectors="core,boundary,surface"/>
</residue_selectors>
<penalties>
<item>
<comp>
PENALTY_DEFINITION
TYPE P
SHAPE ABOVE
TARGET 0
RADIUS 0
BOUNDARY QUADRATIC
STRENGTH 100
END_PENALTY_DEFINITION
</comp>
</item>
<item>
<comp>
PENALTY_DEFINITION
TYPE D,E,K,R,H,N,Q,S,T
SHAPE ABOVE
TARGET 0
RADIUS 0
BOUNDARY LINEAR
STRENGTH 50
END_PENALTY_DEFINITION
</comp>
<comp_selector_name>core</comp_selector_name>
</item>
<item>
<comp>
PENALTY_DEFINITION
TYPE D,E,K,R,H,N,Q,S,T,Y
SHAPE BELOW
TARGET 0.6
RADIUS 0
BOUNDARY LINEAR
STRENGTH 30
END_PENALTY_DEFINITION
</comp>
<comp_selector_name>surface</comp_selector_name>
```

```
</item>
</penalties>
<residue_restrictions>
<item>
<type>prohibit</type>
<residues>P</residues>
<selector_name>all_res</selector_name>
</item>
<item>
<type>restrict</type>
<residues>A,V,L,I,M,F,W,Y</residues>
<selector_name>core</selector_name>
</item>
<item>
<type>restrict</type>
<residues>A,V,L,I,M,F,W,Y,S,T,N,Q,H</residues>
<selector_name>boundary</selector_name>
</item>
<item>
<type>restrict</type>
<residues>A,D,E,K,R,H,N,Q,S,T,Y</residues>
<selector_name>surface</selector_name>
</item>
</residue_restrictions>
</action>
```

## G.2. backbone_change

The RosettaScripts XML template for the backbone_change action is:

**Box 20: backbone_change XML action template**

```
<ROSETTASCRIPTS>
<SCOREFXNS>
    <ScoreFunction name="scoring_post" weights="ref2015">
        [scoring_weights]
    </ScoreFunction>
</SCOREFXNS>

<PACKER_PALETTES>
    <CustomBaseTypePackerPalette name="palette" [additional_residue_types] />
</PACKER_PALETTES>

<RESIDUE_SELECTORS>
    <Layer name="_core_residues" select_core="true" select_boundary="false"
    select_surface="false"/>
    {residue_selectors}
</RESIDUE_SELECTORS>

<MOVE_MAP_FACTORIES>
    [movemap_factory]
</MOVE_MAP_FACTORIES>

<SIMPLE_METRICS>
    [simple_metrics]
    <SequenceMetric name="record_sequence" output_mode="basename" />
</SIMPLE_METRICS>

<FILTERS>
    [filters]
</FILTERS>

<MOVERS>
    {mover}
```

```
        <MinMover name="min" bb="false" chi="true" tolerance="1.0e-07"/>
</MOVERS>

<PROTOCOLS>
    <Add mover="backbone_change" />
    <Add mover="min" />
    [protocol]
    <Add metrics="record_sequence" />
</PROTOCOLS>

<OUTPUT scorefxn="scoring_post"/>
</ROSETTASCRIPTS>
```

The user-specified parameters of the actions are:

- [scoring_weights], which edit the weights of the energy function used to score the design after the backbone mover has been applied. These weights may include terms like rg to compute the radius of the gyration of the output protein, or buried_unsatisfied_penalty to estimate the number of buried unsatisfied hydrogen bonds.

- [additional_residue_types], which include the additional non-canonical amino acids to use (*e.g.*, TRF in our experiment).

- [simple_metrics], which include further evaluation terms such as the per-residue stratification of the van der Waals force and the pre-proline Ramachandran preference.

- [filters], which specify additional quality metrics such as cavity volume.

An example action call is:

---

**Box 21: Agent Rosetta's reasoning (Gemini 2.5 Flash)**

```
**Contemplating Backbone Adjustments**

I'm now focusing on the `backbone_change` action. The preceding `rotamer_change`
    yielded solid improvements in energy and ESMFold scores. However, the penalty for
    unsatisfied hydrogen bonds is alarmingly high, suggesting the backbone's current
    configuration is a misfit. Therefore, a backbone adjustment is now the priority.

**Deciding Mover Parameters**

After deciding to employ a `backrub` mover, I'm now wrestling with parameter
    selection. The goal is to carefully refine the backbone without drastically
    altering the overall structure, as measured by a decent ESMFold RMSD. The high
    penalty for unsatisfied hydrogen bonds dictates a need for a delicate touch, so
    I'm leaning toward a relatively conservative set of `backrub` parameters to allow
    for minimal deviations while addressing the hydrogen bonding issue.

**Analyzing Backrub Implementation**

The choice of parameters for the `backrub` mover is almost finalized. I plan to apply
    the mover to all residues, addressing the widespread hydrogen bond issues. I've
    decided to use a `Calpha` pivot atom and the standard minimum number of atoms for
    the move. I'm now drafting the script, specifically focusing on the mover
    selector, ensuring it targets the entire protein sequence.

**Defining the Backrub Protocol**

I'm now putting the finishing touches on the XML script. Specifically, I am writing
    the block that defines the `Index` selector to target all residues for the
    `backrub` mover. After that, I will define the `mover_params`, using default
    values for `pivot_atoms`, `min_atoms`, and `max_atoms`. My intention is to apply
    this to all residues.

**Defining the Backrub Parameters**
```

I've decided on the `backrub` mover with specific parameters. The goal is to address
    the high unsatisfied hydrogen bond penalty from the previous stage. Therefore, I
    will define an XML script. The `backrub` mover will be implemented with
    `pivot_atoms="Calpha" min_atoms="3" max_atoms="12"`. A residue selector will
    target all residues from 1 to 123 using the `Index` selector in the XML script,
    setting the `mover_selector_name`.

**Box 22: Agent Rosetta's action call (Gemini 2.5 Flash)**

```
<action tag="run">
<name>backbone_change</name>
<mover_name>backrub</mover_name>
<mover_params>pivot_atoms="N,C" min_atoms="3" max_atoms="12"</mover_params>
<residue_selectors>
<Index name="all_res_backbone" resnums="1-123"/>
</residue_selectors>
<mover_selector_name>all_res_backbone</mover_selector_name>
</action>
```

## H. Error Correction

In this section, we include examples of Agent Rosetta correcting its own mistakes.

### H.1. Wrong Backbone Mover Name

**Box 23: Wrong action call (Qwen3 Instruct)**

```
<action tag="run">
<name>backbone_change</name>
<mover_name>backrub</mover_name>
<mover_params>nmoves="100" temperature="0.6"</mover_params>
</action>
```

**Box 24: Error message**

```
The RosettaScripts environment failed to run your action with the following error:

```
[ ERROR ]: Caught exception:
File: src/protocols/rosetta_scripts/RosettaScriptsParser.cc:1315
Input rosetta scripts XML file XXXXX failed to validate against the rosetta scripts
    schema. Use the option -parser::output_schema <output filename> to output the
    schema to a file to see all valid options.
Your XML has failed validation.  The error message below will tell you where in your
    XML file the error occurred.  Here's how to fix it:
1) If the validation fails on something obvious, like an illegal attribute due to a
    spelling error (perhaps you used scorefnction instead of scorefunction), then you
    need to fix your XML file.
2) If you haven't run the XML rewriter script and this might be pre-2017 Rosetta XML,
    run the rewriter script (tools/xsd_xrw/rewrite_rosetta_script.py) on your input
    XML first.  The attribute values not being in quotes (scorefunction=talaris2014
    instead of scorefunction="talaris2014") is a good indicator that this is your
    problem.
3) If you are a developer and neither 1 nor 2 worked - email the developer's mailing
    list or try Slack.
4) If you are an academic or commercial user - try the Rosetta Forums
    https://www.rosettacommons.org/forum
Error messages were:
From line 34:
Error: Element 'Backrub', attribute 'nmoves': The attribute 'nmoves' is not allowed.
29:    <FILTERS>
30:        <CavityVolume name="cav_vol" confidence="0.0" />
```

```
31:      </FILTERS>
32:
33:      <MOVERS>
34:          <Backrub name="backbone_change" nmoves="100" temperature="0.6" />
35:          <MinMover name="min" bb="false" chi="true" tolerance="1.0e-07"/>
36:      </MOVERS>
37:
38:      <PROTOCOLS>
39:          <Add mover="backbone_change" />
From line 34:
Error: Element 'Backrub', attribute 'temperature': The attribute 'temperature' is not
    allowed.
29:      <FILTERS>
30:          <CavityVolume name="cav_vol" confidence="0.0" />
31:      </FILTERS>
32:
33:      <MOVERS>
34:          <Backrub name="backbone_change" nmoves="100" temperature="0.6" />
35:          <MinMover name="min" bb="false" chi="true" tolerance="1.0e-07"/>
36:      </MOVERS>
37:
38:      <PROTOCOLS>
39:          <Add mover="backbone_change" />
------------------------------------------------------------
Warning messages were:
------------------------------------------------------------
```

**Instructions:**

First carefully read the error message, then reason about the possible causes of the
    error, and finally write the corrected full action call with all its arguments.

Your corrected action call must preserve the intent of your previous action. Some
    common sources of errors are:
  - Syntactic mistakes in the action call: wrong parameter names, wrong XML structure.
  - Semantic mistakes in the action call: wrong parameter values, logically invalid
    combinations of parameters.
Be specific in your reasoning: the expert biomolecular scientists on your team should
    be able to understand, review, and critique your solution.

Now write the corrected full action call with all its arguments following the
    formatting instructions in the system prompt:
- First, you must write your step-by-step reasoning inside <reasoning></reasoning>
    tags.
- Then, write your action in the following format

```
<action tag="run">
<name>action_name</name>
<arg_name1>arg_value1</arg_name1>
<arg_name2>arg_value2</arg_name2>
...
</action>
```

---

**Box 25: Agent Rosetta's reasoning (Qwen3 Instruct)**

The error message indicates that the `Backrub` mover does not accept the attributes
    `nmoves` and `temperature` as provided. This suggests that either the parameter
    names are incorrect or they must be nested within a sub-element (such as
    `<MoveMap>` or `<MoverConfiguration>`), which is common in RosettaScripts.

Upon reviewing standard RosettaScripts syntax, the `BackrubMover` typically does not
    use `nmoves` or `temperature` directly as XML attributes. Instead:

```
- The number of moves is usually controlled by wrapping the mover in a `CountedLoop`
    or applying it multiple times in a protocol.
- The "temperature" for Monte Carlo acceptance is not directly exposed in high-level
    XML; instead, small perturbations are made and accepted based on a Metropolis
    criterion with an implicit energy-based scoring.

However, the `BackrubMover` does support a `pivot_atoms` or `pivot_residues`
    definition and operates over a segment of the backbone. Since no parameters were
    specified correctly, and the mover failed validation, I must use valid,
    schema-compliant parameters.

Looking at valid RosettaScripts schema for `BackrubMover`, it supports:
- `name`: identifier
- `pivot_atoms`: comma-separated list of atom names to pivot around (e.g., "CA,CB")
- `min_segment_length`, `max_segment_length`: length of the backrub segment
- `backrub_trials`: number of backrub moves to attempt

But crucially, `nmoves` and `temperature` are not valid attributes.

To achieve gentle backbone sampling without violating the schema, I will use the
    `SmallMover` instead, which is more reliably parameterized in standard
    RosettaScripts and supports controlled, small perturbations.

The `SmallMover` accepts:
- `angle_max`: maximum angular deviation (in degrees) for phi/psi moves
- `temperature`: optional, but often omitted; sampling is governed by `angle_max`
- `nmoves`: number of small moves to apply

Alternatively, to remain consistent with the intent of backbone flexibility while
    ensuring schema compliance, I will switch to `SmallMover` with valid parameters.

Intent: Introduce small, physically realistic backbone perturbations to allow the
    structure to relax and escape unfavorable conformations (e.g., high energy due to
    all-glycine strain), while preserving overall fold.

Best course: Use `SmallMover` with moderate `angle_max` (e.g., 5-10 degrees) and
    `nmoves=10`, applied to the full backbone.
```

**Box 26: Corrected action call (Qwen3 Instruct)**

```
<action tag="run">
<name>backbone_change</name>
<mover_name>small</mover_name>
<mover_params>angle_max="7.0" nmoves="10"</mover_params>
</action>
```

## H.2. Wrong ResidueSelector Argument

**Box 27: Wrong action call (Gemini 2.5 Flash)**

```
<action tag="run">
<name>rotamer_change</name>
<residue_selectors>
<Layer name="core_selector" select_core="true" select_boundary="false"
    select_surface="false"/>
<Layer name="surface_selector" select_core="false" select_boundary="false"
    select_surface="true"/>
<SecondaryStructure name="structured_regions" helix="true" strand="true" turn="false"/>
<Neighborhood name="gly_outlier_neighbors" resnums="86" distance="8.0"/>
</residue_selectors>
<penalties>
<item>
<comp>
# Encourage polar residues in the core to satisfy H-bonds
```

```
PENALTY_DEFINITION
TYPE S,T,N,Q,D,E,H,Y
...
```

**Box 28: Error message**

```
 ERROR ]: Caught exception:

File: src/protocols/rosetta_scripts/RosettaScriptsParser.cc:1315
Input rosetta scripts XML file "XXXXX" failed to validate against the rosetta scripts
    schema. Use the option -parser::output_schema <output filename> to output the
    schema to a file to see all valid options.
Your XML has failed validation.  The error message below will tell you where in your
    XML file the error occurred.  Here's how to fix it:

1) If the validation fails on something obvious, like an illegal attribute due to a
    spelling error (perhaps you used scorefnction instead of scorefunction), then you
    need to fix your XML file.
2) If you haven't run the XML rewriter script and this might be pre-2017 Rosetta XML,
    run the rewriter script (tools/xsd_xrw/rewrite_rosetta_script.py) on your input
    XML first.  The attribute values not being in quotes (scorefunction=talaris2014
    instead of scorefunction="talaris2014") is a good indicator that this is your
    problem.
3) If you are a developer and neither 1 nor 2 worked - email the developer's mailing
    list or try Slack.
4) If you are an academic or commercial user - try the Rosetta Forums
    https://www.rosettacommons.org/forum

Error messages were:
From line 26:
Error: Element 'SecondaryStructure', attribute 'helix': The attribute 'helix' is not
    allowed.

21:
22:     <RESIDUE_SELECTORS>
23:         <Layer name="_core_residues" select_core="true" select_boundary="false"
    select_surface="false"/>
24:         <Layer name="core_selector" select_core="true" select_boundary="false"
    select_surface="false"/>
25: <Layer name="surface_selector" select_core="false" select_boundary="false"
    select_surface="true"/>
26: <SecondaryStructure name="structured_regions" helix="true" strand="true"
    turn="false"/>
27: <Neighborhood name="gly_outlier_neighbors" resnums="86" distance="8.0"/>
28:     </RESIDUE_SELECTORS>
29:
30:     <TASKOPERATIONS>
31:         <ProhibitSpecifiedBaseResidueTypes name="restriction_0" base_types="GLY"
    selector="structured_regions" />
From line 26:
Error: Element 'SecondaryStructure', attribute 'strand': The attribute 'strand' is not
    allowed.

21:
22:     <RESIDUE_SELECTORS>
23:         <Layer name="_core_residues" select_core="true" select_boundary="false"
    select_surface="false"/>
24:         <Layer name="core_selector" select_core="true" select_boundary="false"
    select_surface="false"/>
25: <Layer name="surface_selector" select_core="false" select_boundary="false"
    select_surface="true"/>
```

```
26: <SecondaryStructure name="structured_regions" helix="true" strand="true"
    turn="false"/>
27: <Neighborhood name="gly_outlier_neighbors" resnums="86" distance="8.0"/>
28:     </RESIDUE_SELECTORS>
29:
30:     <TASKOPERATIONS>
31:         <ProhibitSpecifiedBaseResidueTypes name="restriction_0" base_types="GLY"
    selector="structured_regions" />
From line 26:
Error: Element 'SecondaryStructure', attribute 'turn': The attribute 'turn' is not
    allowed.

21:
22:     <RESIDUE_SELECTORS>
23:         <Layer name="_core_residues" select_core="true" select_boundary="false"
    select_surface="false"/>
24:         <Layer name="core_selector" select_core="true" select_boundary="false"
    select_surface="false"/>
25: <Layer name="surface_selector" select_core="false" select_boundary="false"
    select_surface="true"/>
```

---

**Box 29: Agent Rosetta's reasoning (Gemini 2.5 Flash)**

```
The error occurs because the 'SecondaryStructure' residue selector uses invalid
    attributes ('helix', 'strand', 'turn') that are not recognized in the current
    RosettaScripts schema. The correct way to define secondary structure-based
    selection is to use the 'SecondaryStructure' selector with the 'ss' attribute,
    which accepts single-letter codes for secondary structure elements: 'H' for helix,
    'E' for strand, and 'L' for loop.

To preserve the original intent - selecting structured regions (helices and strands)
    to restrict glycine in those elements - I will redefine the 'SecondaryStructure'
    selector using the 'ss' attribute with the value "HE", representing helical and
    beta-strand regions.

Additionally, I will ensure all other residue selectors ('Layer', 'Neighborhood') are
    correctly specified and maintain the compositional penalties and residue
    restrictions logic:
- Penalize low polar residue content in the core to satisfy hydrogen bonds.
- Globally penalize high glycine content to improve backbone stability.
- Prohibit glycine in structured regions and near a known outlier residue (residue 86)
    to reduce flexibility and improve fold stability.

The corrected XML syntax will now conform to the schema while maintaining the
    biophysical design goals.
```

---

**Box 30: Corrected action call (Gemini 2.5 Flash)**

```
<action tag="run">
<name>rotamer_change</name>
<residue_selectors>
<Layer name="core_selector" select_core="true" select_boundary="false"
    select_surface="false"/>
<Layer name="surface_selector" select_core="false" select_boundary="false"
    select_surface="true"/>
<SecondaryStructure name="structured_regions" ss="HE"/>
<Neighborhood name="gly_outlier_neighbors" resnums="86" distance="8.0"/>
</residue_selectors>
<penalties>
<item>
<comp>
# Encourage polar residues in the core to satisfy H-bonds
PENALTY_DEFINITION
TYPE S,T,N,Q,D,E,H,Y
```

```
...
```

## H.3. Invalid Segment for Backbone Move

---

**Box 31: Wrong action all (Qwen3 Instruct)**

```
<action tag="run">
<name>backbone_change</name>
<mover_name>backrub</mover_name>
<residue_selectors>
<Index name="trf_site" resnums="109"/>
</residue_selectors>
<mover_selector_name>trf_site</mover_selector_name>
</action>
```

---

**Box 32: Error message**

```
The RosettaScripts environment failed to run your action with the following error:

```
^^[[0m^^[[31m^^[[1m
ERROR: Assertion 'segments_.size()' failed.
ERROR:: Exit from: src/protocols/backrub/BackrubMover.cc line: 267
^^[[0m^^[[0m^^[[31m^^[[1m
ERROR: Assertion 'segments_.size()' failed.
ERROR:: Exit from: src/protocols/backrub/BackrubMover.cc line: 267
^^[[0m^^[[35m
[ ERROR ]: Caught exception:
File: src/protocols/backrub/BackrubMover.cc:267
[ ERROR ] UtilityExitException
ERROR: Assertion 'segments_.size()' failed.
^^[[0m
```

**Instructions:**

First carefully read the error message, then reason about the possible causes of the
    error, and finally write the corrected full action call with all its arguments.

Your corrected action call must preserve the intent of your previous action. Some
    common sources of errors are:
  - Syntactic mistakes in the action call: wrong parameter names, wrong XML structure.
  - Semantic mistakes in the action call: wrong parameter values, logically invalid
    combinations of parameters.
Be specific in your reasoning: the expert biomolecular scientists on your team should
    be able to understand, review, and critique your solution.

Now write the corrected full action call with all its arguments following the
    formatting instructions in the system prompt:
- First, you must write your step-by-step reasoning inside <reasoning></reasoning>
    tags.
- Then, write your action in the following format:

<action tag="run">
<name>action_name</name>
<arg_name1>arg_value1</arg_name1>
<arg_name2>arg_value2</arg_name2>
...
</action>
```

---

**Box 33: Agent Rosetta's reasoning (Qwen3 Instruct)**

```
The error occurs in the BackrubMover with the message "Assertion 'segments_.size()'
    failed", which indicates that the mover did not find any valid backbone segments
```

to perturb. The Backrub mover requires a defined segment (typically 3-9 residues)
    around a pivot point to perform local backbone adjustments. Simply specifying a
    single residue selector is insufficient - the mover needs a proper movemap or
    segment definition to operate.

In our previous call, we selected only residue 109 via an Index selector and applied
    the backrub mover, but we did not define a proper segment or movemap for the mover
    to act on. The Backrub mover expects a region of the backbone to be designated as
    flexible, centered on the pivot residue.

To fix this, we need to define a local region around residue 109 (e.g., residues
    107-111) as the moving segment. This can be achieved by using a properly
    constructed residue selector that includes neighboring residues. Additionally, we
    must ensure the mover has a valid segment to act upon.

We will revise the residue selector to include a short segment centered on residue 109
    and apply the backrub mover to that segment. This preserves our intent to locally
    perturb the backbone near the target site for potential NCAA incorporation while
    satisfying the mover's requirements.

**Box 34: Corrected action call (Qwen3 Instruct)**

```
<action tag="run">
<name>backbone_change</name>
<mover_name>backrub</mover_name>
<residue_selectors>
<Index name="trf_site" resnums="107-111"/>
</residue_selectors>
<mover_selector_name>trf_site</mover_selector_name>
</action>
```

## I. Simplified Rosetta Syntax

In this section, we include the `rotamer_change` action documentation which describe our simplified syntax for composition penalties and TaskOperations:

**Box 35: `rotamer_change` action documentation**

```
'rotamer_change' runs Rosetta's FastDesign Mover, an iterative pack-and-minimize
    protocol that samples side-chain rotamers and uses gradient-based minimization to
    settle on the lowest-energy configuration. Each cycle repacks the designated
    residues with the rotamer library, then performs all-atom minimization before a
    Monte-Carlo accept/reject decision.

**'residue_selectors' argument:**

ResidueSelectors are used to define logical selections of residues in a structure.
    They act as a flexible query language for picking out subsets of residues based on
    various criteria such as residue type, position, secondary structure, chain,
    neighborhood, and more.

Some example residue selectors are:

- Conformation independent residue selectors: 'Index', 'Slice', 'ResidueName', e.g.:

<Index name="string" resnums="string">
<ResidueName name="string" residue_names="string">

- Conformation dependent residue selectors: 'Layer', 'Bonded', 'Neighborhood', e.g.:

<Layer name="string" select_core="bool" select_boundary="bool" select_surface="bool">
<Neighborhood name="string" resnums="string" distance="float"/>
```

```
- Logical residue selectors: 'And', 'Or', 'Not', e.g.:

<And name="string" selectors="string">
<Not name="string" selector="string">

If no residue selectors are needed for the action, leave this argument empty or omit
    it.

**'penalties' argument:**

A List of compositional penalties that define how Rosetta penalizes (or rewards)
    certain residue types or residue properties, depending on their relative abundance
    in the sequence.

The 'penalties' argument must follow this syntax:

<penalties>
    <item>
        <comp>comp1</comp>
    </item>
    <item>
        <comp>comp2</comp>
        <selector_name>selector</selector_name>
    </item>
    ...
</penalties>

, and each item must have these fields:

- 'comp' (required): one or more penalty definition blocks.
- 'selector_name' (optional): the name of a previously defined residue selector that
    specifies which residues the penalty blocks applies to. If left empty or omitted,
    the penalty definition blocks in 'comp' will be applied globally to the entire
    sequence.

Each penalty definition block must follow this modified RosettaScripts syntax:

# Brief description of the goal of the block
PENALTY_DEFINITION
TYPE <string>                      # list of one- or three-letter residue codes
    separated by a comma
SHAPE <OUTSIDE | ABOVE | BELOW>    # shape of the penalty, one of OUTSIDE, ABOVE, or
    BELOW
TARGET <int or float>              # target count or ratio of residues
RADIUS <int or float>              # radius of the interval around the target
BOUNDARY <function>                # the type of penalty boundary, one of CONSTANT,
    LINEAR, or QUADRATIC
STRENGTH <int>                     # strength of the penalty
END_PENALTY_DEFINITION

Each shape option defines a range [MIN_RANGE, MAX_RANGE] of acceptable residue counts
    or ratios:

- OUTSIDE: the range is [TARGET - RADIUS, TARGET + RADIUS], and values outside the
    range are penalized.
- ABOVE: the range is (-inf, TARGET], RADIUS is ignored, and values above the target
    are penalized.
- BELOW: the range is [TARGET, inf), RADIUS is ignored, and values below the target
    are penalized.

For all shapes, the penalty boundary options are:

- CONSTANT: constant penalty equal to STRENGTH.
- LINEAR: linear penalty that increases with slope of STRENGTH.
```

```
– QUADRATIC: quadratic penalty, STRENGTH is the first value of the penalty outside the
    range.

Finally, the STRENGTH value is of the same unit of measure as other terms in the
    Rosetta energy function. As a general guideline:

– Use small values (e.g., 1-10) for weak penalties.
– Use medium values (e.g., 10-100) for moderate penalties.
– Use large values (e.g., 100-1000) for strong penalties.

You should write penalties that steer the Monte-Carlo search in Rosetta towards the
    task objective while avoiding mutually impossible requirements.

**'residue_restrictions' argument:**

A list of restrictions that define which residue types are permitted or prohibited at
    different positions along the sequence. Residue restrictions reduce the
    combinatorially large optimization space of the Monte-Carlo search in Rosetta.

The 'residue_restrictions' argument must follow this syntax:

<residue_restrictions>
    <item>
        <type>type1</type>
        <residues>res1</residues>
        <selector_name>selector1</selector_name>
    </item>
    <item>
        <type>type2</type>
        <residues>res2</residues>
        <selector_name>selector2</selector_name>
    </item>
    ...
</residue_restrictions>

, and each item must have these fields:

– 'type' (required, either 'restrict' or 'prohibit'): whether the restriction
    specifies the residue types that are allowed or prohibited.
– 'residues' (required): the list of one- or three-letter residue codes separated by a
    comma (e.g., '<residues>A,G</residues>' or '<residues>ALA,GLY</residues>' for
    alanine and glycine).
– 'selector_name' (required): the name of a previously defined residue selector that
    specifies which residues the restriction applies to.

If you leave this argument empty or omit it, Rosetta will perfom design with all
    residues types at every position.

**'packing_restrictions' argument:**

A list of residue selectors that define which residues should not be designed but
    repacked only. Packing restrictions maintain the identities of the specified
    residues.

The 'packing_restrictions' argument must follow this syntax:

<packing_restrictions>
    selector1,selector2,...
</packing_restrictions>

, where 'selector1,selector2,...' are the names of previously defined residue
    selectors separated by a comma.
```

```
If you leave this argument empty or omit it, Rosetta will allow packing and design at
    every position.

---

**Example `rotamer_change` action call:**

The following action call

<action tag="run">
<name>rotamer_change</name>
<residue_selectors>
<ResidueSelector1 name="selector1" />
<ResidueSelector3 name="selector3" />
<And name="selector4" selectors="selector1,selector3">
<ResidueSelector2 name="selector2" />
</residue_selectors>
<penalties>
<item>
<comp>comp1</comp>
</item>
<item>
<comp>comp2</comp>
<comp_selector_name>selector4</comp_selector_name>
</item>
</penalties>
</action>

will:

- Apply the penalty definition blocks in `comp1` globally to the entire sequene.
- Apply the penalty definition blocks in `comp2` to `selector4`, which in turn
    composes `selector1` with `selector3`.

Remember to use valid RosettaScripts syntax while leveraging the expressivity of all
    arguments.
```

## J. Evaluation of LLMs at Generating Amino Acid Compositional Penalty Blocks

In this section, we include further details on how we compared different LLMs at generating amino acid penalty blocks with the original RosettaScripts syntax versus our simplified one. First, we include the list of 9 prompts we used for evaluation, 3 per penalty shape type:

| |
|---|
| Write a compositional penalty block that penalizes more than 5 prolines. Use a linear boundary with slope of 10. |
| Write a compositional penalty block that penalizes more than 10 lysines. Use a linear boundary with slope of 20. |
| Write a compositional penalty block that penalizes more than 20 glycines. Use a linear boundary with slope of 30. |
| Write a compositional penalty block that penalizes less than 5 prolines. Use a linear boundary with slope of 10. |
| Write a compositional penalty block that penalizes less than 10 lysines. Use a linear boundary with slope of 20. |
| Write a compositional penalty block that penalizes less than 20 glycines. Use a linear boundary with slope of 30. |
| Write a compositional penalty block that penalizes proline content outside the range of 4 to 10 residues. Use a linear boundary with slope of 10. |
| Write a compositional penalty block that penalizes lysine content outside the range of 5 to 15 residues. Use a linear boundary with slope of 20. |
| Write a compositional penalty block that penalizes glycine content outside the range of 5 to 25 residues. Use a linear boundary with slope of 30. |

RosettaScripts accepts penalty blocks that select residues by type or property with constant, linear, or quadratic boundaries. We limited our evaluation to compositional penalties by residue type with linear boundaries because they are easier to verify.

The system prompt for generating responses with the original RosettaScripts syntax is:

**Box 36: System prompt for generating compositional penalties with RosettaScripts syntax**

```
You are an expert RosettaScripts coding agent that supports scientists in biomolecular
    design tasks:
```

- Rosetta is a computational toolkit for modeling, predicting, and designing
  biomolecular structures and interactions, using physics-based energy functions and
  stochastic search.
- RosettaScripts is Rosetta's XML-based interface for assembling custom modeling
  protocols by combining movers, filters, and scoring terms.

Your task is to write compositional penalty blocks as instructed by the user.
  Compositional penalty blocks are used to bias the Monte Carlo search in Rosetta.

Each penalty definition block must follow this syntax:

```
PENALTY_DEFINITION
TYPE <string>                             # list of one- or three-letter residue
   codes separated by commas
- Exactly one of:
ABSOLUTE <int>                            # target count of residues
DELTA_START <int>                         # target range start, relative to target
   count, can be negative
DELTA_END <int>                           # target range end, relative to target
   count
- or -
FRACTION <float>                          # target ratio of residues
FRACT_DELTA_START <float>                 # target range start, relative to target
   fraction, can be negative
FRACT_DELTA_END <float>                   # target range end, relative to target
   fraction
PENALTIES <float1> <float2> <float3> ...     # one or more values, interpolated
   over the range
BEFORE_FUNCTION <shape>                       # shape before DELTA_START, default is
   QUADRATIC. Can be QUADRATIC, LINEAR, or CONSTANT.
AFTER_FUNCTION <shape>                        # shape after DELTA_END, default is
   QUADRATIC. Can be QUADRATIC, LINEAR, or CONSTANT.
END_PENALTY_DEFINITION
```

Rosetta computes the penalty definition range as:
- For ABSOLUTE / DELTA: [ABSOLUTE + DELTA_START, ABSOLUTE + DELTA_END]
- For FRACTION / FRACT_DELTA: [FRACTION + FRACT_DELTA_START, FRACTION +
  FRACT_DELTA_END]
All delta values (i.e., DELTA_START, DELTA_END, FRACT_DELTA_START, FRACT_DELTA_END)
  must be nonzero.

The PENALTIES line follows these rules:
- If DELTA_START and DELTA_END were used, then entries must be provided for every
  integer value from DELTA_START to DELTA_END. These values represent the energetic
  penalty for having N residues too few, N+1 residues too few, N+2 residues too few
  ... M-1 residues too many, M residues too many.
- If FRACT_DELTA_START and FRACT_DELTA_END are used, then any number of penalty values
  may be specified; they will be linearly interpolated within the range
  [FRACT_DELTA_START, FRACT_DELTA_END].
In either case, the end functions are applied if residue type counts fall outside of
  the range.

The BEFORE_FUNCTION and AFTER_FUNCTION define the behavior of the penalty function
  outside of the range:
- If CONSTANT, the first or last penalty repeats.
- If LINEAR, the slope is determined by the first two or last two penalty values.
- If QUADRATIC, the penalty is centered at zero and it passes through the first two or
  last two penalty values.

{reasoning_formatting}
Write your final response inside <response></response> tags, for example:

<response>

```
PENALTY_DEFINITION
...
END_PENALTY_DEFINITION
</response>

Do not include any extra text or comments in your final response.
```

We note how the `PENALTIES` can be counter-intuitive and difficult to parse. Its behavior changes depending on other choices like the use of integer or fractional targets, and it requires the generation of several numbers, which increases the chances of error. We include an example prompt with wrong responses from different LLMs:

**Box 37: Prompt**

```
Write a compositional penalty block that penalizes more than 5 prolines.
Use a linear boundary with slope of 10.
```

**Box 38: Qwen3 235B Instruct (CoT)**

```
PENALTY_DEFINITION
TYPE P
ABSOLUTE 5
DELTA_START 0
DELTA_END 1
PENALTIES 0 10
AFTER_FUNCTION LINEAR
END_PENALTY_DEFINITION
```

**Mistake:** Missing `BEFORE_FUNCTION` line equal to `CONSTANT`.

**Box 38: Gemini 2.5 Flash**

```
PENALTY_DEFINITION
TYPE P
ABSOLUTE 5
DELTA_START 1
DELTA_END 2
PENALTIES 10 20
BEFORE_FUNCTION CONSTANT
AFTER_FUNCTION LINEAR
END_PENALTY_DEFINITION
```

**Mistake:** `DELTA_START` should be less than or equal to 0. The number of integers on the `PENALTIES` line should be 3.

**Box 38: Sonnet 4.5**

```
PENALTY_DEFINITION
TYPE P
ABSOLUTE 5
DELTA_START -5
DELTA_END 0
PENALTIES 0 0 0 0 -10 0
BEFORE_FUNCTION CONSTANT
AFTER_FUNCTION LINEAR
END_PENALTY_DEFINITION
```

**Mistake:** The `PENALTIES` line favors 4 prolines instead of penalizing more than 5 only.

In our simplified syntax, Agent Rosetta does generate a `PENALTIES` line, but it directly specifies the shape of the penalty (*i.e.*, `ABOVE`, `BELOW`, `OUTSIDE`). The system prompt for the simple syntax is:

**Box 39: System prompt for generating compositional penalties with our simple syntax**

```
You are an expert RosettaScripts coding agent that supports scientists in biomolecular
    design tasks:

- Rosetta is a computational toolkit for modeling, predicting, and designing
    biomolecular structures and interactions, using physics-based energy functions and
    stochastic search.
- RosettaScripts is Rosetta's XML-based interface for assembling custom modeling
    protocols by combining movers, filters, and scoring terms.

Your task is to write compositional penalty blocks as instructed by the user.
    Compositional penalty blocks are used to bias the Monte Carlo search in Rosetta.

Each penalty definition block must follow this modified RosettaScripts syntax:

PENALTY_DEFINITION
TYPE <string>                      # list of one- or three-letter residue codes
    separated by commas
SHAPE <OUTSIDE | ABOVE | BELOW>    # shape of the penalty, one of OUTSIDE, ABOVE, or
    BELOW
TARGET <int or float>              # target count or ratio of residues
RADIUS <int or float>              # radius of the interval around the target
BOUNDARY <function>                # the type of penalty boundary, one of CONSTANT,
    LINEAR, or QUADRATIC
STRENGTH <int>                     # strength of the penalty
END_PENALTY_DEFINITION
```

```
Each shape option defines a range [MIN_RANGE, MAX_RANGE] of acceptable residue counts
    or ratios:
- OUTSIDE: the range is [TARGET - RADIUS, TARGET + RADIUS], and values outside the
    range are penalized.
- ABOVE: the range is (-inf, TARGET], RADIUS is ignored, and values above the target
    are penalized.
- BELOW: the range is [TARGET, inf), RADIUS is ignored, and values below the target
    are penalized.

For all shapes, the penalty boundary options are:
- CONSTANT: constant penalty equal to STRENGTH.
- LINEAR: linear penalty that increases with slope of STRENGTH.
- QUADRATIC: quadratic penalty, STRENGTH is the first value of the penalty outside the
    range.

{reasoning_formatting}
Write your final response inside <response></response> tags, for example:

<response>
PENALTY_DEFINITION
...
END_PENALTY_DEFINITION
</response>

Do not include any extra text or comments in your final response.
```

We include the same example prompt with correct responses from different LLMs with the simple syntax:

**Box 40: Prompt**

```
Write a compositional penalty block that penalizes more than 5 prolines.
Use a linear boundary with slope of 10.
```

**Box 41: Qwen3 235B Instruct (CoT)**

```
PENALTY_DEFINITION
TYPE P
SHAPE ABOVE
TARGET 5
RADIUS 0
BOUNDARY LINEAR
STRENGTH 10
END_PENALTY_DEFINITION
```

**Box 41: Gemini 2.5 Flash**

```
PENALTY_DEFINITION
TYPE P
SHAPE ABOVE
TARGET 5
RADIUS 0
BOUNDARY LINEAR
STRENGTH 10
END_PENALTY_DEFINITION
```

**Box 41: Sonnet 4.5**

```
PENALTY_DEFINITION
TYPE P
SHAPE ABOVE
TARGET 5
RADIUS 0
BOUNDARY LINEAR
STRENGTH 10
END_PENALTY_DEFINITION
```

