# OpenReview forum: "Protein Design with Agent Rosetta: A Case Study for Specialized Scientific Agents"
_ICML.cc/2026/Conference — ICML 2026 regular_

### Official Review · Reviewer_NTmc · 2026-02-28

**Soundness:** 3
**Presentation:** 3
**Significance:** 3
**Originality:** 3
**Overall Recommendation:** 4
**Confidence:** 4

**Summary:**

Agent Rosetta is an LLM-based multi-turn agentic framework that operates the Rosetta macromolecular modeling suite through a structured, Gym-like RosettaScripts environment, enabling iterative protein design via sequential reasoning and error correction. The central finding is that prompt engineering alone fails for reliable RosettaScripts syntax generation, and that a simplified action abstraction with a Python transpiler raises success rates to near-perfect across all tested models. The system matches ProteinMPNN on fixed-backbone sequence design and outperforms a fixed expert protocol on non-canonical amino acid insertion.

**Compliance With Llm Reviewing Policy:**

Affirmed.

**Final Justification:**

I checked the authors' responses to the comments from other reviewers. As I previously indicated, I have no objections to the acceptance of this paper should the other reviewers express their support. Accordingly, I wish to recommend its acceptance.

**Key Questions For Authors:**

## Questions

- Even a small-scale iterative human comparision on one or two structures, with matched ensemble sizes, would substantially strengthen the parity claim. Is such an experiment feasible to add?

- The system prompt explicitly instructs the agent to produce reasoning that "expert biomolecular scientists can understand, review, and criticize," positioning interpretability as a core feature. Has any such expert evaluation actually been conducted on representative trajectories, and were the action choices and parameter selections judged to reflect sound biophysical reasoning?

**Strengths And Weaknesses:**

## Strengths

- The syntax simplification experiment provides a concrete, quantified negative result: original RosettaScripts composition penalties cause frequent silent semantic errors even with documentation provided, while the simplified abstraction achieves near-perfect success at much lower cost. This finding is transferable to anyone who want to build agents on top of domain-specific scientific software.

- Evaluating on non-canonical amino acid design is well-motivated, since mainstream ML models simply cannot handle this setting. Using AlphaFold 3 for post hoc validation adds independent corroboration that goes beyond Rosetta's internal metrics.

## Weaknesses

- The claim of parity with human scientists rests on fixed, single-run baseline protocols that the authors themselves describe as reflecting "a scientist's first few days of work." Even a reduced-scale iterative human baseline, say three rounds of revision on one or two structures, would give a more honest upper bound, but this is absent. Agent Rosetta's much larger evaluation budget further makes the comparision inherently unequal.

- The staged protocol, described as standard expert practice, underperforms the simpler one-shot protocol on the large majority of test structures, which is contrary to expectation. This reversal is not analyzed in the paper, and raises doubts about wether the human baselines adequately represent actual expert capability.

- The paper lacks a baseline that replaces LLM parameter selection with automated search within the same iterative framework. Without this control, it is not clear how much of the gain comes from LLM biophysical reasoning versus scaling iterations. Can you design some random policy and try to improve the result with test-time scaling?

- ProteinMPNN has negligible inference cost, so it can generate thousands of sequences for the price of a single agent trajectory. Has the authors considered a best-of-N comparision where ProteinMPNN is given the same total number of candidate evaluations as Agent Rosetta? The current single-trial comparision may understate ProteinMPNN's true capability under matched compute.

- ESMFold RMSD serves double duty as both the Pareto selection criterion at each step and the final evaluation metric, creating a circular improvment loop. The paper also acknowledges that ESMFold is biased toward naturalistic sequences, yet the agent optimizes directly for this biased metric, which makes the comparision with ProteinMPNN hard to interpret.

---

> ### Author Rebuttal · Authors · 2026-03-29
>
> We sincerely thank the reviewer for their constructive comments. We have performed several new experiments that will be included in the revised version of the paper.
>
> We look forward to clarifying any outstanding questions.
>
> ---
> # Iterative human baseline
> > A small-scale iterative comparison would strengthen evidence.
>
> Thank you for this question. To address it, **we have iterated on the one-shot human protocol for 9KGY and 9C14**:
> 1. First, we added a hydrogen bond score term (hydrogen bond networks are important for stability),
> 2. Then, we increased the weights of `hbon_lr_bb` and `hbond_sr_bb` to capture $\beta$-sheet and $\alpha$-helical hydrogen bonding based on the geometry of the two conformations.
>
> Neither modification improved performance relative to the agent's:
>
> |pdb|iteration|ESMFold RMSD (Å)|ESMFold pLDDT|
> |-|:-:|:-:|:-:|
> |9KGY|1|20.9|69|
> |9KGY|2|20.9|69|
> |9C14|1|25.2|83|
> |9C14|2|23.8|84|
>
> Adding the hydrogen bond score term led the design protocols to perform worse than the base protocol without the term. As an additive term, its relative weight can disproportionately influence the total score. Compared to other design score terms, the optimal weighting for hydrogen bonding remains largely a matter of trial and error, a tedious process for scientists.
>
> ---
> # Comparison with ProteinMPNN and randomized baseline
> > ESMFold RMSD serves double duty.
>
> We thank the reviewer for raising this point. To address it, we have:
> 1. Evaluated designs with AF3 and MD simulations (see Reviewer aYPJ),
> 2. Ablated ESMFold metrics from the context of the LLM (see Reviewer NewJ).
>
> > The comparison may understate ProteinMPNN's capability.
>
> As suggested by the reviewer, we have:
> 1. Increased sampling from ProteinMPNN to match ensemble size as the agent: 128 designs/step * 16 steps = 2048 designs,
> 2. Repeated sampling 16 times: the same number of traces as the agent,
> 3. Performed best-of-n analysis over 1,000 bootstrap replicates of 8 traces out of the 16. As suggested by Reviewer MVZK, we did not filter designs at 85 pLDDT.
>
> > Can you design a random policy and improve result?
>
> We thank the reviewer for this question. In addition to the naive baseline that iteratively invokes FastDesign without custom parameters, **we have included a randomized baseline** that, at each step of a trajectory:
> - Randomly chooses between rotamer/backbone change,
> - If rotamer change is chosen, it performs FastDesign without custom parameters,
> - If backbone change is chosen, it performs one of Small, Shear, or Backrub at random with default parameters.
>
> We summarize results of the updated analysis below. We test the statistical significance of the difference between our agent with all LLMs and the randomized baseline with a one-sided paired Wilcoxon signed-rank test. We control significance with family-wise error rate at level 0.05.
>
> |method|ESMFold RMSD (Å)|ESMFold pLDDT|agent vs. random (RMSD/pLDDT)|
> |-|:-:|:-:|:-:|
> |naive baseline|0.93 ± 0.60|88 ± 7|-|
> |random baseline|0.64 ± 0.35|91 ± 4|-|
> |||||
> |ProteinMPNN|0.58 ± 0.24|94 ± 2|-|
> |||||
> |Gemini 2.5 Flash|0.57 ± 0.31|93 ± 3|✅ / ✅
> |GPT-5|0.61 ± 0.33|92 ± 3|✅ / ❌
> |Sonnet 4.5|0.62 ± 0.35|92 ± 0.03|✅ / ✅
> |Qwen3 235b|0.61 ± 0.32|92 ± 0.03|✅ / ✅
> |||||
> |Qwen3 235b (w/o ESMFold metrics)|0.61 ± 0.36|92 ± 0.03|✅ / ✅
>
> **We found that:**
> - **Agent Rosetta remains competitive with ProteinMPNN when matching compute and including all designs (also those with pLDDT < 85),**
> - **Agent Rosetta performs better than the random baseline in terms of RMSD and pLDDT (except with GPT-5, for which pLDDT is not significantly higher),**
> - **Importantly, Agent Rosetta performs better than the random baseline even when ESMFold metrics are not included in the context of the LLM.**
>
> This updated analysis significantly strengthens empirical evidence in support of the agent.
>
> ---
> # Interpretability of reasoning traces
> > The system prompt explicitly instructs the agent to produce reasoning that "expert biomolecular scientists can understand, review, and criticize". Has any such expert evaluation actually been conducted?
>
> This is a great question! The choice of wording was motivated by two factors:
> 1. We anecdotally found that prompting as if the agent was a member of a team did lead to generally longer reasoning traces. This was helpful during development to make sure the agent was following instructions, notwithstanding the soundness of the traces from a biophysical perspective.
> 2. We did show and discuss the reasoning traces with expert scientists during development, mostly to qualitatively assess the most apparent failure modes and hurdles for the agent. It was one of these conversations that prompted us to simplify the syntax of penalty definition blocks.
>
> We are happy to expand on the plausibility of the reasoning traces from a biophysical perspective, but we did not perform any quantitative analysis that would lead to specific interpretability findings.

---

> > ### Author Rebuttal · Reviewer_NTmc · 2026-04-02
> >
> > I thank the authors for the detailed reply. Although I keep my original score, I hold no opposition if other reviewers support acceptance.

---

> > > ### Author Response · Authors · 2026-04-05
> > >
> > > We sincerely thank the reviewer for their constructive feedback and suggestions of additional experiments, and for acknowledging our rebuttal partially resolved their concerns.
> > >
> > > We will make sure to include the iterative human baseline results and the expanded comparison with ProteinMPNN and the randomized baseline in the revised version of the paper.
> > >
> > > We would be happy to address any outstanding questions or concerns!

---

### Official Review · Reviewer_NewJ · 2026-03-09

**Soundness:** 3
**Presentation:** 3
**Significance:** 3
**Originality:** 3
**Overall Recommendation:** 5
**Confidence:** 3

**Summary:**

This paper benchmarks multiple LLM agents on using Rosetta, a dominant computational protein design software, for two protein design tasks: stabilizing backbone conformations and designing non-canonical amino acids.

**Compliance With Llm Reviewing Policy:**

Affirmed.

**Final Justification:**

This paper presents an interesting and crucial direction for constructing agentic computational protein engineering tasks, and will be a valuable resource for the biotech/AI4Sci community in designing benchmarks for automated bio-engineering. Thus, I recommend acceptance.

**Key Questions For Authors:**

1. How were the proteins selected from the PDB, and have you taken steps to ensure the test proteins were not seen during pretraining of the evaluated models?

2. The agent receives the ESMfold prediction error as an intermediate signal during design, and the final evaluation is also based on ESMfold predictions. Does this mean the agent is effectively optimizing directly toward the evaluation metric? Does the human baseline receive the same signal?

3. How sensitive are the results to the choice of proteins?

4. Are you going to open-source the benchmark?

**Limitations:**

Yes

**Strengths And Weaknesses:**

## Strengths

1. The paper is well-motivated and tackles an interesting problem. Applying LLMs to computational protein design is a promising direction.

2. The benchmarks are constructed with solid domain knowledge and careful experiment analysis. For instance, the authors abstracted the syntax of composition penalties and TaskOperations to simplify their use, which substantially improves success rates.

3. The paper is well-structured, sufficiently detailed for reproducibility, and clearly written.

## Weaknesses

1. Given that fewer than 10 proteins are used for each task, it's hard to know how much we can generalize from these results. It's also unclear how these proteins were selected from the PDB, and whether some models may have memorized them.

2. The agent receives intermediate feedback signals during design (e.g., the error between the ESMfold predicted fold and the initial backbone conformation), which may make the task easier than real-world settings and raises fairness concerns. Since the final evaluation also uses ESMfold predictions, this essentially gives the agent access to an intermediate version of the ground truth signal. It's also unclear whether the human baseline receives the same signal; if not, the comparison may be unfair.

3. The human baseline is weaker than it sounds. It uses human-written protocols executed automatically, rather than a human actually running each case interactively. The authors acknowledge this as a limitation.

4. More ablation studies would be valuable, for example:
* Which types of proteins do LLMs tend to perform better on?
* How much does performance drop when the intermediate error signal is withheld?

---

> ### Author Rebuttal · Authors · 2026-03-29
>
> We sincerely thank the reviewer for their encouraging comments. We have performed additional experiments to address the reviewer's main concerns. Updated results and discussion will be included in the revised version of the paper.
>
> We look forward to clarifying any outstanding questions.
>
> ---
> # Selection of PDB structures
> > It is hard to know how much we can generalize from these results.
>
> We agree that generalization is central. We evaluated Agent Rosetta on sequence design with CAAs, and then demonstrated the same environment works for design with NCAAs. We have included several new experiments in our rebuttal to strengthen evidence. **Based on this, we expect our results to hold when LLMs have seen related scientific literature and the scientific software has a structured interface.**
> > It is unclear how these proteins were selected from the PDB.
>
> This is an important point:
> - For sequence design with CAAs, **we used a release-date cutoff of May 15, 2025 to guarantee the exact structures were not in ProteinMPNN's training set (released Aug. 2022).** For LLMs, we do not input the PDB in the context, so it is unlikely they could identify them.
> - For the NCAA task, we selected *de novo* protein structures to assess the agent on diverse structural contexts. We do not threshold by  a release-date cutoff . The same argument holds for LLMs.
>
> We will clarify this in the revised version of the paper.
>
> ---
> # Ablation of ESMFold signal
> > Since the final evaluation also uses ESMFold predictions, the comparison may be unfair.
>
> Thank you for this question. First, we note that **ESMFold was used during development of the human baseline protocols for quality evaluation. This makes comparison fair.**
>
> As suggested by the reviewer, we repeated evaluation on the CAA task with Qwen3-235b-a22b-2507 without providing the ESMFold RMSD and pLDDT in the context of the LLM (we could not evaluate all models for time and resource constraints). At each step, however, we still selected the Pareto-optimal candidates in terms of ESMFold quality metrics.
>
> **We found that removing ESMFold quality metrics from the context of the LLM did not degrade performance:**
> |ESMFold metrics|ESMFold RMSD (Å)|ESMFold pLDDT|
> |-|:-:|:-:|
> |Yes|0.6 ± 0.3|92 ± 2|
> |No|0.6 ± 0.4|92 ± 3|
>
> These additional results show that the agent is robust to ablation of the intermediate quality metrics, an important point we will include in the revised version of the paper.
>
> To strengthen experimental evidence, we have also included:
> - Environment ablations (see Reviewer aYPJ),
> - AF3 validation and MD simulations for the CAA task (see Reviewer aYPJ),
> - MD simulations for the NCAA task (see Reviewer MVZK),
> - A randomized baseline (see Reviewer NTmc).
>
> ---
> # Diverse proteins
> > How sensitive are the results to the choice of proteins?
> For sequence design with canonical residues, we found 8VWO to be a particularly challenging conformation for all methods. We speculate this might be because the original structure is a ubiquitin receptor, but we removed the binding peptide before sequence design. This might affect overall stability of the structure.
>
> For the NCAA task, the designed transmembrane beta-barrel 8UZL has the lowest TRF inclusion rate. 8UZL does not present an obvious core, which makes inclusion of TRF harder. Specifically, the most common failure mode we observed is as follows: at iteration t, the agent selects the core of the candidate designs and defines compositional penalties to favor exactly one TRF in those positions. However, after rotamer optimization, the inclusion of TRF in those positions has changed the location of the core. After several iterations of failed attempts, the agent is able to find positions that allow TRF without perturbing its conformation.
>
> We will expand on these points in the revised version of the paper.
>
> ---
> # Iterative human baseline
> > The human baseline uses human-written protocols rather than a human actually running each case interactively.
>
> Thank you for letting us elaborate on this point. We stress that protein design is a trial-and-error process, and it may take scientists weeks to finetune protocols adaptively. A major limitation to iterative improvement is that given a starting ensemble of candidate designs, it is difficult for scientists to predict how the ensemble will evolve under simple protocol modifications. Only after several time-consuming attempts, are scientists able to narrow down successful strategies. This hardness precisely motivates the need of our agent, capable of exploring strategies at a much larger scale and faster pace.
>
> **In this work, we consider parity in terms of "equal amount of human work". The human baselines are the result of a few days of work of scientists with years of experience with Rosetta. They are not initial guesses, but the outcome of a few iterations.**
>
> ---
> # Public release
> > Are you going to release the benchmark?
>
> Yes, we will release all code, traces, and designs.

---

> > ### Author Rebuttal · Reviewer_NewJ · 2026-04-03
> >
> > Thank the authors for the rebuttal. My concerns regarding weaknesses 1 and 2 are fully resolved. Weakness 3 remains, but I agree with the authors' justification. I have adjusted my score accordingly.

---

> > > ### Author Response · Authors · 2026-04-05
> > >
> > > We sincerely thank the reviewer for their consideration of our rebuttal, and for acknowledging all concerns have been addressed adequately.
> > >
> > > We will make sure to include the iterative human baseline experiment and to expand on weakness 3 in the revised version of the paper!

---

### Official Review · Reviewer_aYPJ · 2026-03-12

**Soundness:** 3
**Presentation:** 3
**Significance:** 2
**Originality:** 2
**Overall Recommendation:** 4
**Confidence:** 5

**Summary:**

This paper introduces Agent Rosetta, an LLM-based scientific agent designed to operate the Rosetta molecular modeling suite for protein design tasks. The system integrates a large language model with a structured execution environment that allows the agent to iteratively generate, execute, and refine Rosetta commands. The authors position this framework as a general interface enabling non-expert users to perform advanced protein design workflows while retaining the flexibility of physics-based modeling.

The evaluation considers two settings: (1) protein design with canonical amino acids, where the system is compared with specialized machine-learning models and expert-engineered pipelines, and (2) design tasks involving non-canonical residues, where most ML methods struggle due to limited training data and representation constraints. The authors report that Agent Rosetta can match the performance of existing specialized approaches in the canonical case and produce viable solutions in the non-canonical regime. The paper argues that the main contribution lies in demonstrating how properly structured environments allow LLM agents to operate complex scientific software effectively.

**Compliance With Llm Reviewing Policy:**

Affirmed.

**Key Questions For Authors:**

To what extent does performance depend on the specific LLM used? Would smaller models with the same environment achieve similar results?

Can the authors provide ablations isolating the effect of the structured environment compared to prompt-only tool usage?

How robust is the agent across repeated runs of the same design task? What is the variance in outcomes?

How does the system compare with modern generative protein design models such as diffusion-based or structure-conditioned approaches?

What are the main failure modes observed when the agent attempts complex Rosetta workflows?

**Limitations:**

The paper focuses on feasibility demonstrations rather than rigorous evaluation. The experiments cover a limited range of design tasks and do not fully explore robustness, reproducibility, or scalability. Additionally, the approach relies heavily on the availability of a powerful LLM, which may limit accessibility and reproducibility for some users.

**Strengths And Weaknesses:**

Strengths

The paper addresses an emerging direction in machine learning: LLM-driven scientific agents capable of interacting with domain-specific tools. The choice of protein design as a testbed is compelling because Rosetta represents a widely used but complex modeling environment that is difficult for non-experts to operate. Demonstrating that an LLM agent can orchestrate design workflows within such a system is a relevant and timely contribution.

A second strength is the structured environment design that constrains and guides the agent’s interaction with Rosetta. The authors correctly emphasize that prompt engineering alone is insufficient for reliable tool usage. The proposed environment abstraction therefore highlights an important engineering insight: reliable scientific agents require tight integration between language models and execution frameworks.

The evaluation across canonical and non-canonical design settings is also interesting. The latter scenario is particularly relevant because many ML-based protein design models rely heavily on training distributions that exclude non-standard residues. Demonstrating reasonable performance in this regime strengthens the argument that physics-based tools combined with LLM reasoning can provide broader coverage than purely data-driven approaches.

Finally, the paper is generally clear and the motivation is well articulated. The problem setup and system architecture are understandable and the empirical results provide a first demonstration that such agents can produce viable design solutions.

Weaknesses

Despite the interesting idea, the work remains primarily a systems integration effort rather than a machine learning contribution.

First, the novelty of the proposed method is limited. The paper mainly combines an existing LLM with Rosetta via a structured execution interface. While the environment design is useful from an engineering perspective, the work does not introduce new learning algorithms, optimization techniques, or theoretical insights. As a result, the contribution may be perceived as incremental relative to recent work on tool-using LLM agents.

Second, the evaluation is relatively narrow. The experiments demonstrate feasibility but do not convincingly establish that the approach advances the state of the art in protein design. For instance, comparisons against specialized ML models are limited and the metrics used to assess design quality are not deeply analyzed. It remains unclear whether the agent consistently produces designs competitive with modern generative protein design models.

Third, the role of the LLM itself is somewhat under-analyzed. The paper does not clearly separate improvements due to the structured environment from those due to the reasoning capabilities of the language model. Ablation studies examining different agent architectures or environment constraints would strengthen the claims.

Finally, scalability and reliability remain open questions. Autonomous scientific agents operating complex software pipelines can suffer from instability or brittle behavior. The paper provides limited quantitative analysis of failure modes, robustness, or reproducibility across different design tasks.

---

> ### Author Rebuttal · Authors · 2026-03-28
>
> We sincerely thank the reviewer for their overall positive assessment of our work. We have performed additional experiments to address the reviewer’s main concerns. We will update results and discussion in the revised version of the paper.
>
> We are looking forward to clarifying any outstanding questions.
>
> ---
> # Novelty
> > The contribution may be perceived as incremental.
>
> **Agent Rosetta is the first agentic system capable of designing proteins with Rosetta**. Our key finding that environment design is crucial is valuable beyond engineering considerations, informing generalizable practices across fields.
> > The experiments do not establish that the approach advances the state of the art.
>
> **Agent Rosetta advances protein design as the first autonomous system handling both canonical and non-canonical residues.** Modern generative models need extensive training data, but real-world tasks may require residues for which data collection is impossible.
>
> ---
> # Ablations
> > Can the authors provide ablations isolating the effect of the environment compared to prompt-only tool usage?
>
> This is a great point! Without the environment, LLMs need to generate RosettaScripts XML from scratch. **To answer the reviewer's question, we have designed 4 verifiable tasks of increasing difficulty, from in-context example recall to general Rosetta knowledge.** Including all possible instructions/examples in the context is infeasible, so generalization to unseen syntax is important.
>
> Each task instructs the LLM to generate parts of RosettaScripts XML scaffold usually automated by our environment. The system prompt describes the XML schema, section contents, and examples. The tasks are:
> 1. (easy) Define a custom score function with modified weights (explicit examples provided)
> 2. (medium) Store the amino acid sequence (metric name provided, no usage examples)
> 3. (medium) Store per-residue fa_rep and rama_prepro energy terms (metric name not provided, similar components mentioned)
> 4. (hard) Store cavity volume, radius of gyration (requires ad-hoc knowledge infeasible to include in prompts)
>
> We benchmarked models with 10 queries/task and report mean success rate (%):
> |model|task 1|task 2|task 3|task 4|
> |-|:-:|:-:|:-:|:-:|
> |GPT-5|100|100|40|0|
> |Gemini 2.5 Flash|90|10|0|0|
> |Qwen3 Instruct (CoT)|90|20|0|0|
> |Sonnet 4.5|90|100|90|0|
>
> **All LLMs fail sharply when tasks fall outside prompt instructions, highlighting our structured environment's crucial role in enabling sustained Rosetta interaction. Without it, frontier LLMs cannot generate valid RosettaScripts XML protocols from scratch.**
>
> ---
> # Empirical evidence
> > It remains unclear whether the agent produces competitive designs.
>
> To address this question, **we have evaluated designs on the CAA task using AF3:**
>
> |method|AF3 RMSD (Å)|AF3 pLDDT|
> |-|:-:|:-:|
> |ProteinMPNN|0.5 ± 0.2|94 ± 2|
> |GPT-5|0.8 ± 0.3|91 ± 3|
>
> and **have performed three replicate MD simulations** of 1$\mu$s each for the best design on 9PL1 due to time constraints. Median RMSD between the initial and final structures was:
>
> |method|RMSD (Å)|
> |-|:-:|
> |ProteinMPNN|2.3|
> |GPT-5|2.1|
>
> We will include results for all target conformations in the revised version of the paper.
>
> > How does the agent compare with diffusion-based approaches?
>
> **We included comparison with BoltzGen.** For time constraints, we ran one replicate of 2048 designs per target structure. Ensemble metrics with matched size were:
>
> |method|ESMFold RMSD (Å)|ESMFold pLDDT|
> |-|:-:|:-:|
> |BoltzGen|0.7 ± 0.3|93 ± 2|
> |GPT-5|0.7 ± 0.4|91 ± 2|
>
> We will repeat sampling and update results for the revised version of the paper.
>
> **These results strengthen evidence that Agent Rosetta is competitive with ProteinMPNN and modern diffusion-based models.**
>
> ---
> # Robustness
> > Would smaller models achieve similar results?
>
> **We have included results for Qwen3-30b-a3b on CAA design:**
> |method|action success (%)|ESMFold RMSD (Å)|ESMFold pLDDT|
> |-|:-:|:-:|:-:|
> |Qwen-3-30b|25 ± 11|0.7 ± 0.4|91 ± 3|
> |Qwen-3-235b|86 ± 5|0.6 ± 0.3|92 ± 2|
>
> Performance degraded compared to Qwen-3-235b, especially in action success rate. However, although smaller models may be worse out-of-the-box, our environment enables training with reinforcement learning (e.g., GRPO).
> > What are the main failure modes?
>
> Qualitatively, we identify these two major failure modes:
> 1. Misalignment between the agent's stated intention and the generated action.
> 2. RosettaScripts movers use slightly different naming conventions for arguments. Agents can get stuck in self-correction loops. This could be addressed by allowing the agent to call the `-info MoverName` flag of Rosetta, which displays the documentation for a particular mover. We consider this as a direction of future improvement.
>
> > What is the variance of the outcomes?
>
> For each task, method, and input structure, we ran the agent 16 times and subsequently performed best-of-8 analysis over 1000 bootstrap replicates. We report 95% CI and standard deviation in the Appendix.

---

### Official Review · Reviewer_MVZK · 2026-03-13

**Soundness:** 3
**Presentation:** 3
**Significance:** 3
**Originality:** 2
**Overall Recommendation:** 4
**Confidence:** 3

**Summary:**

This paper introduces Agent Rosetta, an LLM-based autonomous agent that interfaces with the Rosetta Macromolecular Modeling Suite to perform protein design tasks. The agent starts with a “design brief” and then cycles through a multi-turn interaction loop to execute a chosen Rosetta action (out of 3 actions, rotamer_change, backbone_change, and go_back_to) to produce 128 candidate designs per cycle; only Pareto-optimal designs are carried forward. The authors found that in the process of designing Agent Rosetta, environment design was critical: frontier LLMs failed to generate correct RosettaScripts at consistently successful rates, even with sophisticated prompting and correct reasoning. This motivated the creation of a simplified syntax for interfacing with Rosetta as part of the presented framework. The simplified syntax lowers costs and allows all tested models to achieve high (>98%) success rates on two tasks, fixed-backbone sequence design and non-canonical amino acid (NCAA) insertion. Results are compared with standalone ProteinMPNN and human baseline performances of varying complexity. This framework increases Rosetta accessibility and opens up Rosetta’s complexities, including NCAAs, to more researchers and LLM agents.

**Compliance With Llm Reviewing Policy:**

Affirmed.

**Final Justification:**

After the rebuttals, the soundness of the methodology is greatly boosted by the additions mentioned across all rebuttals: MD simulations, a compute-matched comparison, removal of the pLDDT filter, AF3 cross-validation, a randomized baseline, environment ablation, and failure mode analysis.

Concerns about related work are also addressed in the rebuttals.

I look forward to the revision addressing the limitations and societal implications.

**Key Questions For Authors:**

1. Does replacing the LLM agent with simple scripted policies approach the agent’s performance? In other words, is the LLM agent’s contribution significant? Given the same Pareto-optimal selection infrastructure, a policy that cycles deterministically between designing constraints and perturbing the backbone would shed light on the LLM agent’s contributions if baselined.
2. Does the environment design finding hold if PyRosetta scripts are generated rather than RosettaScripts XML? Whether models still struggle with generating PyRosetta scripts as much as RosettaScripts XML seems to be an open question; investigating this could alter the paper’s core findings (i.e. “the complexity of the tool necessitates simplifying the agent-tool interface for agent reliability” vs. “agent reliability can be achieved via Python rather than XML”).
3. What does agent behavior look like across the entire workflow? Do different LLMs pursue different strategies? How do agents fail: are there any behaviors that lead to agents getting stuck?
4. Is there a way to experimentally validate the findings in the paper, especially for the novel NCAA designs?

**Limitations:**

Some limitations are acknowledged (imperfect proxy metrics biased towards naturalistic sequences, simplified human baseline profiles). More discussion should be given to the limitation of designs scoring highly in silico but not transferring to experimental success, a central challenge of computational protein design. Restriction to three action types is not framed as a limitation; this should be framed especially in the context of claiming that the framework enables “a broad range of real-world design pipelines.”

For potential negative societal impact, **dual use concerns regarding harmful applications are not addressed.** The presented framework automates protein design workflows, which could in principle be directed towards properties that enhance virulence. A potential increase in accessibility via this framework could in turn make harmful applications more accessible, especially if they involve presented novelties like NCAA design; this should be addressed.

There also exists a risk that the agent could produce confidently wrong designs. Combined with the argument that this framework extends Rosetta access to non-experts, this could lead to progression of flawed designs into downstream applications that are not caught. This should be addressed to some extent.

Discussion of guardrails (e.g. flagging designs that contain unusual features for expert review) would address both of these concerns further.

**Strengths And Weaknesses:**

Soundness: 2
- Strengths
  - The framework is technically coherent.
    - Methods are appropriate. The Pareto-optimal selection strategy is a defensible choice.
    - Statistics are computed appropriately.
    - Experiments run correctly.
  - The bootstrap evaluation procedure is well-designed and appropriate for estimating best-of-n performance.
  - The environment design experiment comparing performance with standalone RosettaScripts generation and with the simplified syntax is clean and well-controlled.
- Weaknesses
  - Validation pipelines and thresholds may have been selected in a way that inflates reported performance.
    - Thresholds (0.20 Angstrom tolerance, filtering out pLDDT < 0.85 designs) are potentially arbitrary; this should be addressed.
    - Evaluation metrics are applied inconsistently in comparisons. For example, lower ESMFold RMSD for Agent Rosetta on the design task is presented as evidence of competitive performance, while higher ESMFold pLDDT from ProteinMPNN is explained away as potential ESMFold bias. The paper should either present a case that RMSD is more robust to this bias than pLDDT, or acknowledge that the entire ESMFold-based comparison is unreliable.
    - Selection is done on metrics (e.g. Rosetta/ESMFold RMSD and pLDDT) that are also used as “validation.” For example, the validation pipeline of the NCAA task filters successful designs based on Rosetta’s RMSD and reports/validates using AlphaFold3 (AF3)’s RMSD. More information on the distribution of AF3 RSMD across all successful designs would strengthen this point.
  - Failure rates are hidden in the process of selective filtering, and failure modes are not discussed. Reporting the fraction of filtered designs and how models fail would strengthen the findings.
  - The profiles of human baseline tasks fall short of the “expert baseline” level at which it is framed. Tasks baselined by humans are static scripts, fixed and independent of the backbone that represent “a scientist’s first few days of work” on the pipeline. This is acknowledged by the authors but still does not represent standard practice.
    - The staged protocol performs worse than the naive baseline (iterative FastDesign without LLM guidance), suggesting an error in protocol design rather than a differentiation between human expertise and LLM guidance.
  - Cost and performance comparison with ProteinMPNN is imbalanced.
    - Costs of running Rosetta’s FastDesign, especially at the scale that the agent pipeline demands, should be addressed in the body of the paper for proper comparison to standalone ProteinMPNN, even if these costs are negligible compared to LLM inference.
    - Agent Rosetta’s total cost may consist of 128 designs per step across ~15 steps, effectively generating many more sequences than the 128 runs of ProteinMPNN.
  - The action space is restrictive (3 actions) in comparison to Rosetta’s full suite of capabilities. While this is justified in the context of this paper, more evidence is needed on how the framework generalizes in order to support the claim that it enables “a broad range of real-world design pipelines.”

Presentation: 3
- Strengths
  - The writing is clear and technically precise. Figures are generally effective and relevant.
  - The paper correctly positions itself as a step toward accessibility, in addressing the known barriers of onboarding scientists without deep RosettaScripts expertise to Rosetta workflows.
  - Appendices are thorough enough to address most reproducibility questions.
- Weaknesses
  - Presentation of novelty is slightly diminished by structure. The novel environment design finding should be more central. Discussion of the more novel NCAA task is relatively compressed compared to the canonical design task.
  - Some critical reproducibility details are missing: Rosetta version, OpenRouter specifications (which model providers).
  - Related work should be addressed to some degree in the main body of the text to position Agent Rosetta against scientific agent frameworks like ProtAgents.
  - Some references are corrupted or rendered incorrectly (e.g. stray backticks in citations, a “see ??” reference).

Significance: 3
- Strengths
  - The paper addresses important and relevant problems of accessibility gaps for physics-based scientific software and how to build reliable LLM agents for specialized scientific tools.
  - The finding that environment design matters more to specialized scientific agent performance is significant and can influence future research, especially the development of future LLM agents interfacing with scientific tools.
- Weaknesses
  - The scope of impact is limited by the specialized focus of this framework on Rosetta. Generalizing the environment design finding across multiple scientific tools would strengthen this.

Originality: 2
- Strengths
  - The environment abstraction finding that reliable agent performance necessitates simplifying RosettaScripts syntax is novel and well-supported. Though narrow in scope, this is the paper’s most novel finding.
  - Applying an LLM agent to Rosetta to enable NCAA design is novel, albeit only in application target.
- Weaknesses
  - The agent architecture is standard ReAct-style and closely follows patterns from previous work (e.g. ChemCrow, SWE-Agent, ProtAgents).
  - Differentiation from the aforementioned closely related prior work is not discussed in depth. Addressing how Agent Rosetta compares to these frameworks in architecture, philosophy, etc. would greatly strengthen originality.
  - The paper lacks comparisons with sensible alternatives in architecture (e.g. generating PyRosetta scripts instead of RosettaScripts XML, which may be more extensively contained in the training data in a language LLMs are more fluent in). Addressing this would strengthen the reasoning behind the chosen combination of architecture.

---

> ### Author Rebuttal · Authors · 2026-03-31
>
> We sincerely thank the reviewer for their thorough assessment of our work. We have performed additional experiments to address the reviewer's main concerns. We will expand on discussion of all points raised by the reviewer in the revised version of the paper.
>
> We look forward to clarifying any outstanding questions.
>
> ---
> # Novelty and impact
> > Differentiation from related works should be discussed.
>
> We thank the reviewer for letting us elaborate on this. **Our work focuses on low-level integration of an LLM agent with specialized scientific software.** This reflects deployment in real-world pipelines. We differ from existing works addressing high-level planning or orchestration of several LLM-friendly tools. For example, Agent Rosetta could be invoked by ProtAgents to perform specific tasks.
> > The scope is limited to Rosetta.
>
> Our agent has the potential to reach all scientists using Rosetta. Only an in-depth exploration of tools and workflows can identify insufficiencies of prompt-only tool-usage like we have.
>
> ---
> # RosettaScripts (RS) vs PyRosetta (PyR)
> > Do findings hold if PyRosetta is used?
>
> This is a great question! Our goal is to develop a generalist agent, and RS offers a more realistic setting for production-level scientific exploration. It natively supports scaling to thousands of designs, whereas PyR requires additional engineering to match ensemble size. **Furthermore, PyR uses the same composition penalty syntax, so the major difficulty found in this work would persist.**
>
> ---
> # Validation pipeline
> > The paper should acknowledge that ESMFold-based comparison might be unreliable.
>
> ESMFold and ProteinMPNN are trained on natural protein sequences, which may introduce bias. We disclose this for the sake of transparency, not to dismiss ProteinMPNN's higher pLDDT. We will clarify this sentence to avoid confusion. **We have included AF3 as an additional validation metric.**
>
> >- Thresholds are arbitrary.
> >- Selection is done on validation metrics.
> >- The agent generates more sequences than ProteinMPNN.
> >- Do simple scripted policies approach the agent’s performance?
>
> We thank the reviewer for highlighting these important points. **We have updated our analysis to address them as well as questions from other reviewers:**
> - In our response to Reviewer NTmc, we remove filtering, match compute with ProteinMPNN, and include a random scripted baseline to quantify the contribution of the LLM.
> - In our response to Reviewer aYPj, we include AF3 (not ESMFold) validation of the designs generated by our agent compared to ProteinMPNN.
>
> **These results strengthen evidence in support of Agent Rosetta as a competitive design method.**
>
> ---
> # Experimental validation
> > Is there a way to experimentally validate NCAA designs?
>
> We agree that experimental validation is the gold-standard. Unfortunately, we could not synthesize designs in a lab. **Instead, we performed molecular dynamics simulations to compare the stability of Agent Rosetta's design with the human protocol's for the NCAA task.**
>
> Given the computational cost of MD simulations, we validated designs of GPT-5 only. For both methods, we selected the top-5 designs in terms of RMSD between the Rosetta pose and the native structure. For each design, we simulated 3 replicates of 1$\mu$s each.
>
> We measured RMSD between the simulation at 1$\mu$s and the native structure. Large RMSDs suggest unfolding due to a bad placement of TRF in the core. We report median (95% CI) across structures, ranks, and replicates:
>
> |method|RMSD (Å)|
> |-|-|
> |human|3.3 (2.6/7.6)|
> |GPT-5|2.5 (2.3/4.0)|
>
> We found the agent to be statistically better than the human (p < 1e-03, one-sided paired Wilcoxon signed-rank test).
>
> Beyond RMSD, we computed the relative change (in %) of the fraction of helices, strands, and loops in the simulated structures at 1$\mu$s. Large variations indicate unstable secondary structure. We report median (95% CI) across structures, ranks, and replicates.
>
> |method|helix|loop|strand|
> |-|-|-|-|
> |human|3 (0/37)|22 (3/137)|18 (2/73)|
> |GPT-5|2 (0/10)|13 (1/60)|8 (1/22)|
>
> We found the agent to be statistically better than the human across all types of secondary structure with family-wise error rate of 0.05.
>
> ---
> # Human baseline
> > The staged protocol performs worse than the naive baseline.
>
> The naive baseline is primarily included for comparison with the agent. We have carefully reviewed the protocols, and they are correct.
>
> **We explored an iterative human baseline in our response to Reviewer NTmc. We found iterative human refinement to be brittle.**
>
> ---
> # Behavior
> > Do LLMs pursue different strategies?
>
> Yes, we will include transition matrices in the revised version of the paper to evaluate this point quantitatively. **We found that LLMs can get stuck in self-correction loops.** We will expand on this in the revised version of the paper.
>
> ---
> # Harmful usage
> Thank you for raising this important consideration. **We will discuss this in the revised version of the paper.**

---

> > ### Author Rebuttal · Reviewer_MVZK · 2026-04-04
> >
> > I thank the authors for their thorough and thoughtful replies across all reviews. The soundness of the methodology is greatly boosted by the addition of MD simulations, a compute-matched comparison, removal of the pLDDT filter, AF3 cross-validation, a randomized baseline, environment ablation, and failure mode analysis.
> >
> > The clarification and positioning with regards to higher-level related work is much appreciated as well.
> >
> > I look forward to the revision addressing the limitations and societal implications.

---

> > > ### Author Response · Authors · 2026-04-05
> > >
> > > We sincerely thank the reviewer for their constructive feedback and for acknowledging our rebuttal addressed all their concerns adequately.
> > >
> > > We will make sure to expand on limitations and societal implications in the revised version of the paper!

---

### Decision · Program_Chairs · 2026-04-30

**Decision:**

Accept (regular)

**Comment:**

This paper introduces Agent Rosetta, an LLM agent framework that utilizes structured execution environment to reliably operate the physics-based Rosetta software, overcoming the limitation of purely data-driven ML models (e.g., ProteinMPNN) in designing proteins with non-canonical amino acids. While the majority of the reviewers note limited ML novelty (e.g., standard ReAct-style architecture) and initially questioned the use of static, single-run script as "expert" baselines, they believe the application domain is highly important, and agree that a structured syntax abstraction is necessary over prompt-engineering alone. The reviewers raised important questions regarding the circularity of using ESMFold for evaluation and the fairness of the computational comparison against baselines like ProteinMPNN, and the authors successfully addressed that with evidence that the agent remains competitive under a matched compute budget and effectively operates even when ESMFold metrics are ablated from the context. Furthermore, they provided evidence through rigorous Molecular Dynamics simulations that the agent's non-canonical designs are statistically more stable than those produced by human protocols.
Overall, while the ML components are known, their systems-level integration into a strictly physics-grounded environment lowers accessibility barriers for a crucial bioinformatics tool and effectively handles design with non-canonical amino acids. Therefore, the merits of this work outweigh the remaining weaknesses.